# VARIATIONAL SEARCH DISTRIBUTIONS

**Daniel M. Steinberg, Rafael Oliveira, Cheng Soon Ong & Edwin V. Bonilla**
Data61, CSIRO, Australia
{dan.steinberg, rafael.dossantosdeoliveira, cheng-soon.ong, edwin.bonilla}@data61.csiro.au

## ABSTRACT

We develop variational search distributions (VSD), a method for conditioning a generative model of discrete, combinatorial designs on a rare desired class by efficiently evaluating a black-box (e.g. experiment, simulation) in a batch sequential manner. We call this task active generation; we formalize active generation's requirements and desiderata, and formulate a solution via variational inference. VSD uses off-the-shelf gradient based optimization routines, can learn powerful generative models for desirable designs, and can take advantage of scalable predictive models. We derive asymptotic convergence rates for learning the true conditional generative distribution of designs with certain configurations of our method. After illustrating the generative model on images, we empirically demonstrate that VSD can outperform existing baseline methods on a set of real sequence-design problems in various protein and DNA/RNA engineering tasks.

## 1 INTRODUCTION

We consider a variant of the active search problem (Garnett et al., 2012; Jiang et al., 2017; Vanchinathan et al., 2015), where we wish to find members (designs) of a rare desired class in a batch sequential manner with a fixed black-box evaluation (e.g. experiment) budget. We call sequential active learning of a *generative* model of these designs **active generation**. Examples of rare designs are compounds that could be useful pharmaceutical drugs, or highly active enzymes for catalyzing chemical reactions. We assume the design space is discrete or partially discrete, high-dimensional and practically *innumerable*. For example, the number of possible configurations of a single protein is $20^{\mathcal{O}(100)}$ (see, e.g., Sarkisyan et al., 2016). Learning a generative model of these designs allows us to circumvent the need for traversing the whole search space.

We are interested in this active generation objective for a variety of reasons. We may wish to study the properties of the "fitness landscape" (Papkou et al., 2023) to gain a better scientific understanding of a phenomenon such as natural evolution. Or, we may not be able to completely specify the constraints and objectives of a task, but we would like to characterize the space of, and generate new feasible designs. For example, we want enzymes that can degrade plastics in an industrial setting, but we may not yet know the exact conditions (e.g. temperature, pH), some of which may be anti-correlated with enzyme catalytic activity. Alternatively, if we know these multiple objectives and constraints, we may only want to generate designs from a Pareto set.

Assuming we can take advantage of a prior distribution over designs, we formulate the search problem as inferring the posterior distribution over rare, desirable designs. Importantly, this posterior can be used for *generating new designs*. Specifically, we use (black-box) variational inference (VI) (Ranganath et al., 2014), and so refer to our method as variational search distributions (VSD). Our major contributions are: (1) we formulate the batch active generation objective over a (practically) innumerable discrete design space, (2) we present a variational inference algorithm, VSD, which solves this objective, (3) we show that VSD performs well theoretically and empirically, and (4) we connect active generation to other recent advances in black-box optimization (BBO) of discrete sequences that use generative models. VSD uses off-the-shelf gradient based optimization routines, is able to learn powerful generative models, and can take advantage of scalable predictive models. In our experiments we show that VSD can outperform existing baseline methods on a set of real applications. Finally, we evaluate our approach on the related sequential BBO problem, where we want to find the globally optimal design for a specific objective and show competitive performance when compared with state-of-the-art methods, e.g., based on latent space optimization (LSO) (Gruver et al., 2023).

## 2 METHOD

In this section we formalize our problem and describe its requirements and desiderata. We also develop our proposed solution, based on variational inference, which we will refer to as variational search distributions (VSD).

### 2.1 THE PROBLEM OF ACTIVE GENERATION

We are given a design space $\mathcal{X}$, which can be discrete or mixed discrete-continuous and high dimensional, and where for each instance that we choose $\mathbf{x} \in \mathcal{X}$, we measure some corresponding property of interest (so-called fitness) $y \in \mathbb{R}$. For example, in our motivating application of DNA/RNA or protein sequences, $\mathcal{X} = \mathcal{V}^M$, where $\mathcal{V}$ is the sequence vocabulary (e.g., amino acid labels, $|\mathcal{V}| = 20$) and $M$ is the length of the sequence. However, we do not limit the application of our method to sequences. Using this framing, a real world experiment (e.g., measuring the activity of an enzyme) can be modeled as an unknown relationship,

$$y = f_\bullet(\mathbf{x}) + \epsilon, \tag{1}$$

for some black-box function (the experiment), $f_\bullet$, and measurement error $\epsilon \in \mathbb{R}$, distributed according to $p(\epsilon)$ with $\mathbb{E}_{p(\epsilon)}[\epsilon] = 0$. Instead of modeling the whole space, $\mathcal{X}$, we are only interested in a set of events which we choose based on fitness, $\mathcal{S} \subset \mathcal{X}$. In particular for active generation we wish to learn a generative model, $q(\mathbf{x})$, that only returns samples $\mathbf{x}^{(s)} \in \mathcal{S}$ by efficiently querying the black-box function in Equation 1. We assume that $\mathcal{S}$ are rare events in a high dimensional space, and that we have access to a prior belief, $p(\mathbf{x})$, which helps narrow in on this subset of $\mathcal{X}$. We are given an initial dataset, $\mathcal{D}_N := \{(y_n, \mathbf{x}_n)\}_{n=1}^N$, which may contain only a few instances of $\mathbf{x}_n \in \mathcal{S}$. Given $p(\mathbf{x})$ and $\mathcal{D}_N$ we aim to generate batches of unique candidates, $\{\mathbf{x}_{bt}\}_{b=1}^B$, for black-box (experimental) evaluation in a series of rounds, $t \in \{1, \ldots, T\}$, where $B = \mathcal{O}(1000)$ and we desire $\mathbf{x}_{bt} \in \mathcal{S}$. After each round $\mathcal{D}_N$ is augmented with the black-box results of the batch, i.e. $\mathcal{D}_N \leftarrow \mathcal{D}_N \cup \{(\mathbf{x}_{bt}, y_{bt})\}_{b=1}^B$. As we shall see later, our solution allows us to satisfy the following requirements and additional desiderata for active generation.

**Requirements & Desiderata.** *Active generation requirements (R) and other desiderata (D).*

**(R1) Rare** *feasible designs, $\mathcal{S}$, are rare events in $\mathcal{X}$ that need to be identified*

**(R2) Sequential** *non-myopic candidate generation, $\mathbf{x} \in \mathcal{S}$, for sequential black-box evaluation*

**(R3) Discrete** *search over (combinatorially) large design spaces, e.g. $\mathbf{x} \in \mathcal{X} = \mathcal{V}^M$*

**(R4) Batch** *generation of up to $\mathcal{O}(1000)$ diverse candidate designs per round*

**(R5) Generative** *models, $\mathbf{x}^{(s)} \sim q(\mathbf{x})$, that are task-specific for rare, feasible designs*

**(D1) Guaranteed** *convergence for certain choices of priors, variational distributions and predictive models*

**(D2) Gradient** *based optimization strategies for candidate searching*

**(D3) Scalable** *predictive models that enable high-throughput evaluation/experiments.*

Like active search (Garnett et al., 2012) in our case we are interested in the solution space of the super level-set, $\mathcal{S}_{\text{SLS}} := \{\mathbf{x} : f_\bullet(\mathbf{x}) > \tau\}$ for a threshold $\tau \in \mathbb{R}$ (e.g., wild-type fitness). As we only have access to noisy measurements, $y$, *our task is to estimate the super level-set distribution, $p(\mathbf{x}|y > \tau)$, using active generation*. Estimating this distribution over $\mathcal{S}_{\text{SLS}}$ is computationally and statistically challenging and, therefore, we cast this as a *variational inference* problem. We also consider the case of BBO for which $\mathcal{S}_{\text{BBO}} := \operatorname{argmax}_{\mathbf{x}} f_\bullet(\mathbf{x})$, and we show that we can accommodate this in our variational framework by iteratively raising $\tau_t$ per round, $t$. We visualize the properties and models involved in active generation as applied to a continuous "fitness landscape" in Figure 1.

### 2.2 VARIATIONAL SEARCH DISTRIBUTIONS

We cast the estimation of $p(\mathbf{x}|y > \tau)$ as a sequential optimization problem. A suitable objective for a round, $t$, is to minimize a divergence,

$$\phi_t^* = \operatorname*{argmin}_\phi \mathbb{D}[p(\mathbf{x}|y > \tau)\|q(\mathbf{x}|\phi)] \tag{2}$$

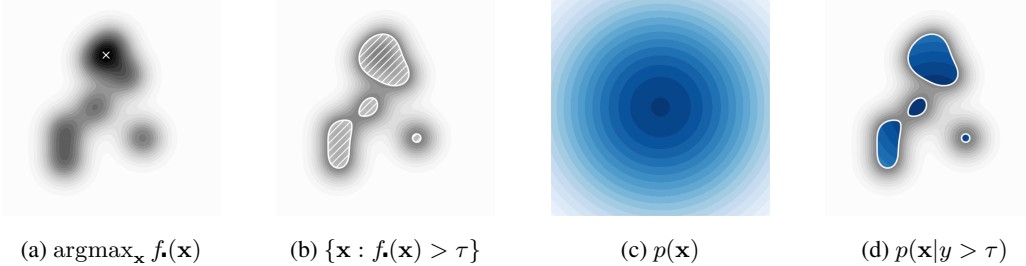

(a) $\text{argmax}_{\mathbf{x}}\, f_{\bullet}(\mathbf{x})$    (b) $\{\mathbf{x} : f_{\bullet}(\mathbf{x}) > \tau\}$    (c) $p(\mathbf{x})$    (d) $p(\mathbf{x}|y > \tau)$

Figure 1: Fitness landscape properties and models. (a) A noise-less fitness landscape, $f_{\bullet}(\mathbf{x})$, and the maximum fitness design, $\mathcal{S}_{\text{BBO}} = \{\mathbf{x}^*\}$, as the white '$\times$'. (b) The super level-set, $\mathcal{S}_{\text{SLS}}$, of all fit designs as the white hatched area. (c) Prior belief $p(\mathbf{x})$. (d) The density/mass function of the super level-set, $p(\mathbf{x}|y > \tau)$, as blue contours. Our goal is to sequentially estimate a generative model for the distribution of the super level-set (d). We assume a noisy relationship between $f_{\bullet}$ and $y$, so the super level-set will not have a hard boundary, and $p(\mathbf{x}|y > \tau)$ will be defined over all $\mathcal{X}$.

where $q(\mathbf{x}|\phi)$ is a parameterized distribution from which we sample candidate designs $\mathbf{x}_{bt}$, (R5), and which we aim to match to $p(\mathbf{x}|y > \tau)$. The difficulty is that we cannot directly evaluate or empirically sample from $p(\mathbf{x}|y > \tau)$. However, if we consider the reverse Kullback-Leibler (KL) divergence,

$$\underset{\phi}{\text{argmin}}\; \mathbb{D}_{\text{KL}}[q(\mathbf{x}|\phi)\|p(\mathbf{x}|y > \tau)] = \underset{\phi}{\text{argmin}}\; \mathbb{E}_{q(\mathbf{x}|\phi)}\left[\log \frac{q(\mathbf{x}|\phi)}{p(\mathbf{x})} - \log p(y > \tau|\mathbf{x})\right], \quad (3)$$

where we have expanded $p(\mathbf{x}|y > \tau)$ using Bayes rule and dropped the constant term $p(y > \tau)$, we note that we no longer require evaluation of $p(\mathbf{x}|y > \tau)$ directly. We recognize the right-hand side of Equation 3 as the well known (negative) variational evidence lower bound (ELBO),

$$\mathcal{L}_{\text{ELBO}}(\phi) := \mathbb{E}_{q(\mathbf{x}|\phi)}[\log p(y > \tau|\mathbf{x})] - \mathbb{D}_{\text{KL}}[q(\mathbf{x}|\phi)\|p(\mathbf{x})]. \quad (4)$$

For this we assume access to a prior distribution over the space of designs, $p(\mathbf{x})$, that may be informed from the data at hand (or pre-trained). Henceforth, as we will develop a sequential algorithm, we will denote this prior as $p(\mathbf{x}|\mathcal{D}_0)$. We note the relationship between $\log p(y > \tau|\mathbf{x})$ and the probability of improvement (PI) acquisition function from Bayesian optimization (BO) (Kushner, 1964),

$$\log p(y > \tau|\mathbf{x}) \approx \log p(y > \tau|\mathbf{x}, \mathcal{D}_N) = \log \mathbb{E}_{p(y|\mathbf{x}, \mathcal{D}_N)}[\mathbb{1}[y > \tau]] = \log \alpha_{PI}(\mathbf{x}, \mathcal{D}_N, \tau). \quad (5)$$

Here $\mathbb{1} : \{\text{false}, \text{true}\} \rightarrow \{0, 1\}$ is the indicator function and $p(y|\mathbf{x}, \mathcal{D}_N)$ is typically estimated using the posterior predictive distribution of a Gaussian process (GP) given data, $\mathcal{D}_N$. So $p(y > \tau|\mathbf{x}, \mathcal{D}_N) = \Psi((\mu_N(\mathbf{x}) - \tau)/\sigma_N(\mathbf{x}))$, where $\Psi(\cdot)$ is a cumulative standard normal distribution function, and $\mu_N(\mathbf{x}), \sigma_N^2(\mathbf{x})$ are the posterior predictive mean and variance, respectively, of the GP. We refer to this estimation strategy as GP-PI, and rewrite the ELBO accordingly,

$$\mathcal{L}_{\text{ELBO}}(\phi, \tau, \mathcal{D}_N) = \mathbb{E}_{q(\mathbf{x}|\phi)}[\log \alpha_{PI}(\mathbf{x}, \mathcal{D}_N, \tau)] - \mathbb{D}_{\text{KL}}[q(\mathbf{x}|\phi)\|p(\mathbf{x}|\mathcal{D}_0)]. \quad (6)$$

We refer to the method that maximizes the objective in Equation 6 as variational search distributions (VSD), since we are using the variational posterior distribution as a means of searching the space of fit designs, satisfying (R1), (R2) and (R4). It is well known that when the true posterior is a member of the variational family indexed by $\phi$, the above variational inference procedure has the potential to recover the exact posterior distribution. To recommend candidates for black-box evaluation we sample a set of designs from our search distribution each round,

$$\{\mathbf{x}_{bt}\}_{b=1}^{B} \sim \prod_{b=1}^{B} q(\mathbf{x}|\phi_t^*), \quad \text{where} \quad \phi_t^* = \underset{\phi}{\text{argmax}}\, \mathcal{L}_{\text{ELBO}}(\phi, \tau, \mathcal{D}_N). \quad (7)$$

We discuss the relationship between VSD and BO in Appendix G. In general, because of the discrete combinatorial nature of our problem, we cannot use the re-parameterization trick (Kingma & Welling, 2014) to estimate the gradients of the ELBO straightforwardly. Instead, we use the score function gradient, also known as REINFORCE (Williams, 1992; Mohamed et al., 2020) with standard gradient descent methods (D2) such as Adam (Kingma & Ba, 2014),

$$\nabla_{\phi}\mathcal{L}_{\text{ELBO}}(\phi, \tau, \mathcal{D}_N) = \mathbb{E}_{q(\mathbf{x}|\phi)}[(\log \alpha_{PI}(\mathbf{x}, \mathcal{D}_N, \tau) - \log q(\mathbf{x}|\phi) + \log p(\mathbf{x}|\mathcal{D}_0)) \nabla_{\phi} \log q(\mathbf{x}|\phi)].$$
$$(8)$$

Here we use Monte-Carlo sampling to approximate the expectation with a suitable variance reduction scheme, such as control variates (Mohamed et al., 2020). We find that the exponentially smoothed average of the ELBO works well in practice, and is the same strategy employed in Daulton et al. (2022). VSD implements black-box variational inference (Ranganath et al., 2014) for parameter estimation, and despite the high-dimensional nature of $\mathcal{X}$, we find we only need $\mathcal{O}(1000)$ samples to estimate the required expectations for ELBO optimization on problems with $M = \mathcal{O}(100)$, satisfying (R3). Note that Equation 6 – 8 do not involve any data ($\mathcal{D}_N$) directly, only indirectly through the acquisition function. Hence the scalability of VSD is dependent on the complexity of training the underlying estimator of $p(y|\mathbf{x}, \mathcal{D}_N)$.

## 2.3 CLASS PROBABILITY ESTIMATION

So far our method indirectly computes PI by transforming the predictions of a GP surrogate model, $p(y|\mathbf{x}, \mathcal{D}_N)$, as in Equation 5. Instead we may choose to follow the reasoning used by Bayesian optimization by density-ratio estimation (BORE) in Tiao et al. (2021); Oliveira et al. (2022); Song et al. (2022), and directly estimate the quantity we care about, $p(y > \tau|\mathbf{x}, \mathcal{D}_N)$. This can be accomplished using class probability estimation (CPE) on the labels $z := \mathbb{1}[y > \tau] \in \{0, 1\}$ so $p(y > \tau|\mathbf{x}, \mathcal{D}_N) = p(z = 1|\mathbf{x}, \mathcal{D}_N) \approx \pi_\theta(\mathbf{x})$, where $\pi_\theta : \mathcal{X} \to [0, 1]$. We can estimate the class probabilities using a proper scoring rule (Gneiting & Raftery, 2007) such as log-loss,

$$\mathcal{L}_{\text{CPE}}(\theta, \mathcal{D}_N^z) := -\frac{1}{N} \sum_{n=1}^{N} z_n \log \pi_\theta(\mathbf{x}_n) + (1 - z_n) \log(1 - \pi_\theta(\mathbf{x}_n)), \quad (9)$$

where $\mathcal{D}_N^z = \{(z_n, \mathbf{x}_n)\}_{n=1}^{N}$. The VSD objective and gradient estimator using CPE then become,

$$\mathcal{L}_{\text{ELBO}}(\phi, \theta) = \mathbb{E}_{q(\mathbf{x}|\phi)}[\log \pi_\theta(\mathbf{x})] - \mathbb{D}_{\text{KL}}[q(\mathbf{x}|\phi) \| p(\mathbf{x}|\mathcal{D}_0)], \quad (10)$$

$$\nabla_\phi \mathcal{L}_{\text{ELBO}}(\phi, \theta) = \mathbb{E}_{q(\mathbf{x}|\phi)}[(\log \pi_\theta(\mathbf{x}) - \log q(\mathbf{x}|\phi) + \log p(\mathbf{x}|\mathcal{D}_0)) \nabla_\phi \log q(\mathbf{x}|\phi)]. \quad (11)$$

into which we plug $\theta_t^* = \arg\min_\theta \mathcal{L}_{\text{CPE}}(\theta, \mathcal{D}_N^z)$. We refer to this strategy as CPE-PI. Using a CPE enables the use of more scalable estimators than GP-PI, satisfying our desideratum (D3). This is crucial if we choose to run more than a few rounds of experiments with $B = \mathcal{O}(1000)$. Since VSD is a black-box method we may choose to use CPEs that are non-differentiable, such as decision tree ensembles. The complete VSD algorithm is given in Algorithm 1 and depicted in Figure B.1. We have allowed for a threshold function, $\tau_t = f_\tau(\{y : y \in \mathcal{D}_N\}, \gamma_t)$, that can be used to modify the threshold each round. For example, an empirical quantile function $\tau_t = \hat{Q}_y(\gamma_t)$ where $\gamma_t \in (0, 1)$ as in Tiao et al. (2021). Or a constant $\tau$ for estimating a constant distribution of the super level-set.

---

**Algorithm 1** VSD optimization loop with CPE-PI.

---

**Require:** Threshold function $f_\tau$ and $\gamma_1$, dataset $\mathcal{D}_N$, black-box $f_\bullet$, prior $p(\mathbf{x}|\mathcal{D}_0)$, CPE $\pi_\theta(\mathbf{x})$, variational family $q(\mathbf{x}|\phi)$, budget $T$ and $B$.
1: **function** FITMODELS($\mathcal{D}_N, \tau$)
2:     $\mathcal{D}_N^z \leftarrow \{(z_n, \mathbf{x}_n)\}_{n=1}^{N}$, where $z_n = \mathbb{1}[y_n > \tau]$
3:     $\theta^* \leftarrow \arg\min_\theta \mathcal{L}_{\text{CPE}}(\theta, \mathcal{D}_N^z)$
4:     $\phi^* \leftarrow \arg\max_\phi \mathcal{L}_{\text{ELBO}}(\phi, \theta^*)$
5:     **return** $\phi^*, \theta^*$
6: **for** round $t \in \{1, \ldots, T\}$ **do**
7:     $\tau_t \leftarrow f_\tau(\{y : y \in \mathcal{D}_N\}, \gamma_t)$
8:     $\phi_t^*, \theta_t^* \leftarrow$ FITMODELS($\mathcal{D}_N, \tau_t$)
9:     $\{\mathbf{x}_{bt}\}_{b=1}^{B} \leftarrow q(\mathbf{x}|\phi_t^*)$
10:    $\{y_{bt}\}_{b=1}^{B} \leftarrow \{f_\bullet(\mathbf{x}_{bt}) + \epsilon_{bt}\}_{b=1}^{B}$
11:    $\mathcal{D}_N \leftarrow \mathcal{D}_N \cup \{(\mathbf{x}_{bt}, y_{bt})\}_{b=1}^{B}$
12: $\tau_* \leftarrow f_\tau(\{y : y \in \mathcal{D}_N\}, \gamma_*)$
13: $\phi^*, \theta^* \leftarrow$ FITMODELS($\mathcal{D}_N, \tau_*$)
14: **return** $\phi^*, \theta^*$

---

## 2.4 THEORETICAL ANALYSIS

We show that VSD sampling distributions converge to a target distribution that characterizes the level set given by $\tau$, satisfying (D1) in two general settings. We first derive results assuming $f_\bullet$ is drawn

from a Gaussian process, i.e., $f_\bullet \sim \mathcal{GP}(0, k)$, with a positive-semidefinite covariance (or kernel) function $k : \mathcal{X} \times \mathcal{X} \to \mathbb{R}$ (Appendix E), using GP-PI as the CPE for VSD. These results are then extended to probabilistic classifiers based on wide neural networks (NNs) (Appendix F) by means of the neural tangent kernel (NTK) for the given architecture (Jacot et al., 2018). For the analysis, we set $B = 1$ and $N = t$, though having $B > 1$ should improve the rates by a multiplicative factor.

**Theorem 2.1.** *Under mild assumptions (E.1 to E.5), the variational distribution of VSD equipped with GP-PI converges to the level-set distribution in probability at the following rate:*

$$\mathbb{D}_{\mathrm{KL}}[p(\mathbf{x}|y > \tau_t, \mathcal{D}_t) \| p(\mathbf{x}|y > \tau_t, f_\bullet)] \in \mathcal{O}_\mathbb{P}(t^{-1/2}) . \tag{12}$$

This result is based on showing that the GP posterior variance vanishes at an optimal rate of $\mathcal{O}(t^{-1})$ in our setting (Lemma E.5). We also analyze the rate at which VSD finds feasible designs, or "hits", compared to an oracle with full knowledge of $f_\bullet$. After $T$ rounds, the number of hits found by VSD is $H_T = \sum_{t=1}^{T} \mathbb{1}[y_t > \tau_{t-1}]$, where $y_t$ follows Equation 1 and $\mathbf{x}_t \sim p(\mathbf{x}|y > \tau_{t-1}, \mathcal{D}_{t-1})$. The number of hits, $H_T^*$, from an agent that fully knows $f_\bullet$ is the same but for generating conditioned on $f_\bullet$ with $\mathbf{x}_t \sim p(\mathbf{x}|y > \tau_{t-1}, f_\bullet)$. Using this definition and Theorem 2.1, we prove the following.

**Corollary 2.1.** *Under the settings in Theorem 2.1, we also have that:*

$$\mathbb{E}[|H_T - H_T^*|] \in \mathcal{O}(\sqrt{T}) . \tag{13}$$

$\mathbb{E}[H_T]$ is related to the empirical recall measure (18) up to the normalization constant, but it does not account for repeated hits, which are treated as false discoveries (false positives) under recall. Lastly, for NN-based CPEs, we obtain convergence rates dependent on the spectrum of the NTK (Proposition F.2), which we instantiate for infinitely wide ReLU networks below. For the full results and proofs, please see Appendix E for the GP-based analysis and Appendix F for the NTK results.

**Corollary 2.2.** *Let $\pi_\theta$ be modeled via a fully connected ReLU network. Then, under assumptions on identifiability and sampling (F.1 to F.6), in the infinite-width limit, VSD with CPE-PI achieves:*

$$\mathbb{D}_{\mathrm{KL}}[p(\mathbf{x}|y > \tau_t, \mathcal{D}_t) \| p(\mathbf{x}|y > \tau_t, f_\bullet)] \in \widetilde{\mathcal{O}}_\mathbb{P}\left(t^{-\frac{1}{2(M+1)}}\right) . \tag{14}$$

This result finally indicates that, when equipped with flexible NN-based CPEs, VSD is also capable of recovering the target distribution for arbitrary sequence lengths in combinatorial problems.

## 3 RELATED WORK

We will consider related work firstly in terms of methods that have similar components to VSD, then secondly in terms of related problems to our specification of active generation. VSD can be viewed as one of many methods that makes use of the variational bound (Staines & Barber, 2013),

$$\max_{\mathbf{x}} f_\bullet(\mathbf{x}) \geq \max_{\phi} \mathbb{E}_{q(\mathbf{x}|\phi)}[f_\bullet(\mathbf{x})] . \tag{15}$$

The maximum is always greater than or equal to the expected value of a random variable. This bound is useful for black-box optimization (BBO) of $f_\bullet$, and becomes tight if $q(\mathbf{x}|\phi) \to \delta(\mathbf{x}^*)$. See Appendix G for more detail and VSD's relation to BO. Other well known methods that make use of this bound are natural evolution strategies (NES) (Wierstra et al., 2014), variational optimization (VO) (Staines & Barber, 2013; Bird et al., 2018), estimation of distribution algorithms (EDA) (Larrañaga & Lozano, 2001; Brookes et al., 2020), and Bayesian optimization with probabilistic reparametrization (BOPR) (Daulton et al., 2022). For learning the parameters of the variational distribution, $\phi$, they variously make use of maximum likelihood estimation or the score function gradient estimator (REINFORCE) (Williams, 1992). Algorithms that explicitly modify Equation 15 to stop the collapse of $q(\mathbf{x}|\phi)$ to a point mass for batch design include design by adaptive sampling (DbAS) (Brookes & Listgarten, 2018) and conditioning by adaptive sampling (CbAS) (Brookes et al., 2019). They use fixed samples $\mathbf{x}^{(s)}$ from $q(\mathbf{x}|\phi_{t-1}^*)$ for approximating the expectation, and then optimize $\phi$ using a weighted maximum-likelihood or variational style procedure. Though DbAS and CbAS were formulated for offline (non-sequential) tasks, they have often been used in a sequential setting (Ren et al., 2022). We can take a unifying view of algorithms that use a surrogate model for $f_\bullet$ by recognizing the general gradient estimator,

$$\mathbb{E}_{q(\mathbf{x}|\phi')}[w(\mathbf{x})\nabla_\phi \log q(\mathbf{x}|\phi)] . \tag{16}$$

| Method | $w(\mathbf{x})$ | $\phi'$ | Fixed $\mathbf{x}^{(s)} \sim q(\mathbf{x}|\phi')$ per round? |
|--------|-----------------|---------|-------------------------------------------------------------|
| VSD | $\log \pi_{\theta^*}(\mathbf{x}) + \log p(\mathbf{x}|\mathcal{D}_0) - \log q(\mathbf{x}|\phi)$ | $\phi$ | No (REINFORCE) |
| CbAS | $\pi_{\theta^*}(\mathbf{x})p(\mathbf{x}|\mathcal{D}_0)/q(\mathbf{x}|\phi^*_{t-1})$ | $\phi^*_{t-1}$ | Yes (importance Monte Carlo) |
| DbAS | $\pi_{\theta^*}(\mathbf{x})$ | $\phi^*_{t-1}$ | Yes (Monte Carlo) |
| BORE* | $\pi_{\theta^*}(\mathbf{x})$ | $\phi$ | No (REINFORCE) |
| BOPR | $\alpha(\mathbf{x}, \mathcal{D}_N)$ | $\phi$ | No (REINFORCE) |

Table 1: How related methods can be adapted from Equation 16. VSD, CbAS and DbAS may also use a cumulative distribution representation of $\alpha_{\mathrm{PI}}(\mathbf{x}, \mathcal{D}_N, \tau)$ in place of $\pi_{\theta^*}(\mathbf{x})$.

where we give each component in Table 1. For our experiments BORE has been adapted to discrete $\mathcal{X}$ by using the score function gradient estimator, which we denote by BORE*, while CbAS and DbAS have been adapted to use a CPE – their original derivations use a PI acquisition function.

A number of finite horizon methods have been applied to biological sequence BBO tasks, such as Amortized BO (Swersky et al., 2020), GFlowNets (Jain et al., 2022), and the reinforcement learning based DynaPPO (Angermueller et al., 2019). LSO-like methods (Gómez-Bombarelli et al., 2018; Tripp et al., 2020; Stanton et al., 2022; Gruver et al., 2023) tackle optimization of sequences by encoding them into a continuous latent space within which candidate optimization or generation takes place. Selected candidates are decoded back into sequences before black box evaluation; see González-Duque et al. (2024) for a comprehensive survey. VSD does not require a latent space nor an encoder, and as such can be seen as an amortized variant of probabilistic reparameterisation methods (Daulton et al., 2022) or continuous relaxations (Michael et al., 2024). Heuristic stochastic search methods such as AdaLead (Sinai et al., 2020) and proximal exploration (PEX) (Ren et al., 2022) have also demonstrated strong empirical performance on these tasks. We compare the properties of the most relevant methods to our problem in Table 2.

In contrast to just finding the maximum using BBO, active generation considers another problem – generating samples from a rare set of feasible solutions. Generation methods that estimate the super level-set distribution, $p(\mathbf{x}|y > \tau)$, include CbAS, which optimizes the forward KL divergence, $\mathbb{D}_{\mathrm{KL}}[p(\mathbf{x}|y > \tau)\|q(\mathbf{x}|\phi)]$ using importance weighted cross entropy estimation (Rubinstein, 1999). Batch-BORE (Oliveira et al., 2022) also optimizes the reverse KL divergence and uses CPE, but with Stein variational inference (Liu & Wang, 2016) for diverse batch candidates (with a continuous relaxation for discrete variables). There is a rich literature on the related task of active learning and BO for level set estimation (LSE) (Bryan et al., 2005; Gotovos et al., 2013; Bogunovic et al., 2016; Zhang et al., 2023a). However, we focus on learning a generative model of a discrete space.

For active generation VSD, CbAS and DbAS all use an acquisition function defined in the *original* domain, $\mathcal{X}$, to weight gradients (see Equation 16) for learning a conditional generative model, from which $\mathbf{x}_{bt}$ are sampled. An alternative is to use *guided generation*, that is to train an unconditional generative model, and then have a discriminative model guide (condition) the samples from the unconditional model at test time. This plug-and-play of a discriminative model has shown promise for controlled image and text generation of pre-trained models (Nguyen et al., 2017; Dathathri et al., 2020; Li et al., 2022; Zhang et al., 2023b). LaMBO (Stanton et al., 2022) and LaMBO-2 (Gruver et al., 2023) take a guided generation approach to solve the active generation problem. LaMBO uses an (unconditional) masked language model auto-encoder, and then optimizes sampling from its latent space using an acquisition function as a guide. LaMBO-2 takes a similar approach, but uses a diffusion process as the unconditional model, and modifies a Langevin sampling de-noising process with an acquisition function guide.

## 4  EXPERIMENTS

Firstly we test our method, VSD, on its ability to generate complex, structured designs, $\mathbf{x}$, in a single round by training it to generate a subset of handwritten digits from flattened MNIST images (LeCun et al., 1998) in Sec. 4.1. We then compare VSD on two sequence design tasks against existing baseline methods. The first of these tasks (Sec. 4.2) is to generate as many unique, fit sequences as possible using the datasets DHFR (Papkou et al., 2023), TrpB (Johnston et al., 2024) and TFBIND8 (Barrera et al., 2016). These datasets contain near complete evaluations of $\mathcal{X}$, and to our knowledge DHFR and TrpB are novel in the machine learning literature. The second (Sec. 4.3) is a more traditional black-box optimization task of finding the maximum of an unknown function; using

| Method | Rare $\mathbf{x} \in S$ (R1) | Sequential (R2) | Discrete $\mathcal{X}$ (R3) | Batch $\{\mathbf{x}_{bt}\}_{b=1}^{B}$ (R4) | Generative $q(\mathbf{x}|\phi)$ (R5) | Guaranteed (D1) | Gradient descent (D2) | Scalable (D3) | General acq./reward fn. | Amortization |
|---|---|---|---|---|---|---|---|---|---|---|
| BOPR (Daulton et al., 2022) | ✗ | ✓ | ✓ | ✗ | – | ✓ | ✓ | ✗ | ✓ | – |
| BORE (Tiao et al., 2021) | ✗ | ✓ | – | ✗ | – | ✓ | ✓ | ✓ | ✗ | – |
| Batch BORE (Oliveira et al., 2022) | ✓ | ✓ | ✗ | ✓ | ✓ | ✓ | ✓ | ✓ | ✓ | ✓ |
| DbAS (Brookes & Listgarten, 2018) | ✓ | – | ✓ | ✓ | ✓ | ✗ | ✓ | ✓ | ✗ | ✓ |
| CbAS (Brookes et al., 2019) | ✓ | – | ✓ | ✓ | ✓ | ✗ | ✓ | ✓ | ✗ | ✓ |
| Amortized BO (Swersky et al., 2020) | ✗ | ✓ | ✓ | ✓ | ✓ | ✗ | ✓ | ✓ | ✓ | ✓ |
| GFlowNets (Jain et al., 2022) | ✗ | ✓ | ✓ | ✓ | ✓ | ✗ | ✓ | ✓ | ✓ | ✓ |
| DynaPPO (Angermueller et al., 2019) | ✗ | ✓ | ✓ | ✓ | ✓ | ✗ | ✓ | – | ✓ | ✓ |
| AdaLead (Sinai et al., 2020) | ✗ | ✓ | ✓ | ✓ | ✗ | ✗ | ✗ | – | ✗ | ✗ |
| PEX (Ren et al., 2022) | ✗ | ✓ | ✓ | ✓ | ✗ | ✗ | ✗ | – | ✗ | ✗ |
| GGS (Kirjner et al., 2024) | ✗ | ✗ | ✓ | ✓ | ✓ | ✗ | ✗ | ✗ | ✗ | ✗ |
| LSO e.g. (Tripp et al., 2020) | ✗ | ✓ | ✓ | ✗ | ✓ | ✗ | ✓ | – | ✓ | – |
| LaMBO (Stanton et al., 2022) | ✓ | ✓ | ✓ | ✓ | ✓ | ✗ | ✓ | – | ✓ | ✓ |
| LaMBO-2 (Gruver et al., 2023) | ✓ | ✓ | ✓ | ✓ | ✓ | ✗ | ✓ | ✓ | ✓ | ✓ |
| VSD (ours) | ✓ | ✓ | ✓ | ✓ | ✓ | ✓ | ✓ | ✓ | ✗ | ✓ |

Table 2: Feature table of competing methods: ✓ has feature, ✗ does not have feature, – partially has feature, or requires only simple modification. We follow Swersky et al. (2020) in their definition of amortization referring to the ability to use $q(\mathbf{x}|\phi_{t-1}^*)$ for warm-starting the optimization of $\phi_t$.

datasets AAV (Bryant et al., 2021), GFP (Sarkisyan et al., 2016) and the biologically inspired Ehrlich functions (Stanton et al., 2024). The corresponding datasets involve $|\mathcal{V}| \in \{4, 20\}$, $4 \leq M \leq 237$ and $65,000 < |\mathcal{X}| < 20^{237}$. We discuss the settings and properties of these datasets in greater detail in Appendix C. For the biological sequence experiments we run a predetermined number of experimental rounds, $T = 10$ or 32 for the Ehrlich functions. We set the batch size to $B = 128$, and use five different seeds for random initialization. We compare against DbAS (Brookes & Listgarten, 2018), CbAS (Brookes et al., 2019), AdaLead (Sinai et al., 2020), and PEX (Ren et al., 2022) – all of which we have adapted to use a CPE, BORE (Tiao et al., 2021) – which we have adapted to use the score function gradient estimator, and a naïve baseline that uses random samples from the prior, $p(\mathbf{x}|\mathcal{D}_0)$. To reduce confounding, all methods share the same surrogate model, acquisition functions, priors and variational distributions where possible. We compare against LaMBO-2 (Gruver et al., 2023) on the Ehrlich functions, it uses its own surrogate and generative models.

## 4.1 CONDITIONAL GENERATION OF HANDWRITTEN DIGITS

Our motivating application for VSD is to model the space of fit DNA and protein sequences, which are string-representations of complex 3-dimensional structures. In this experiment we aim to demonstrate, by analogy, that VSD can generate sequences that represent 2-dimensional structures. For this task, we have chosen to 'unroll' (reverse the order of every odd row, and flatten) down-scaled ($14 \times 14$ pixel, 8-bit) MNIST (LeCun et al., 1998) images into sequences, $\mathbf{x}$, where $M = 196$ and $|\mathcal{V}| = 8$. We then train long short-term memory (LSTM) recurrent neural network (RNN) and decoder-only causal transformer generative models on the entire MNIST training set by maximum likelihood (ML). These generative distributions are used as the prior models, $p(\mathbf{x}|\mathcal{D}_0)$, for VSD and we detail their form in Appendix C.3. The task is then to use VSD in one round to estimate the pos-

(a) LSTM Prior     (b) Transformer Prior     (c) LSTM Posterior     (d) Transformer Posterior

Figure 2: (a) and (b) are samples from the LSTM and transformer priors, respectively. (c) and (d) show samples from the LSTM and transformer VSD variational distributions respectively. We also report the samples mean scores according to the CPE probabilities.

terior $p(\mathbf{x}|y \in \{3,5\})$ using a CPE trained on labels $z_n = \mathbb{1}[y_n \in \{3,5\}]$. We use a convolutional architecture for the CPE given in Appendix C.4, and it achieves a test balanced accuracy score of $\sim 99\%$. We parameterize the variational distributions, $q(\mathbf{x}|\phi)$, in the same way as the priors, and initialize these distribution parameters from the prior distribution parameters. During training with ELBO the prior distribution parameters are locked, and we run training for 5000 iterations. This is exactly lines 8 and 9 in Algorithm 1. Samples are visualized from the resulting variational distributions with the corresponding priors in Figure 2. We see that the prior LSTM and transformer are able to generate convincing digits once the sampled sequences are 're-rolled', and that VSD is able to effectively refine these distributions, even though it does not have access to any data directly – only scores from the CPE. Both the LSTM and transformer yield qualitatively similar results, and have similar mean scores from the CPE.

## 4.2 FITNESS LANDSCAPES

In this setting we wish to find fit sequences $\mathbf{x} \in \mathcal{S}_{\text{SLS}}$, so we fix $\tau$ over all rounds. We only consider the combinatorially (near) complete datasets to avoid any pathological behavior from relying on machine learning oracles (Surana et al., 2024). Results are presented in Figure 3. The primary measures by which we compare methods are precision, recall and performance,

$$\text{Precision}_t = \frac{1}{\min\{tB, |\mathcal{S}|\}} \sum_{r=1}^{t} \sum_{b=1}^{B} \mathbb{1}[y_{br} > \tau] \cdot \mathbb{1}[\mathbf{x}_{br} \notin \mathcal{X}_{b-1,r}^q], \tag{17}$$

$$\text{Recall}_t = \frac{1}{\min\{TB, |\mathcal{S}|\}} \sum_{r=1}^{t} \sum_{b=1}^{B} \mathbb{1}[y_{br} > \tau] \cdot \mathbb{1}[\mathbf{x}_{br} \notin \mathcal{X}_{b-1,r}^q], \tag{18}$$

$$\text{Performance}_t = \sum_{r=1}^{t} \sum_{b=1}^{B} y_{br} \cdot \mathbb{1}[\mathbf{x}_{br} \notin \mathcal{X}_{b-1,r}^q]. \tag{19}$$

Here $\mathcal{X}_{br}^q \subset \mathcal{X}$ is the set of experimentally queried sequences by the $b$th batch member of the $r$th round, including the initial training set. These measures are comparable among probabilistic and non probabilistic methods. Precision and recall measure the ability of a method to efficiently explore $\mathcal{S}$, where $\min\{tB, |\mathcal{S}|\}$ is the size of the selected set at round $t$ (bounded by the number of good solutions), and $\min\{TB, |\mathcal{S}|\}$ is the number of positive elements possible in the experimental budget. Performance measures the cumulative fitness of the unique batch members, but unlike Jain et al. (2022) we do not normalize this measure.

For exact experimental settings we refer the reader to Appendix C.1. We set $\tau$ to be that of the wild-type sequences in the DHFR and TrpB datasets, and use $\tau = 0.75$ for TFBIND8. We find that a uniform prior over sequences, and a mean field variational distribution (Equation 22) are adequate for these experiments, as is a simple MLP for the CPE. Results are presented in Figure 3. VSD is the best performing method by most of the measures. We have found the AdaLead and PEX evolutionary-search based methods to be effective on lower-dimensional problems (TFBIND8 being the lowest here), however we consistently observe their performance degrading as the dimension of the problem increases. We suspect this is a direct consequence of their random mutation strategies being suited to exploration in low dimensions, but less efficient in higher dimensions compared to the learned generative models employed by VSD, CbAS, and DbAS. Our modified version of BORE (which is just the expected log-likelihood component of Equation 10) performs badly in all cases, and this is a direct consequence of its variational distribution collapsing to a point mass. In a non-batch setting this behavior is not problematic, but shows the importance of the KL divergence of VSD in this batch setting. We replicate these experiments in Appendix D.1 using GP-PI, also backed by our guarantees. In all cases VSD's results remain similar or improve slightly, whereas the other methods results remain similar or degrade. We report on batch diversity scores in Appendix D.3.

## 4.3 BLACK-BOX OPTIMIZATION

In this experiment we use VSD on the related task of BBO for finding $\mathcal{S}_{\text{BBO}}$. We set $\tau_t$ adaptively by specifying it as an empirical quantile $\tilde{Q}_y^t$ of the observed target values at round $t$,

$$\tau_t = \tilde{Q}_y^t(\gamma_t = p_{t-1}^\eta) \tag{20}$$

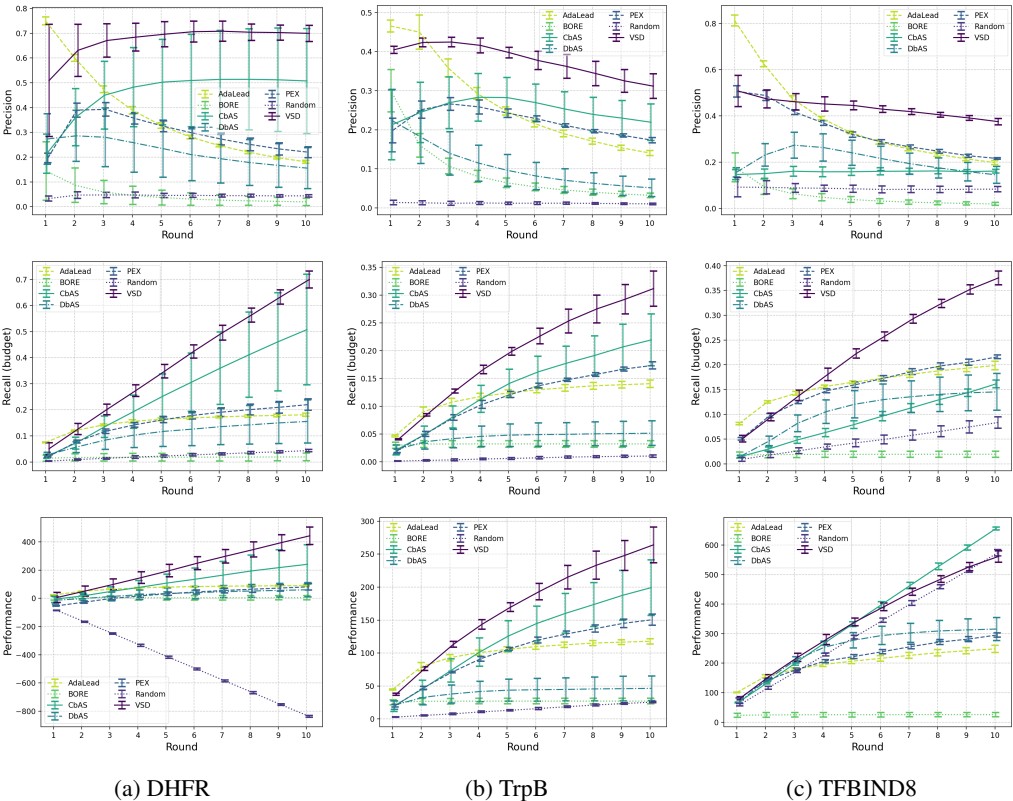

| (a) DHFR | (b) TrpB | (c) TFBIND8 |

Figure 3: Fitness landscape results. Precision (Equation 17), recall (Equation 18) and performance (Equation 19) – higher is better – for the combinatorially (near) complete datasets, DHFR, TrpB and TFBIND8. The random method is implemented by drawing $B$ samples uniformly.

where $p_{t-1}$ is a percentile from the previous round, and $\eta \in [0,1]$ is an annealing parameter for $\tau_t$ that trades off exploration and exploitation. Performance is measured by simple regret $r_t$, which quantifies the fitness gap between the globally optimal design and the best design found,

$$r_t = y^* - \max_y \{y_{bi}\}_{b=1,i=1}^{B,t}. \tag{21}$$

Here $y^*$ is the fitness value of the globally optimal sequence $\mathbf{x}^*$. We use the higher dimensional AAV ($y^*$=19.54), GFP ($y^*$=4.12) and Ehrlich functions ($y^*$=1) datasets/benchmarks to show that VSD can scale to higher dimensional problems. $\mathcal{X}$ of AAV and GFP is completely intractable to fully explore experimentally, and so we use a predictive oracle trained on all the original experimental data as the ground-truth black-box function. We use the CNN-based oracles from Kirjner et al. (2024) for these experiments. However, we note here that some oracles used in these experiments do not predict well out-of-distribution (Surana et al., 2024), which limits their real-world applicability. The Ehrlich functions (Stanton et al., 2024) are challenging biologically inspired closed-form simulations that cover all $\mathcal{X}$. We compare against a genetic algorithm (GA), CbAS and LaMBO-2 (Gruver et al., 2023) for sequences of length $M = \{15, 32, 64\}$ using the POLI and POLI-BASELINES benchmarks and baselines software (González-Duque et al., 2024). For these experiments we use CNNs for the CPEs – all experimental settings are in Appendix C.2.

The results are summarized in Figure 4 and 5. Batch diversity scores for these experiments are presented in Appendix D.3, and for HOLO Ehrlich function implementation results see Appendix D.2. VSD is among the leading methods for all experiments. VSD takes better advantage of the more complex variational distributions than CbAS and DbAS since it can sample from the adapted variational distribution while learning it. We can see that AdaLead, PEX and often BORE all perform worse than random for reasons previously mentioned. Simple regret can drop below zero for AAV & GFP since an oracle is used as the black box function, but the global maximizer is taken from the experimental data. VSD outperforms CbAS on the Ehrlich function benchmarks, and is competitive with LaMBO-2. We also present an ablation study in Appendix D.4.

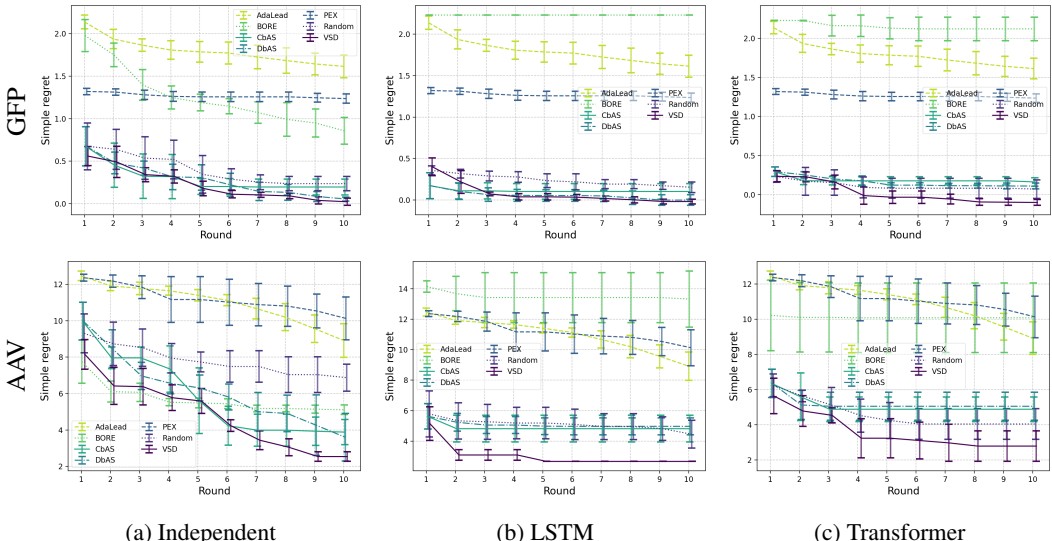

Figure 4: AAV & GFP BBO results. Simple regret (Equation 21) – lower is better – on GFP and AAV with independent and auto-regressive variational distributions. The PEX and AdaLead results are replicated between the plots, since they are unaffected by choice of variational distribution.

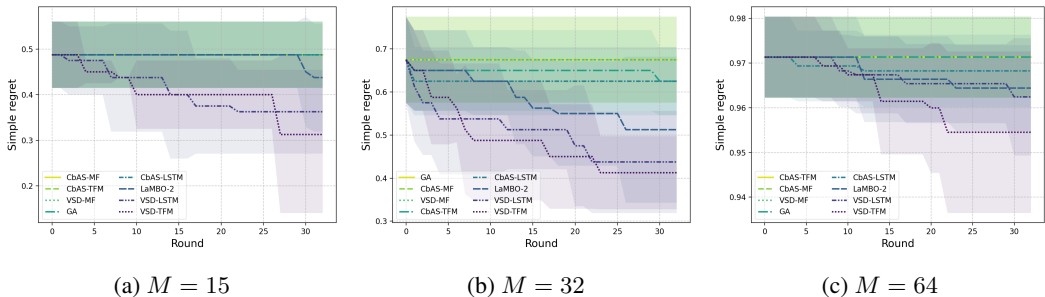

Figure 5: Ehrlich function (POLI implementation) BBO results. VSD and CbAS with different variational distributions; mean field (MF), LSTM and transformer (TFM), compared against genetic algorithm (GA) and LaMBO-2 baselines.

## 5 CONCLUSION

We have presented the problem of active generation — sequentially learning a generative model for designs of a rare class by efficiently evaluating a black-box function — and a method for efficiently generating samples which we call variational search distributions (VSD). Underpinned by variational inference, VSD satisfies critical requirements and important desiderata, and we show that VSD converges asymptotically to the true level-set distribution at the same rate as a Monte-Carlo estimator with full knowledge of the true distribution. We showcased the benefits of our method empirically on a set of combinatorially complete and high dimensional sequential-design biological problems and show that it can effectively learn powerful generative models of fit designs. There is a close connection between active generation and black-box optimization, and with the advent of powerful generative models we hope that our explicit framing of generation of fit sequences will lead to further study of this connection. Finally, our framework can be generalized to more complex application scenarios, involving learning generative models over Pareto sets, $\mathcal{S}_{\text{Pareto}}$, in a multi-objective setting, or other challenging combinatorial optimization problems (Bengio et al., 2021), such as graph structures (Annadani et al., 2023), and mixed discrete-continuous variables. All of which are worth investigating as future work directions. For the code implementing the models and experiments in this paper, please see https://github.com/csiro-funml/variationalsearch.

ACKNOWLEDGMENTS

This work is funded by the CSIRO Science Digital and Advanced Engineering Biology Future Science Platforms, and was supported by resources and expertise provided by CSIRO IMT Scientific Computing. We would like to thank the anonymous reviewers, and especially reviewer xamp for the constructive feedback and advice, which greatly increased the relevance and quality of this work.

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

# A  ACRONYMS

## ACRONYMS

**BBO**  black-box optimization. 1, 2, 5, 6, 8, 10, 16, 19–21, 23

**BO**  Bayesian optimization. 3, 5, 6, 35

**BOPR**  Bayesian optimization with probabilistic reparametrization. 5–7, 18

**BORE**  Bayesian optimization by density-ratio estimation. 4, 6–9, 16–18, 20, 22

**CbAS**  conditioning by adaptive sampling. 5–10, 16–18, 20, 22

**CPE**  class probability estimation. 4–9, 16–19, 21, 22, 35

**DbAS**  design by adaptive sampling. 5–9, 16–18, 22

**EDA**  estimation of distribution algorithms. 5

**ELBO**  evidence lower bound. 3, 4, 8

**ES**  evolution strategies. 5

**GA**  genetic algorithm. 9

**GP**  Gaussian process. 3–5, 8, 18, 20, 29, 31

**KL**  Kullback-Leibler. 3, 6, 8

**LSE**  level set estimation. 6

**LSO**  latent space optimization. 1, 6, 7

**LSTM**  long short-term memory. 7, 8, 10, 17, 18, 20–23

**ML**  maximum likelihood. 7, 21, 22

**NES**  natural evolution strategies. 5

**NTK**  neural tangent kernel. 5

**PEX**  proximal exploration. 6–10, 17, 18, 21

**PI**  probability of improvement. 3–6, 8, 18, 20, 29, 35

**RNN**  recurrent neural network. 7, 18

**UCB**  upper confidence bound. 9

**VI**  variational inference. 1

**VO**  variational optimization. 5

**VSD**  variational search distributions. 1–10, 16–18, 20–23, 29, 31, 35, 36

## B DEPICTION OF ACTIVE GENERATION

See Figure B.1 for graphical depictions of active generation as implemented by VSD, compared to batch BO. Active generation using VSD follows Algorithm 1 to sequentially approximate $p(\mathbf{x}|y > \tau)$. It uses samples from the current learned approximation of this distribution, $q(\mathbf{x}|\phi_t^*)$, for proposing candidates to evaluate each round. Unmodified batch BO in the discrete setting without any specialization requires a list of candidates, from which a batch of candidates are selected per round using a surrogate model with a batch acquisition function, e.g. see Wilson et al. (2017). The surrogate model's hyper-parameters, $\theta$, are estimated by minimizing negative log marginal likelihood, $\mathcal{L}_{\text{NLML}}(\theta, \mathcal{D}_N)$. Mechanistically, active generation learns a generative model of valuable candidates to circumvent the requirement of having to select candidates from a list. This is especially important for searching over the space of sequences, $\mathcal{X}$, where enumerating feasible candidates is often intractable. Furthermore, active generation naturally lends itself to large batch sizes, while explicit batch optimization is often computationally intensive and is limited to $\mathcal{O}(10)$ candidates per batch Wilson et al. (2017). As mentioned in section 3, alternatives to active generation for specializing batch BO to the discrete domain also include LSO (Tripp et al., 2020) and amortized BO (Swersky et al., 2020) among others.

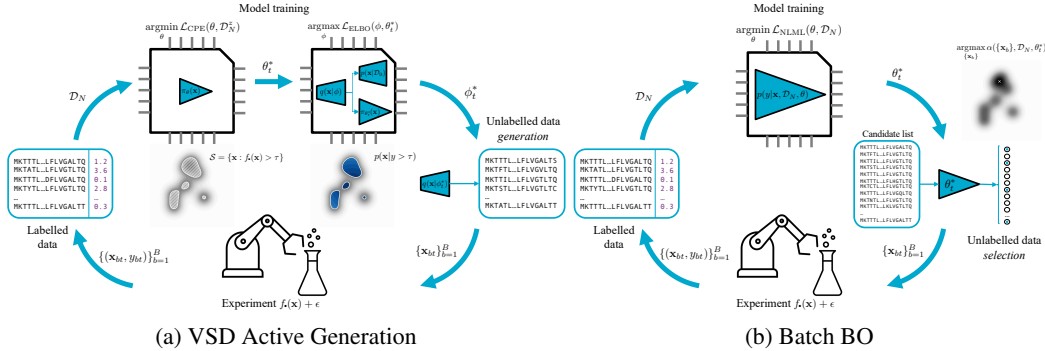

(a) VSD Active Generation          (b) Batch BO

Figure B.1: Depictions of (a) active generation as implemented by VSD, and (b) batch Bayesian optimization as applied to discrete sequences. Please see the text for a discussion of the differences between these approaches.

## C EXPERIMENTAL DETAILS

We use three well established datasets; a green fluorescent protein (GFP) from Aequorea Victoria (Sarkisyan et al., 2016), an adeno-associated virus (AAV) Bryant et al. (2021); and DNA binding activity to a human transcription factor (TFBIND8) (Trabucco et al., 2022; Barrera et al., 2016). These datasets have been used variously by Brookes & Listgarten (2018); Brookes et al. (2019); Angermueller et al. (2019); Kirjner et al. (2024); Jain et al. (2022) among others. The GFP task is to maximize fluorescence, this protein consists of 238 amino acids, of which 237 can mutate. The AAV task us to maximize the genetic payload that can be delivered, and the associated protein has 28 amino acids, all of which can mutate. A complete combinatorial assessment is infeasible for these tasks, and so we use the convolution neural network oracle presented in Kirjner et al. (2024) as *in-silico* ground truth. TFBIND8 contains a complete combinatorial assessment of the effect of changing 8 nucleotides on binding to human transcription factor SIX6 REF R1 (Barrera et al., 2016). The dataset we use contains all 65536 sequences prepared by Trabucco et al. (2022).

We also use two novel datasets from recent works that experimentally assess the (near) complete combinatorial space of short sequences. The first dataset measures the antibiotic resistance of Escherichia coli metabolic gene folA, which encodes dihydrofolate reductase (DHFR) (Papkou et al., 2023). Only a sub-sequence of this gene is varied (9 nucleic acids which encode 3 amino acids), and so a near-complete (99.7%) combinatorial scan is available. For variants that have no fitness (resistance) data available, we give a score of $-1$. The next dataset is near-complete combinatorial scan of four interacting amino acid residues near the active site of the enzyme tryptophan synthase (TrpB) (Johnston et al., 2024), with 159,129 unique sequences and fitness values, we use $-0.2$ for

the missing fitness values (we do not use the authors' imputed values). These residues are explicitly shown to exhibit epistasis – or non-additive effects on catalytic function – which makes navigating this landscape a more interesting challenge from an optimization perspective.

Finally, we use the recently proposed Ehrlich functions (Stanton et al., 2024) benchmark. These functions are challenging closed form biological analogues, specifically designed to test BBO methods on high dimensional sequence design tasks without having to resort to physical experimentation or machine learning oracles. We use the POLI and POLI-BASELINES software package for the benchmark and baselines (González-Duque et al., 2024), and test on both the original HOLO implementation (Stanton et al., 2024) as well as the native POLI implementation of these functions.

The properties of these datasets and benchmarks are presented in Table C.1.

| Dataset | $|\mathcal{V}|$ | $M$ | $|\mathcal{X}_{\text{available}}|$ | $|\mathcal{X}|$ |
|---|---|---|---|---|
| TFBIND8 | 4 | 8 | 65,536 | 65,536 |
| TrpB | 20 | 4 | 159,129 | 160,000 |
| DHFR | 4 | 9 | 261,333 | 262,144 |
| AAV | 20 | 28 | 42,340 | $20^{28}$ |
| GFP | 20 | 237 | 51,715 | $20^{237}$ |
| Ehrlich-15 | 20 | 15 | $20^{15}$ | $20^{15}$ |
| Ehrlich-32 | 20 | 32 | $20^{32}$ | $20^{32}$ |
| Ehrlich-64 | 20 | 64 | $20^{64}$ | $20^{64}$ |

Table C.1: Alphabet size, sequence length, and number of available sequences for each of the datasets we use in this work.

We optimize VSD, CbAS, DbAS and BORE for a minimum of 3000 iterations each round (5000 for all experiments but the Ehrlich functions) using Adam (Kingma & Ba, 2014). When we use a CPE, AdaLead's $\kappa$ parameter is set to 0.5 since the CPE already incorporates the appropriate threshold.

## C.1 FITNESS LANDSCAPES SETTINGS

For the DHFR and TrpB experiments we set maximum fitness in the training dataset to be that of the wild type, and $\tau$ to be slightly below the wild type fitness value (so we have $\sim 10$ positive examples to train the CPE with). We use a randomly selected $N_{\text{train}} = 2000$ below the wild-type fitness to initially train the CPE, we also explicitly include the wild-type. The thresholds and wild-type fitness values are; DHRF: $\tau = -0.1$, $y_{\text{wt}} = 0$, TrpB: $\tau = 0.35$, $y_{\text{wt}} = 0.409$. We follow the same procedure for the TFBIND8 experiment, however, there is no notion of a wild-type sequence in this data, and so we set $\tau = 0.75$, and $y_{\text{train max}} = 0.85$. We use a uniform prior over sequences, $p(\mathbf{x}) = \prod_{m=1}^{M} \text{Categ}(x_m | \mathbf{1} \cdot |\mathcal{V}|^{-1})$, since these are relatively small search spaces, and the subsequences of nucleic/amino acids have been specifically selected for their task. Similarly, we find that relatively simple independent (mean-field) variational distributions of the form in Equation 22 and MLP based CPEs work best for these experiments (details in Sec. C.4).

## C.2 BLACK-BOX OPTIMIZATION SETTINGS

We follow Kirjner et al. (2024) in the experimental settings for the AAV and GFP datasets, but we modify the maximum fitness training point and training dataset sizes to make them more amenable to a sequential optimization setting. The initial percentiles, schedule, and max training fitness values are; AAV: $p_0 = 0.8$, $\eta = 0.7$, $y_{\text{max}} = 5$, GFP: $p_0 = 0.8$, $\eta = 0.7$ $y_{\text{max}} = 1.9$. We aim for $p_T = 0.99$. The edit distance between $\mathbf{x}^*$ and the fittest sequence in the CPE training data is 8 for GFP, and 13 for AAV. We again use a random $N_{\text{train}} = 2000$ for training the CPEs, which in this case are CNNs – architecture specifics are in Sec. C.4.

For the Ehrlich function experiment, we use sequence lengths of $M = \{15, 32, 64\}$ with 2 motifs for the shorter sequence lengths, and 8 motifs for $M = 64$. All use a motif length of 4 and a quantization of 4. $B = 128$, $T = 32$ and *only* 128 random samples of the function are used for $\mathcal{D}_N$ – these are resampled for each seed. As before, 5 different random seeds are used for these trials, and for VSD we use an the same scheduling function for $\tau_t$ as in Equation 20, with $p_0 = 0.5$ and $\eta = 0.87$ (so $p_T = 0.99$). The lower initial percentile is used since the training dataset is much

smaller than in the other experiments, and we find allowing for more exploration initially improves VSD's performance.

In these higher dimensional settings, we find that performance of the methods heavily relies on using an informed prior (in the case of VSD and CbAS), or initial variational distribution (in the case of DbAS and BORE). To this end, we follow Brookes et al. (2019) and fit the initial variational distribution to the CPE training sequences (regardless of fitness), but we use maximum likelihood. For the more complex variational distributions (LSTM and transformer), we have to be careful not to over-fit – so we implement early stopping and data augmentation techniques. Then for VSD and CbAS we copy this distribution and fix its parameters for the remainder of the experiment for use as a prior. We also use this prior for the Random method, but AdaLead and PEX use alternative generative heuristics. For these experiments we use the simple independent variational distribution and the same LSTM and causal decoder-only transformer models from Sec. 4.1.

### C.3 VARIATIONAL DISTRIBUTIONS

In this section we summarize the main variational distribution architectures considered for VSD, BORE, CbAS and DbAS, and the sampling distributions for the Random baseline method. Somewhat surprisingly, we find that we often obtain good results for the biological sequence experiments using a simple independent (or mean-field) variational distribution, especially in lower dimensional settings,

$$q(\mathbf{x}|\phi) = \prod_{m=1}^{M} \mathrm{Categ}(x_m|\mathrm{softmax}(\phi_m)), \tag{22}$$

where $x_m \in \mathcal{V}$ and $\phi_m \in \mathbb{R}^{|\mathcal{V}|}$. However, this simple mean-field distribution was not capable of generating convincing handwritten digits or, in some cases, higher-dimensional sequences. We have also tested a variety of transition variational distributions,

$$q(\mathbf{x}_t|\mathbf{x}_{t-1}, \phi) = \prod_{m=1}^{M} \mathrm{Categ}(x_{tm}|\mathrm{softmax}(\mathrm{NN}_m(\mathbf{x}_{t-1}, \phi))), \tag{23}$$

where $\mathrm{NN}_m(\mathbf{x}_{t-1}, \phi)$ is the $m^{\mathrm{th}}$ vector output of a neural network that takes a sequence from the previous round, $\mathbf{x}_{t-1}$, as input. We have implemented multiple neural net encoder/decoder architectures for $\mathrm{NN}_m(\mathbf{x}_{t-1}, \phi)$, but we did not consider architectures of the form $\mathrm{NN}_m(\phi)$ since the variational distribution in Equation 22 can always learn a $\phi_m = \mathrm{NN}_m(\phi')$. We found that none of these transition architectures significantly outperformed the mean-field distribution (Equation 22) when it was initialized well (e.g. fit to the CPE training sequences), see Sec. D.4 for results. We also implemented auto-regressive variational distributions of the form,

$$q(\mathbf{x}|\phi) = \mathrm{Categ}(x_1|\mathrm{softmax}(\phi_1)) \prod_{m=2}^{M} q(x_m|x_{1:m-1}, \phi_{1:m}) \quad \text{where,} \tag{24}$$

$$q(x_m|x_{1:m-1}, \phi_{1:m}) = \begin{cases} \mathrm{Categ}(x_m|\mathrm{softmax}(\mathrm{LSTM}(x_{m-1}, \phi_{m-1:m}))), \\ \mathrm{Categ}(x_m|\mathrm{softmax}(\mathrm{DTransformer}(x_{1:m-1}, \phi_{1:m}))). \end{cases}$$

For a LSTM RNN and a decoder-only transformer with a causal mask, for the latter see Phuong & Hutter (2022, Algorithm 10 & Algorithm 14) for maximum likelihood training and sampling implementation details respectively. We list the configurations of the LSTM and transformer variational distributions in Table C.2. We use additive positional encoding for all of these models. When using these models for priors or initialization of variational distributions, we find that over-fitting can be an issue. To circumvent this, we use early stopping for larger training datasets, or data augmentation techniques for smaller training datasets (as in the case of the Ehrlich functions).

### C.4 CLASS PROBABILITY ESTIMATOR ARCHITECTURES

For the fitness landscape experiments on the smaller combinatorially complete datasets we use a two-hidden layer MLP, with an input embedding layer. The architecture is given in Figure C.2 (a). For the larger dimensional AAV and GFP datasets and Ehrlich function benchmark, we use the

|  | Configuration | Digits | AAV | GFP | Ehrlich 15 | Ehrlich 32 | Ehrlich 64 |
|---|---|---|---|---|---|---|---|
| LSTM | Layers | 5 | 4 | 4 | 3 | 3 | 3 |
|  | Network size | 128 | 32 | 32 | 32 | 32 | 64 |
|  | Embedding size | 4 | 10 | 10 | 10 | 10 | 10 |
| Transformer | Layers | 4 | 1 | 1 | 2 | 2 | 2 |
|  | Network Size | 256 | 64 | 64 | 32 | 64 | 128 |
|  | Attention heads | 8 | 2 | 2 | 1 | 2 | 3 |
|  | Embedding size | 32 | 20 | 20 | 10 | 20 | 30 |

Table C.2: LSTM and transformer network configuration.

convolutional architecture given in Figure C.2 (b). On all but the Ehrlich benchmark, five fold cross validation was used to select the hyper parameters before the CPEs are trained on the whole training set for use in the subsequent experimental rounds. For the Ehrlich benchmark we do not use cross-validation to select the CPE hyper parameters – but we do use an additive ensemble of 10 randomly initialized CNNs for the CPE following LaMBO-2. Model updates are performed by retraining on the whole query set.

# D  ADDITIONAL EXPERIMENTAL RESULTS

## D.1  FITNESS LANDSCAPES – GAUSSIAN PROCESS PROBABILITY OF IMPROVEMENT

Here we present additional fitness landscape experimental results, where we have used a GP as a surrogate model for $p(y|\mathbf{x}, \mathcal{D}_N)$ in conjunction with a complementary Normal CDF as the PI acquisition function. This is one of the main frameworks supported by our theoretical analysis. VSD, DbAS, CbAS and BORE can make use of the GP-PI acquisition function, and so BORE is BOPR (Daulton et al., 2022) in this instance since we are not using a CPE. PEX and AdaLead only use the GP surrogate, as per their original formulation. The GP uses a simple categorical kernel with automatic relevance determination from Balandat et al. (2020),

$$k(\mathbf{x}, \mathbf{x}') = \sigma \exp\left( -\frac{1}{M} \sum_{m=1}^{M} \frac{\mathbb{1}[x_m = x'_m]}{l_m} \right), \tag{25}$$

where $\sigma$ and $l_m$ are hyper-parameters controlling scale and length-scale respectively. See Figure D.3 for the results.

## D.2  EHRLICH FUNCTION HOLO RESULTS

See Figure D.4 for BBO results on the original HOLO Ehrlich function implementation (Stanton et al., 2024). We present additional diversity scores for these and the POLI implementation in Sec. D.3.

## D.3  DIVERSITY SCORES

The diversity of batches of candidates is a common thing to report in the literature, and to that end we present the diversity of our results here. We have taken the definition of pair-wise diversity from (Jain et al., 2022) as,

$$\text{Diversity}_t = \frac{1}{B(B-1)} \sum_{\mathbf{x}_i \in \mathcal{D}_{Bt}} \sum_{\mathbf{x}_j \in \mathcal{D}_{Bt} \setminus \{\mathbf{x}_i\}} \text{Lev}(\mathbf{x}_i, \mathbf{x}_j), \tag{26}$$

where $\text{Lev} : \mathcal{X} \times \mathcal{X} \to \mathbb{N}_0$ is the Levenshtein distance. We caution the reader as to the interpretation of these results however, as more diverse batches often do not lead to better performance, precision, recall or simple regret (as can be seen from the Random method results). Though insufficient diversity can also explain poor performance, as in the case of BORE. Results for the fitness landscape experiment are presented in Figure D.5, and black-box optimization for AAV & GFP in Figure D.6 and Ehrlich functions in Figure D.7.

```
Sequential(
    Embedding(
        num_embeddings=A,
        embedding_dim=8
    ),
    Dropout(p=0.2),
    Flatten(),
    LeakyReLU(),
    Linear(
        in_features=8 * M,
        out_features=32
    ),
    LeakyReLU(),
    Linear(
        in_features=32,
        out_features=1
    ),
)
```

```
Sequential(
    Embedding(
        num_embeddings=A,
        embedding_dim=10
    ),
    Dropout(p=0.2),
    Conv1d(
        in_channels=10,
        out_channels=16,
        kernel_size=3 or 7,
    ),
    LeakyReLU(),
    MaxPool1d(
        kernel_size=2 or 4,
        stride=2 or 4,
    ),
    Conv1d(
        in_channels=16,
        out_channels=16,
        kernel_size=7,
    ),
    LeakyReLU(),
    MaxPool1d(
        kernel_size=2 or 4,
        stride=2 or 4,
    ),
    Flatten(),
    LazyLinear(
        out_features=128
    ),
    LeakyReLU(),
    Linear(
        in_features=128,
        out_features=1
    ),
)
```

(a) MLP architecture      (b) CNN architecture

Figure C.2: CPE architectures used for the experiments in PyTorch syntax. $A = |\mathcal{V}|$, $M = M$, GFP uses a max pooling kernel size and stride of 4, all other datasets and benchmarks use 2. The Ehrlich function benchmark uses and ensemble of 10 randomly initialized CNNs that are additively combined. The Ehrlich-15 functions use a kernel size of 3, all other BBO experiments use a kernel size of 7. LaMBO-2 uses the same kernel size as our CNNs for the Ehrlich functions.

## D.4 ABLATIONS – VARIATIONAL AND PRIOR DISTRIBUTIONS

In Figure D.8 we present ablation results for VSD using different priors and variational distributions. We use the BBO experimental datasets for this task as they are higher-dimensional and so more sensitive to these design choices. We test the following prior and variational posterior distributions:

**IU** Independent categorical variational posterior distribution of the form in Equation 22, and a uniform prior distribution, $p(\mathbf{x}) = \prod_{m=1}^{M} \text{Categ}(x_m | \mathbf{1} \cdot |\mathcal{V}|^{-1})$.

**I** Independent categorical prior and variational posterior of the form in Equation 22. The prior is fit using ML on the initial CPE training data.

**LSTM** LSTM prior and variational posterior of the form Equation 24. The prior is fit using ML on the initial CPE training data.

**DTFM** Decoder-only causal transformer prior and variational posterior of the form Equation 24. The prior is fit using ML on the initial CPE training data.

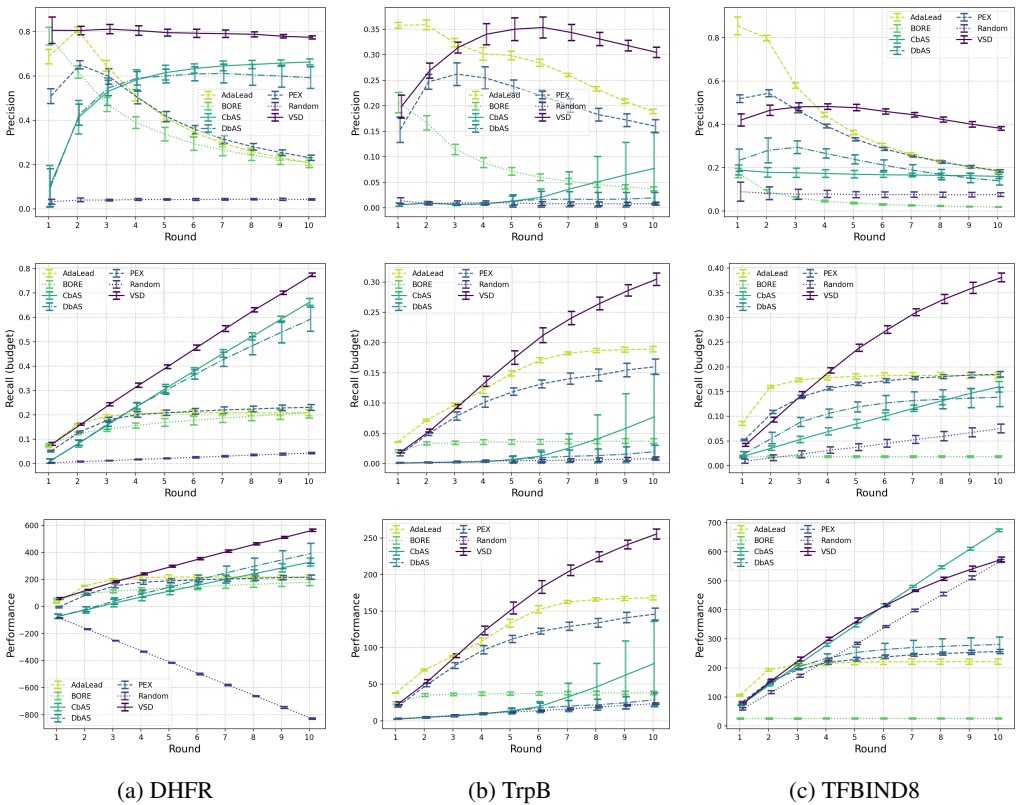

Figure D.3: Fitness landscape results using GP-PI. Precision (Equation 17), recall (Equation 18) and performance (Equation 19) – higher is better – for the combinatorially (near) complete datasets, DHFR and TrpB and TFBIND8. The random method is implemented by drawing $B$ samples uniformly.

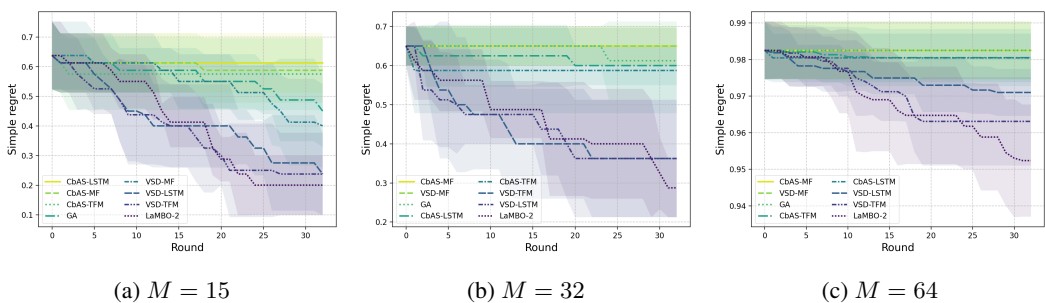

Figure D.4: Ehrlich function (HOLO implementation) BBO results. VSD and CbAS with different variational distributions; mean field (MF), LSTM and transformer (TFM), compared against genetic algorithm (GA) and LaMBO-2 baselines.

**TAE**   Independent categorical prior and a transition-style auto-encoder variational posterior of the form Equation 23, where we use two-hidden layer MLPs for the encoder and decoder. The prior is fit using ML on the initial CPE training data.

**TCNN**  Independent categorical prior and a transition-style convolutional auto-encoder variational posterior of the form Equation 23, where we use a convolutional encoder, and transpose convolutional decoder. The prior is fit using ML on the initial CPE training data.

We use the informed-independent priors with the transition variational distributions since they are somewhat counter-intuitive to use as priors themselves.

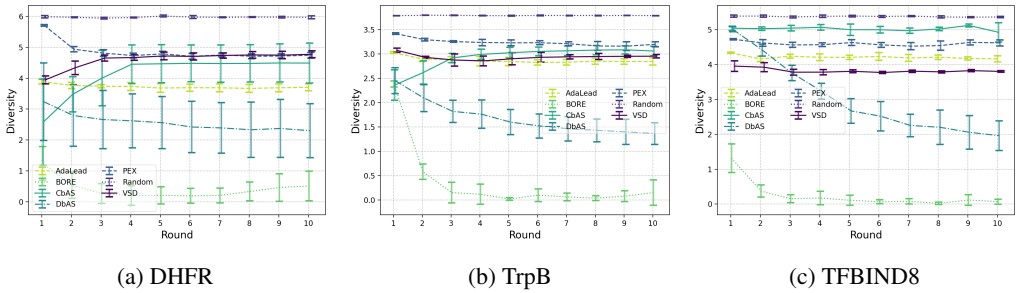

(a) DHFR        (b) TrpB        (c) TFBIND8

Figure D.5: Fitness landscape diversity results. Higher is more diverse, as defined by Equation 26.

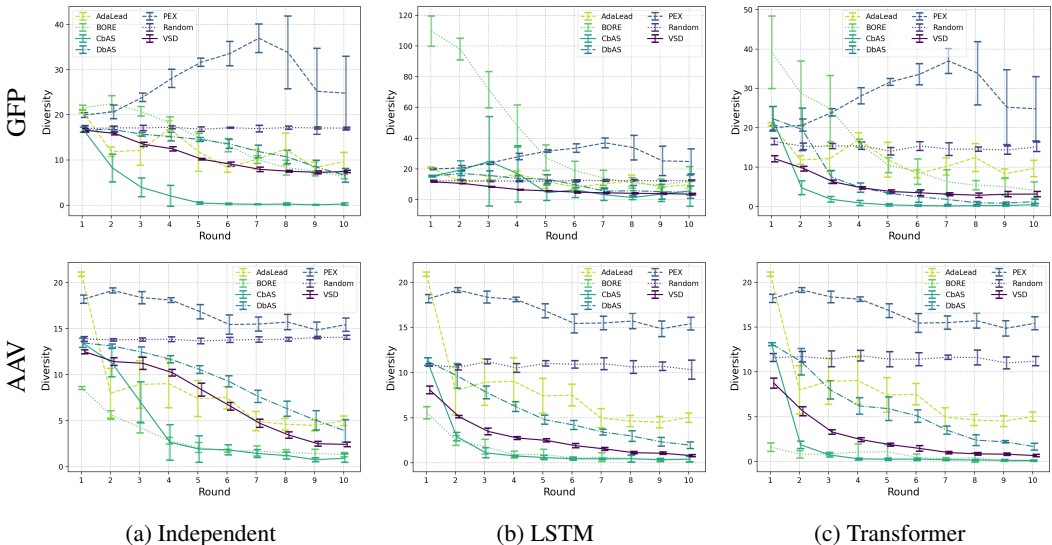

(a) Independent        (b) LSTM        (c) Transformer

Figure D.6: Black-box optimization results for diversity on GFP and AAV with independent and auto-regressive variational distributions. Higher is more diverse, as defined by Equation 26. The PEX and AdaLead results are replicated between the plots, since they are unaffected by choice of variational distribution.

From Figure D.8 we can see that while using an uninformative prior works in the lower-dimensional fitness landscape experiments, using an informative prior is crucial for these higher dimensional problems. We found a similar result when using this uninformative prior with CbAS, or using a uniform initialization with DbAS and BORE. The methods are not able to make any significant progress within the experimental budget given. The independent and transition variational distributions achieve similar performance, whereas the auto-regressive models generally outperform all others. This is because of the LSTM and transformer's superior generalization performance when generating sequences – measured both when training the priors (on held-out sequences) and during VSD adaptation.

# E   THEORETICAL ANALYSIS FOR GP-BASED CPES

In this section, we present theoretical results concerning VSD and its estimates when equipped with Gaussian process regression models (Rasmussen & Williams, 2006). We show that VSD sampling distributions converge to a target distribution that characterizes the level set given by $\tau$. The approximation error mainly depends on the predictive uncertainty of the probabilistic model with respect to the true underlying function $f_\bullet$. For the analysis, we will assume that $f_\bullet$ is drawn from a Gaussian process, i.e., $f_\bullet \sim \mathcal{GP}(0, k)$, with a positive-semidefinite covariance (or kernel) function $k : \mathcal{X} \times \mathcal{X} \to \mathbb{R}$. In this case, we can show that the predictive uncertainty of the model converges

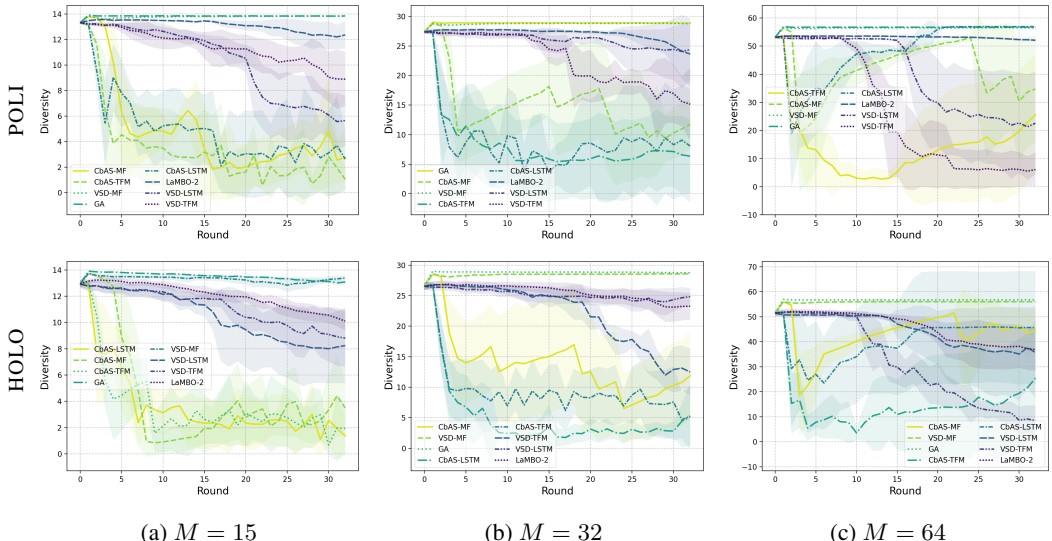

(a) $M = 15$        (b) $M = 32$        (c) $M = 64$

Figure D.7: Black-box optimization results for diversity on the POLI and HOLO implementations of the Ehrlich functions. Higher is more diverse, as defined by Equation 26.

(in probability) to zero as the number of observations grows. From this result, we prove asymptotic convergence guarantees for VSD equipped with GP-PI-based CPEs. These results form the basis for our analysis of CPEs based on neural networks (Appendix F).

### E.1 Gaussian process posterior

Let $f_\bullet \sim \mathcal{GP}(0, k)$ be a zero-mean Gaussian process with a positive-semidefinite covariance function $k : \mathcal{X} \times \mathcal{X} \to \mathbb{R}$. Assume that we are given a set $\mathcal{D}_N := \{(\mathbf{x}_i, y_i)\}_{i=1}^N$ of $N \geq 1$ observations $y_i = f_\bullet(\mathbf{x}_i) + \epsilon_i$, where $\epsilon \sim \mathcal{N}(0, \sigma_\epsilon^2)$ and $\mathbf{x}_i \in \mathcal{X}$. The GP posterior predictive distribution at any $\mathbf{x} \in \mathcal{X}$ is then given by (Rasmussen & Williams, 2006):

$$f_\bullet(\mathbf{x}) | \mathcal{D}_N \sim \mathcal{N}(\mu_N(\mathbf{x}), \sigma_N^2(\mathbf{x})) \tag{27}$$

$$\mu_N(\mathbf{x}) = \mathbf{k}_N(\mathbf{x})^\top (\mathbf{K}_N + \sigma_\epsilon^2 \mathbf{I})^{-1} \mathbf{y}_N \tag{28}$$

$$k_N(\mathbf{x}, \mathbf{x}') = k(\mathbf{x}, \mathbf{x}') - \mathbf{k}_N(\mathbf{x})^\top (\mathbf{K}_N + \sigma_\epsilon^2 \mathbf{I})^{-1} \mathbf{k}_N(\mathbf{x}') \tag{29}$$

$$\sigma_N^2(\mathbf{x}) = k_N(\mathbf{x}, \mathbf{x}), \tag{30}$$

where $\mathbf{k}_N(\mathbf{x}) := [k(\mathbf{x}, \mathbf{x}_i)]_{i=1}^N \in \mathbb{R}^N$, $\mathbf{K}_N := [k(\mathbf{x}_i, \mathbf{x}_j)]_{i,j=1}^{N,N} \in \mathbb{R}^{N \times N}$, and $\mathbf{y}_N := [y_i]_{i=1}^N \in \mathbb{R}^N$.

**Batch size.** In the following, we will assume a batch of size $B = 1$ to keep the proofs simple. With this assumption, at every iteration $t \geq 1$, we have $N = t$ observations available in the dataset. We would, however, like to emphasize that sampling a batch of multiple observations, instead of a single observation, per iteration should only improve the convergence rates by a constant (batch-size-dependent) multiplicative factor. Therefore, our results remain valid as an upper bound for the convergence rates of VSD in the batch setting.

### E.2 Background

We will consider an underlying probability space $(\Omega, \mathfrak{A}, \mathbb{P})$, where $\Omega$ is the sample space, $\mathfrak{A}$ denotes the $\sigma$-algebra of events, and $\mathbb{P}$ is a probability measure. For any event $\mathcal{A} \in \mathfrak{A}$, we have that $\mathbb{P}[\mathcal{A}] \in [0, 1]$ quantifies the probability of that event. For events involving a random variable, e.g., $\chi : (\Omega, \mathfrak{A}) \to (\mathbb{R}, \mathfrak{B}_\mathbb{R})$, where $\mathfrak{B}_\mathbb{R}$ denotes the Borel $\sigma$-algebra of the real line with its usual topology, we will let:

$$\mathbb{P}[\chi > 0] = \mathbb{P}[\{\omega \in \Omega : \chi(\omega) > 0\}]. \tag{31}$$

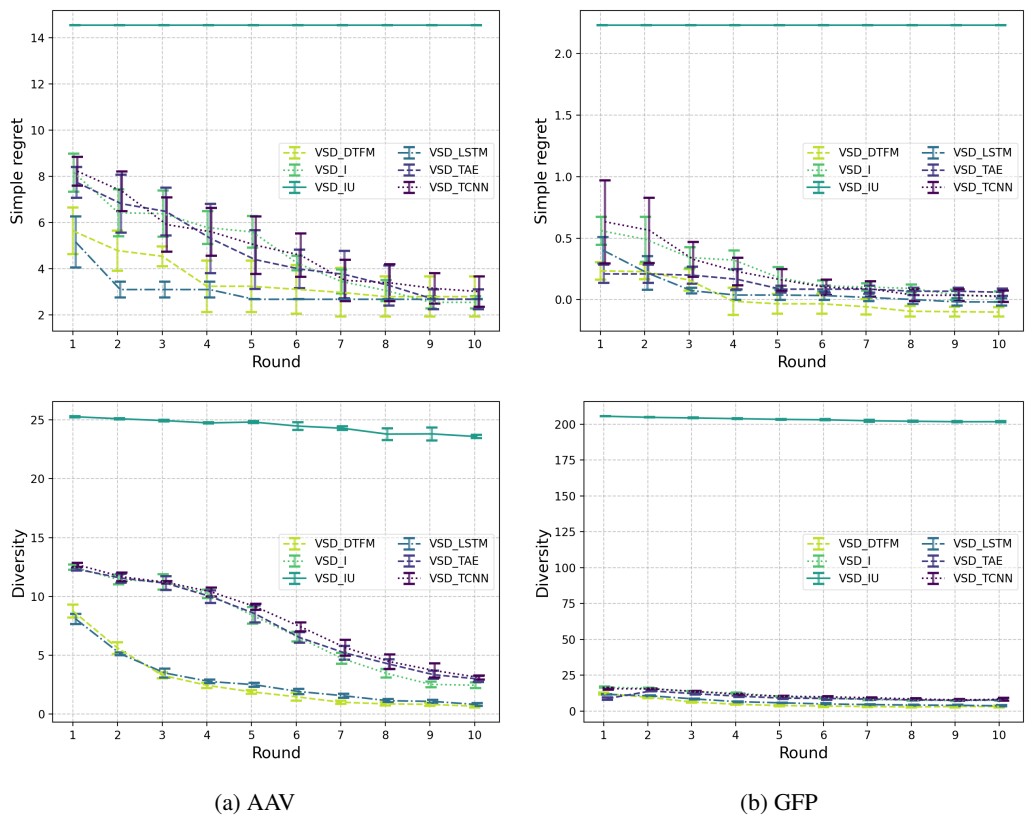

(a) AAV
(b) GFP

Figure D.8: Ablation results for the AAV and GFP BBO experiments. VSD is trialed with different prior and variational posterior combinations, "I" indicates a simple independent informed prior and posterior, "IU" is the same but with a uniform prior, "LSTM" and "DTFM" are the LSTM and decoder only transformer prior and posteriors, "TCNN" and "TAE" are transition convolutional encoder-decoder and auto-encoder posteriors, with informed independent priors. See text for details.

We will also use conditional expectations, i.e., given a $\sigma$-sub-algebra $\mathfrak{S}$ of $\mathfrak{A}$, the conditional expectation $\mathbb{E}[\chi|\mathfrak{S}]$ is a $\mathfrak{S}$-measurable random variable such that:

$$\forall \mathcal{A} \in \mathfrak{S}, \quad \int_{\mathcal{A}} \mathbb{E}[\chi|\mathfrak{S}] \, \mathrm{d}\mathbb{P} = \int_{\mathcal{A}} \chi \, \mathrm{d}\mathbb{P} = \mathbb{E}[\chi|\mathcal{A}]. \tag{32}$$

We will denote by $\{\mathfrak{F}_t\}_{t=0}^{\infty}$ an increasing filtration on $\mathfrak{A}$. For instance, we could set $\mathfrak{F}_t$ as the $\sigma$-algebra generated by the random variables in the algorithm (i.e., the candidates, target observations, etc.) at time $t$. For more details on the measure-theoretic definition of probability, we refer the reader to classic textbooks in the area (e.g. Bauer, 1981; Durrett, 2019)

We will use the following well known notation for asymptotic convergence results. For a given strictly positive function $g : \mathbb{N} \to \mathbb{R}$, we define $\mathcal{O}(g(t))$ as the set of functions asymptotically bounded by $g$ (up to a constant factor) as:

$$\mathcal{O}(g(t)) := \left\{ h : \mathbb{N} \to \mathbb{R} \,\middle|\, \limsup_{t \to \infty} \frac{|h(t)|}{g(t)} < \infty \right\}, \tag{33}$$

and for convergence in probability we use its stochastic counterpart:

$$\mathcal{O}_{\mathbb{P}}(g(t)) := \left\{ \rho : \mathbb{N} \times (\Omega, \mathfrak{A}) \to (\mathbb{R}, \mathfrak{B}_{\mathbb{R}}) \,\middle|\, \lim_{C \to \infty} \limsup_{t \to \infty} \mathbb{P}\left[ \frac{|\rho(t)|}{g(t)} > C \right] = 0 \right\}, \tag{34}$$

which is equivalent to:

$$\forall \varepsilon > 0, \quad \exists C_{\varepsilon} \in (0, \infty): \quad \mathbb{P}\left[ \frac{|\rho_t|}{g(t)} > C_{\varepsilon} \right] \leq \varepsilon, \quad \forall t \geq T_{\varepsilon}, \tag{35}$$

for some $T_\varepsilon \in \mathbb{N}$. For almost sure convergence, we may also say that a sequence of random variables $\rho_t$, $t \in \mathbb{N}$, is almost surely $\mathcal{O}(g(t))$ if $\mathbb{P}[\rho_t \in \mathcal{O}(g(t))] = 1$. A deeper overview on these notations and their properties can be found in García-Portugués (2024). In addition, as it is common in the multi-armed bandits literature, we use the variants $\widetilde{\mathcal{O}}$ and $\widetilde{\mathcal{O}}_{\mathbb{P}}$ to denote asymptotic rates which are valid up to logarithmic factors.

### E.3 Auxiliary results

We start with a few technical results which will form the basis for our derivations. The following recursive relations allow us to derive convergence rates for the variance of a GP posterior by analyzing how much it reduces per iteration.

**Lemma E.1** (Chowdhury & Gopalan (2017, Appendix F)). *The posterior mean and covariance functions of a Gaussian process given $t \geq 1$ observations obey the following recursive identities:*

$$\mu_t(\mathbf{x}) = \mu_{t-1}(\mathbf{x}) + \frac{k(\mathbf{x}, \mathbf{x}_t)}{\sigma_\epsilon^2 + \sigma_{t-1}^2(\mathbf{x}_t)}(y_t - \mu_{t-1}(\mathbf{x})) \tag{36}$$

$$k_t(\mathbf{x}, \mathbf{x}') = k_{t-1}(\mathbf{x}, \mathbf{x}') - \frac{k_{t-1}(\mathbf{x}, \mathbf{x}_t)k_{t-1}(\mathbf{x}_t, \mathbf{x}')}{\sigma_\epsilon^2 + \sigma_{t-1}^2(\mathbf{x}_t)} \tag{37}$$

$$\sigma_t^2(\mathbf{x}) = \sigma_{t-1}^2(\mathbf{x}) - \frac{k_{t-1}^2(\mathbf{x}, \mathbf{x}_t)}{\sigma_\epsilon^2 + \sigma_{t-1}^2(\mathbf{x}_t)}, \tag{38}$$

*for $\mathbf{x}, \mathbf{x}' \in \mathcal{X}$.*

We will also make use of the following version of the second Borel-Cantelli lemma adapted from Durrett (2019, Thr. 4.5.5) and its original statement in Dubins & Freedman (1965).

**Lemma E.2** (Second Borel-Cantelli lemma). *Let $\{\mathcal{A}_t\}_{t=1}^\infty$ be a sequence of events where $\mathcal{A}_t \in \mathfrak{F}_t$, for all $t \in \mathbb{N}$, and let $\chi_t : \omega \mapsto \mathbb{1}[\omega \in \mathcal{A}_t]$, for $\omega \in \Omega$. Then the following holds with probability 1:*

$$\lim_{T \to \infty} \frac{\sum_{t=1}^T \chi_t}{\sum_{t=1}^T \mathbb{P}[\mathcal{A}_t | \mathfrak{F}_{t-1}]} = L < \infty, \tag{39}$$

*assuming $\mathbb{P}[\mathcal{A}_1 | \mathfrak{F}_0] > 0$. In addition, if $\lim_{T \to \infty} \sum_{t=1}^T \mathbb{P}[\mathcal{A}_t | \mathfrak{F}_{t-1}] = \infty$, then $L = 1$.*

The next result provides us with an upper bound on the posterior variance of a Gaussian process which is valid for any covariance function.

**Lemma E.3.** *Let $k : \mathcal{X} \times \mathcal{X} \to \mathbb{R}$ be any positive-semidefinite kernel on $\mathcal{X}$, and let $\tilde{k} : \mathcal{X} \times \mathcal{X} \to \mathbb{R}$ be a kernel defined as:*

$$\tilde{k}(\mathbf{x}, \mathbf{x}') = \begin{cases} k(\mathbf{x}, \mathbf{x}), & \mathbf{x} = \mathbf{x}' \\ 0, & \mathbf{x} \neq \mathbf{x}', \end{cases} \tag{40}$$

*for $\mathbf{x}, \mathbf{x}' \in \mathcal{X}$. Given any set of observations $\{\mathbf{x}_i, y_i\}_{i=1}^t$, for $t \geq 1$, denote by $\sigma_t^2$ the predictive variance of a GP model with prior covariance given by $k$, and let $\tilde{\sigma}_t^2$ denote the predictive variance of a GP model configured with $\tilde{k}$ as prior covariance function, where both models are given the same set of observations. Then the following holds for all $t \geq 0$:*

$$\sigma_t^2(\mathbf{x}) \leq \tilde{\sigma}_t^2(\mathbf{x}) = \frac{\sigma_\epsilon^2 \tilde{\sigma}_0^2(\mathbf{x})}{\sigma_\epsilon^2 + N_t(\mathbf{x})\tilde{\sigma}_0^2(\mathbf{x})}, \quad \forall \mathbf{x} \in \mathcal{X}, \tag{41}$$

*where $N_t(\mathbf{x})$ denotes the number of observations at $\mathbf{x}$, and $\tilde{\sigma}_0^2(\mathbf{x}) = \sigma_0^2(\mathbf{x}) := k(\mathbf{x}, \mathbf{x})$, for $\mathbf{x} \in \mathcal{X}$.*

*Proof.* It is not hard to show that $\tilde{k}$ defines a valid positive-semidefinite covariance function whenever $k$ is positive semidefinite. We will then focus on proving the main statement by an induction argument. The proof that the statement holds for the base case at $t = 0$ is trivial given the definition:

$$\sigma_0^2(\mathbf{x}) = k(\mathbf{x}, \mathbf{x}) = \tilde{k}(\mathbf{x}, \mathbf{x}) = \tilde{\sigma}_0^2(\mathbf{x}), \quad \forall \mathbf{x} \in \mathcal{X}. \tag{42}$$

Now assume that, for a given $t > 0$, it holds that $\sigma_t^2(\mathbf{x}) \leq \tilde{\sigma}_t^2(\mathbf{x})$, for all $\mathbf{x} \in \mathcal{X}$. We will then check if the inequality remains valid at $t + 1$. By Lemma E.1, we have that:

$$\sigma_{t+1}^2(\mathbf{x}) = \sigma_t^2(\mathbf{x}) - \frac{k_t^2(\mathbf{x}, \mathbf{x}_{t+1})}{\sigma_t^2(\mathbf{x}_{t+1}) + \sigma_\epsilon^2} \tag{43}$$

For any $\mathbf{x} \in \mathcal{X}$ such that $\mathbf{x} \neq \mathbf{x}_{t+1}$, we know that $\tilde{k}_t(\mathbf{x}, \mathbf{x}_{t+1}) \geq 0$, so that (again by Lemma E.1):

$$\tilde{k}_t^2(\mathbf{x}, \mathbf{x}_{t+1}) \leq \tilde{k}^2(\mathbf{x}, \mathbf{x}_{t+1}) = 0 \,, \tag{44}$$

which shows that:

$$\forall \mathbf{x} \neq \mathbf{x}_{t+1}, \quad \sigma_{t+1}^2(\mathbf{x}) \leq \sigma_t^2(\mathbf{x}) \leq \tilde{\sigma}_t^2(\mathbf{x}) = \tilde{\sigma}_{t+1}^2(\mathbf{x}) \,. \tag{45}$$

At $\mathbf{x} = \mathbf{x}_{t+1}$, we can rewrite $\sigma_{t+1}^2(\mathbf{x}) = \sigma_{t+1}^2(\mathbf{x}_{t+1})$ as:

$$\sigma_{t+1}^2(\mathbf{x}_{t+1}) = \frac{\sigma_\epsilon^2 \sigma_t^2(\mathbf{x}_{t+1})}{\sigma_t^2(\mathbf{x}_{t+1}) + \sigma_\epsilon^2} \,. \tag{46}$$

We then check the difference:

$$\begin{aligned}
\sigma_{t+1}^2(\mathbf{x}_{t+1}) - \tilde{\sigma}_{t+1}^2(\mathbf{x}_{t+1}) &= \frac{\sigma_\epsilon^2 \sigma_t^2(\mathbf{x}_{t+1})}{\sigma_t^2(\mathbf{x}_{t+1}) + \sigma_\epsilon^2} - \frac{\sigma_\epsilon^2 \tilde{\sigma}_t^2(\mathbf{x}_{t+1})}{\tilde{\sigma}_t^2(\mathbf{x}_{t+1}) + \sigma_\epsilon^2} \\
&= \frac{\sigma_\epsilon^2 \sigma_t^2(\mathbf{x}_{t+1})(\tilde{\sigma}_t^2(\mathbf{x}_{t+1}) + \sigma_\epsilon^2) - \sigma_\epsilon^2 \tilde{\sigma}_t^2(\mathbf{x}_{t+1})(\sigma_t^2(\mathbf{x}_{t+1}) + \sigma_\epsilon^2)}{(\sigma_t^2(\mathbf{x}_{t+1}) + \sigma_\epsilon^2)(\tilde{\sigma}_t^2(\mathbf{x}_{t+1}) + \sigma_\epsilon^2)} \\
&= \frac{\sigma_\epsilon^4(\sigma_t^2(\mathbf{x}_{t+1}) - \tilde{\sigma}_t^2(\mathbf{x}_{t+1}))}{(\sigma_t^2(\mathbf{x}_{t+1}) + \sigma_\epsilon^2)(\tilde{\sigma}_t^2(\mathbf{x}_{t+1}) + \sigma_\epsilon^2)} \\
&\leq 0 \,,
\end{aligned} \tag{47}$$

since $\sigma_t^2(\mathbf{x}_{t+1}) \leq \tilde{\sigma}_t^2(\mathbf{x}_{t+1})$ by our assumption for time $t$. Therefore, we have shown that:

$$\sigma_t^2(\mathbf{x}) \leq \tilde{\sigma}_t^2(\mathbf{x}) \implies \sigma_{t+1}^2(\mathbf{x}) \leq \tilde{\sigma}_{t+1}^2(\mathbf{x}), \quad \forall \mathbf{x} \in \mathcal{X} \,. \tag{48}$$

From the conclusion above and the base case, the inequality in the main result follows by induction.

Now we derive an explicit form for $\tilde{\sigma}_t^2$. Note that this case corresponds to an independent Gaussian model, i.e., $f_\bullet(\mathbf{x}) \perp\!\!\!\perp f_\bullet(\mathbf{x}')$ whenever $\mathbf{x} \neq \mathbf{x}'$, for $f_\bullet \sim \mathcal{GP}(0, \tilde{k})$. For any $t \geq 1$, this model's predictive variance at any $\mathbf{x} \in \mathcal{X}$ is given by:

$$\tilde{\sigma}_t^2(\mathbf{x}) = \begin{cases} \tilde{\sigma}_{t-1}^2(\mathbf{x}), & \mathbf{x} \neq \mathbf{x}_t \\ \dfrac{\sigma_\epsilon^2 \tilde{\sigma}_{t-1}^2(\mathbf{x}_t)}{\sigma_\epsilon^2 + \tilde{\sigma}_{t-1}^2(\mathbf{x}_t)} = \left( \dfrac{1}{\tilde{\sigma}_{t-1}^2(\mathbf{x}_t)} + \dfrac{1}{\sigma_\epsilon^2} \right)^{-1}, & \mathbf{x} = \mathbf{x}_t \end{cases} \tag{49}$$

Looking at the reciprocal, we have that:

$$\forall t \geq 1, \quad \frac{1}{\tilde{\sigma}_t^2(\mathbf{x})} = \frac{1}{\tilde{\sigma}_{t-1}^2(\mathbf{x}_t)} + \frac{\mathbb{1}[\mathbf{x}_t = \mathbf{x}]}{\sigma_\epsilon^2}, \quad \forall \mathbf{x} \in \mathcal{X}. \tag{50}$$

Therefore, every observation at $\mathbf{x}$ is simply adding a factor of $\sigma_\epsilon^{-2}$ to $\tilde{\sigma}_t^{-2}(\mathbf{x})$. Unwrapping this recursion leads us to:

$$\forall t \geq 1, \quad \frac{1}{\tilde{\sigma}_t^2(\mathbf{x})} = \frac{1}{\tilde{\sigma}_0^2(\mathbf{x})} + \frac{1}{\sigma_\epsilon^2} \sum_{i=1}^t \mathbb{1}[\mathbf{x}_i = \mathbf{x}], \quad \forall \mathbf{x} \in \mathcal{X} \,. \tag{51}$$

The result in Lemma E.3 then follows as the reciprocal of the above, which concludes the proof. $\square$

**Lemma E.4.** *Let $f_\bullet \sim \mathcal{GP}(0, k)$ for a given $k : \mathcal{X} \times \mathcal{X} \to \mathbb{R}$, where $\sigma_\mathcal{X}^2 := \sup_{\mathbf{x} \in \mathcal{X}} k(\mathbf{x}, \mathbf{x}) < \infty$, and $|\mathcal{X}| < \infty$. Then $f_\bullet$ is almost surely bounded, and:*

$$\mathbb{E}\left[ \sup_{\mathbf{x} \in \mathcal{X}} |f_\bullet(\mathbf{x})| \right] \leq \sigma_\mathcal{X} \sqrt{2 \log |\mathcal{X}|} \,. \tag{52}$$

*Proof.* The result follows by an application of a concentration inequality for the maximum of a finite collection of sub-Gaussian random variables (Boucheron et al., 2013, Sec. 2.5). Note that $\{f_\bullet(\mathbf{x})\}_{\mathbf{x} \in \mathcal{X}}$ is a collection of $|\mathcal{X}|$ Gaussian, and therefore sub-Gaussian, random variables with sub-Gaussian parameter given by $\sigma_\mathcal{X}^2 \geq \sigma_t^2(\mathbf{x})$, for all $\mathcal{X}$. Applying the maximal inequality for a finite collection sub-Gaussian random variables (Boucheron et al., 2013, Thr. 2.5), we have that:

$$\mathbb{E}\left[ \max_{\mathbf{x} \in \mathcal{X}} f_\bullet(\mathbf{x}) \right] \leq \sigma_\mathcal{X} \sqrt{2 \log |\mathcal{X}|} < \infty \,. \tag{53}$$

By symmetry, we know that $-f_\bullet(\mathbf{x})$ is also sub-Gaussian with the same parameter, so that the bound remains valid for $\max_{\mathbf{x} \in \mathcal{X}} -f_\bullet(\mathbf{x})$. As a consequence, the expected value of the maximum of $|f_\bullet(\mathbf{x})|$ is upper bounded by the same constant. On a finite set, the maximum and the supremum coincide. As the expected value of the supremum is finite, the supremum must be almost surely finite by Markov's inequality, and therefore $f_\bullet$ is almost surely bounded. $\square$

### E.4 ASYMPTOTIC CONVERGENCE

The main assumption we will be working with in this section is the following.

**Assumption E.1.** *The objective function is a sample from a Gaussian process $f_\bullet \sim \mathcal{GP}(0, k)$, where $k : \mathcal{X} \times \mathcal{X} \to \mathbb{R}$ is a bounded positive-semidefinite kernel on $\mathcal{X}$.*

The next result allows us to derive a convergence rate for the posterior variance of a GP as a function of the sampling probabilities. This result might also be useful by itself for other sampling problems involving GP-based approximations.

**Lemma E.5.** *Let $\{\mathbf{x}_t\}_{t \geq 1}$ be a sequence of $\mathcal{X}$-valued random variables adapted to the filtration $\{\mathfrak{F}_t\}_{t \geq 1}$. For a given $\mathbf{x} \in \mathcal{X}$, assume that the following holds:*

$$\exists T_* \in \mathbb{N} : \quad \forall T \geq T_*, \quad \sum_{t=1}^{T} \mathbb{P}[\mathbf{x}_t = \mathbf{x} \mid \mathfrak{F}_{t-1}] \geq B_T > 0 \,, \tag{54}$$

*for a some sequence of lower bounds $\{B_t\}_{t \in \mathbb{N}}$. Then, under Assumption E.1, given observations at $\{\mathbf{x}_i\}_{i=1}^{t}$, the following holds with probability 1:*

$$\sigma_t^2(\mathbf{x}) \in \mathcal{O}(B_t^{-1}). \tag{55}$$

*In addition, if $B_t \to \infty$, then $\lim_{t \to \infty} B_t \sigma_t^2(\mathbf{x}) \leq \sigma_\epsilon^2$.*

*Proof.* At any iteration $t$, the posterior variance $\sigma_t^2$ of a GP model is upper bounded by a worst case assumption of no correlation between observations (see Lemma E.3). In this case, we have that:

$$\sigma_t^2(\mathbf{x}) \leq \tilde{\sigma}_t^2(\mathbf{x}) = \frac{\sigma_\epsilon^2 \tilde{\sigma}_0^2(\mathbf{x})}{\sigma_\epsilon^2 + N_t \tilde{\sigma}_0^2(\mathbf{x})} \,, \tag{56}$$

where $\tilde{\sigma}_0^2(\mathbf{x}) := \tilde{k}(\mathbf{x}, \mathbf{x}) = k(\mathbf{x}, \mathbf{x})$, and $N_t := N_t(\mathbf{x}) \leq t$ denotes the total number of observations taken at $\mathbf{x}$ as of iteration $t$. Without loss of generality, assume that $\tilde{\sigma}_0^2(\mathbf{x}) = 1$.

The only random variable to be bounded in Equation 56 is $N_t$. Let $\chi_t := \mathbb{1}[\mathbf{x}_t = \mathbf{x}]$, so that:

$$N_t = \sum_{i=1}^{t} \chi_i = \sum_{i=1}^{t} \mathbb{1}[\mathbf{x}_t = \mathbf{x}] \,, \quad t \geq 1. \tag{57}$$

We now apply the second Borel-Cantelli lemma (Lemma E.2) to $N_t$. Namely, let $\widehat{N}_t$ denote the sum of conditional expectations of $\{\chi_i\}_{i=1}^{t}$ given available data, i.e.:

$$\widehat{N}_t := \sum_{i=1}^{t} \mathbb{E}[\chi_i \mid \mathfrak{F}_{i-1}] = \sum_{i=1}^{t} \mathbb{E}[\mathbb{1}[\mathbf{x}_t = \mathbf{x}] \mid \mathfrak{F}_{i-1}] = \sum_{i=1}^{t} \mathbb{P}[\mathbf{x}_i = \mathbf{x} \mid \mathfrak{F}_{i-1}] \,. \tag{58}$$

By Lemma E.2, we know that the following holds for some $L \in \mathbb{R}$:

$$\lim_{t \to \infty} \frac{N_t}{\widehat{N}_t} = L < \infty \,. \tag{59}$$

Hence, $N_t$ is asymptotically equivalent to $\widehat{N}_t$. Applying this fact to $\tilde{\sigma}_t^2$, we have that:

$$
\begin{aligned}
\lim_{t \to \infty} B_t \tilde{\sigma}_t^2(\mathbf{x}) &= \lim_{t \to \infty} \frac{B_t \sigma_\epsilon^2}{\sigma_\epsilon^2 + N_t} \\
&= \lim_{t \to \infty} \frac{B_t \sigma_\epsilon^2}{\sigma_\epsilon^2 + L\widehat{N}_t} \\
&\leq \lim_{t \to \infty} \frac{B_t \sigma_\epsilon^2}{\sigma_\epsilon^2 + LB_t} \\
&\leq \frac{1}{L} \lim_{t \to \infty} \min\{LB_t, \sigma_\epsilon^2\} \\
&< \infty \,,
\end{aligned}
\tag{60}
$$

which holds with probability 1. Lastly, note that, if $B_t \to \infty$, then $L = 1$ by Lemma E.2, and the last limit above becomes $\sigma_\epsilon^2$. The main result then follows by an application of Lemma E.3 and the definition of the big-$\mathcal{O}$ notation (see Equation 33).[1]  $\square$

We assume a finite search space, which is the case for spaces of discrete sequences of bounded length. However, we conjecture that our results can be extended to continuous or mixed discrete-continuous search spaces via a discretization argument under further assumptions on the kernel $k$ (e.g., ensuring that $f_\bullet$ is Lipschitz continuous, as in Srinivas et al. (2010)).

**Assumption E.2.** *The search space $\mathcal{X}$ is finite, $|\mathcal{X}| < \infty$.*

We assume that our family of variational distributions is rich enough to be able to represent the PI-based distribution $p(\mathbf{x}|y > \tau_t, \mathcal{D}_t)$, which is the optimum of our variational objective when the optimal classifier is given by GP-PI. Although this assumption could be seen as strong, note that, due to Gaussian noise, the classification probability $p(y > \tau_t|\mathbf{x}, \mathcal{D}_t)$ should be a reasonably smooth function of $\mathbf{x}$, which facilitates the approximation of the resulting posterior by a generative model.

**Assumption E.3.** *For every $t \geq 0$, $p(\mathbf{x}|y > \tau_t, \mathcal{D}_t)$ is a member of the variational family, i.e.:*

$$\exists \phi_t^* : \quad \mathbb{D}[q(\mathbf{x}|\phi_t^*) \| p(\mathbf{x}|y > \tau_t, \mathcal{D}_t)] = 0. \tag{61}$$

The next assumption is a technical one to ensure that the thresholds will not diverge to infinity.

**Assumption E.4.** *The sequence of thresholds is almost surely bounded:[2]*

$$\sup_{t \in \mathbb{N}} |\tau_t| \leq \tau_* < \infty. \tag{62}$$

We can now state our main result regarding the GP-based approximations learned by VSD.

**Theorem E.1.** *Let assumptions E.1 to E.4 hold. Then the following holds with probability 1 for VSD equipped with GP-PI:*

$$\sigma_t^2(\mathbf{x}) \in \mathcal{O}(t^{-1}), \tag{63}$$

*at every $\mathbf{x} \in \mathcal{X}$ such that $p(\mathbf{x}) > 0$.*

*Proof.* Let $\ell_t(\mathbf{x}) := p(y > \tau_t|\mathbf{x}, \mathcal{D}_t)$. For any given $\mathbf{x} \in \mathcal{X}$ where $p(\mathbf{x}) > 0$, by Assumption E.2, we have that the next candidate will be sampled according to:

$$\forall t \geq 0, \quad \mathbb{P}[\mathbf{x}_{t+1} = \mathbf{x} \mid \mathfrak{F}_t] = p(\mathbf{x}|y > \tau_t, \mathcal{D}_t)$$
$$= \frac{\ell_t(\mathbf{x})p(\mathbf{x})}{\mathbb{E}_{p(\mathbf{x})}[\ell_t(\mathbf{x})]} \tag{64}$$
$$\geq \ell_t(\mathbf{x})p(\mathbf{x}),$$

where we used the fact that $\mathbb{E}_{p(\mathbf{x})}[\ell_t(\mathbf{x})] \leq 1$, since $\ell_t(\mathbf{x}) \leq 1$, for all $\mathbf{x} \in \mathcal{X}$. As $p(\mathbf{x}) > 0$, we only have to derive a lower bound on $\ell_t(\mathbf{x})$ to apply Lemma E.5 and derive a convergence rate.

A lower bound on $\ell_t(\mathbf{x})$ is given by:

$$\forall t \geq 0, \quad \ell_t(\mathbf{x}) = \Psi\left(\frac{\mu_t(\mathbf{x}) - \tau_t}{\sqrt{\sigma_t^2(\mathbf{x}) + \sigma_\epsilon^2}}\right) \geq \Psi\left(-\frac{\|\mu_t\|_\infty + \tau_*}{\sigma_\epsilon}\right), \tag{65}$$

where $\Psi(\cdot)$ denotes the cumulative distribution function of a standard normal random variable, and $\|\cdot\|_\infty$ denotes the essential supremum of a function under $\mathbb{P}$ (the probability measure of the underlying abstract probability space). Therefore, if $\lim_{t\to\infty} \|\mu_t\|_\infty < \infty$, we will have that $\lim_{t\to\infty} \ell_t(\mathbf{x}) > 0$, and the sum in Lemma E.5 will diverge.

By Jensen's inequality for conditional expectations, we have that:

$$\forall t \geq 0, \quad \|\mu_t\|_\infty = \|\mathbb{E}[f_\bullet \mid \mathfrak{F}_t]\|_\infty \leq \mathbb{E}[\|f_\bullet\|_\infty \mid \mathfrak{F}_t]. \tag{66}$$

---

[1] Recall that for convergent sequences lim and lim sup coincide.
[2] We do not require $\tau_*$ to be known, only finite.

As $\mathbb{E}[\mathbb{E}[\|f_\bullet\|_\infty \mid \mathfrak{F}_t]] = \mathbb{E}[\|f_\bullet\|_\infty] < \infty$ (cf. Lemma E.4), an application of Markov's inequality implies that:

$$\lim_{a \to \infty} \mathbb{P}[\mathbb{E}[\|f_\bullet\|_\infty \mid \mathfrak{F}_t] \geq a] \leq \lim_{a \to \infty} \frac{1}{a} \mathbb{E}[\|f_\bullet\|_\infty] = 0. \tag{67}$$

Furthermore, $m_t := \mathbb{E}[\|f_\bullet\|_\infty \mid \mathfrak{F}_t]$ also defines a non-negative martingale, and by the martingale convergence theorem (Durrett, 2019, Thr. 4.2.11), $\lim_{t \to \infty} m_t = m_\infty := \mathbb{E}[\|f_\bullet\|_\infty \mid \mathfrak{F}_\infty]$ is well defined and $\mathbb{E}[\mathbb{E}[\|f_\bullet\|_\infty \mid \mathfrak{F}_\infty]] = \mathbb{E}[\|f_\bullet\|_\infty] < \infty$. Again, by Markov's inequality, for any $a > 0$, we have that:

$$\mathbb{P}\left[\lim_{t \to \infty} \|\mu_t\|_\infty \geq a\mathbb{E}[\|f_\bullet\|_\infty]\right] \leq \frac{\mathbb{E}\left[\lim_{t \to \infty} \|\mu_t\|_\infty\right]}{a\mathbb{E}[\|f_\bullet\|_\infty]} \leq \frac{\mathbb{E}\left[\lim_{t \to \infty} \mathbb{E}[\|f_\bullet\|_\infty \mid \mathfrak{F}_t]\right]}{a\mathbb{E}[\|f_\bullet\|_\infty]} = \frac{1}{a}. \tag{68}$$

Therefore, for any $a > 0$ and any given $\mathbf{x} \in \mathcal{X}$, with probability at least $1 - \frac{1}{a}$, the following holds:

$$\begin{aligned}
\lim_{t \to \infty} \mathbb{P}[\mathbf{x}_t = \mathbf{x} \mid \mathfrak{F}_{t-1}] &\geq p(\mathbf{x}) \lim_{t \to \infty} \ell_{t-1}(\mathbf{x}) \\
&\geq p(\mathbf{x}) \lim_{t \to \infty} \Psi\left(-\frac{\|\mu_{t-1}\|_\infty + \tau_*}{\sigma_\epsilon}\right) \\
&\geq p(\mathbf{x}) \Psi\left(-\frac{a\mathbb{E}[\|f_\bullet\|_\infty] + \tau_*}{\sigma_\epsilon}\right) \\
&=: b_\infty(a) > 0.
\end{aligned} \tag{69}$$

Hence, for any $\varepsilon_a \in (0, b_\infty(a))$, there is $N_a \in \mathbb{N}$, such that $\mathbb{P}[\mathbf{x}_t = \mathbf{x} \mid \mathfrak{F}_{t-1}] \geq b_\infty(a) - \varepsilon_a > 0$, for all $t \geq N_a$. As a result, $\sum_{t'=1}^t \mathbb{P}[\mathbf{x}_{t'} = \mathbf{x} \mid \mathfrak{F}_{t'-1}] \geq (b_\infty(a) - \varepsilon_a)(t - N_a)$, for all $t \geq N_a$, which asymptotically diverges at a rate proportional to $t$. By Lemma E.5 and the definition of the big-$\mathcal{O}$ notation, for any $\mathbf{x} \in \mathcal{X}$, we then have that:

$$\forall a > 0, \quad \mathbb{P}\left[\limsup_{t \to \infty} |t\sigma_t^2(\mathbf{x})| \leq \sigma_\epsilon^2 < \infty\right] \geq 1 - \frac{1}{a}. \tag{70}$$

Taking the limit as $a \to \infty$, we can finally conclude that:

$$\mathbb{P}\left[\limsup_{t \to \infty} |t\sigma_t^2(\mathbf{x})| < \infty\right] = 1, \tag{71}$$

i.e., $\sigma_t^2$ is almost surely $\mathcal{O}(t^{-1})$, which concludes the proof. $\qquad\square$

**Remark E.1.** *The convergence rate in Theorem E.1 is optimal and cannot be further improved. As shown by previous works in the online learning literature (Mutný & Krause, 2018; Takeno et al., 2024), a lower bound on the GP variance at each iteration $t \geq 1$ is given by $\sigma_t^2(\mathbf{x}) \geq \sigma_\epsilon^2(\sigma_\epsilon^2 + t)^{-1}$ (assuming $k(\mathbf{x}, \mathbf{x}) = 1$), which is the case when every observation in the dataset was collected at the same point $\mathbf{x} \in \mathcal{X}$ (see Takeno et al., 2024, Lem. 4.2). Therefore, the lower and upper bounds on the asymptotic convergence rates for the GP variance differ by only up to a multiplicative constant.*

The result in Theorem E.1 now allows us to derive a convergence rate for VSD's approximations to the level-set distributions. To do so, however, we will require the following mild assumption, which is satisfied by any prior distribution which has support on the entire domain $\mathcal{X}$.

**Assumption E.5.** *The prior distribution is such that $p(\mathbf{x}) > 0$, for all $\mathbf{x} \in \mathcal{X}$.*

**Theorem 2.1.** *Under mild assumptions (E.1 to E.5), the variational distribution of VSD equipped with GP-PI converges to the level-set distribution in probability at the following rate:*

$$\mathbb{D}_{\mathrm{KL}}[p(\mathbf{x}|y > \tau_t, \mathcal{D}_t) \| p(\mathbf{x}|y > \tau_t, f_\bullet)] \in \mathcal{O}_{\mathbb{P}}(t^{-1/2}). \tag{12}$$

*Proof.* We first prove an upper bound for the KL divergence in terms of the PI approximation error. We then derive a bound for this term and apply Theorem E.1 to obtain a convergence rate.

*KL bound formulation.* Let $\ell_t(\mathbf{x}) := p(y > \tau_t|\mathbf{x}, \mathcal{D}_t)$ and $\ell_t^*(\mathbf{x}) := p(y > \tau_t|\mathbf{x}, f_\bullet)$, for $\mathbf{x} \in \mathcal{X}$. From the definition of the KL divergence, we have that:

$$\begin{aligned}
\mathbb{D}_{\mathrm{KL}}[p(\mathbf{x}|y > \tau_t, \mathcal{D}_t) \| p(\mathbf{x}|y > \tau_t, f_\bullet)] &= \mathbb{E}_{p(\mathbf{x}|y > \tau_t, \mathcal{D}_t)}[\log p(\mathbf{x}|y > \tau_t, \mathcal{D}_t) - \log p(\mathbf{x}|y > \tau_t, f_\bullet)] \\
&= \mathbb{E}_{p(\mathbf{x}|y > \tau_t, \mathcal{D}_t)}[\log \ell_t(\mathbf{x}) - \log \ell_t^*(\mathbf{x})] \\
&\quad + \log \mathbb{E}_{p(\mathbf{x})}[\ell_t^*(\mathbf{x})] - \log \mathbb{E}_{p(\mathbf{x})}[\ell_t(\mathbf{x})] \\
&= \mathbb{E}_{p(\mathbf{x}|y > \tau_t, \mathcal{D}_t)}\left[\log\left(\frac{\ell_t(\mathbf{x})}{\ell_t^*(\mathbf{x})}\right)\right] + \log\left(\frac{\mathbb{E}_{p(\mathbf{x})}[\ell_t^*(\mathbf{x})]}{\mathbb{E}_{p(\mathbf{x})}[\ell_t(\mathbf{x})]}\right).
\end{aligned} \tag{72}$$

For logarithms, we know that $\log(1 + a) \leq a$, for all $a > -1$, which shows that:

$$\log\left(\frac{\ell_t(\mathbf{x})}{\ell_t^*(\mathbf{x})}\right) = \log\left(1 + \frac{\ell_t(\mathbf{x}) - \ell_t^*(\mathbf{x})}{\ell_t^*(\mathbf{x})}\right) \leq \frac{\ell_t(\mathbf{x}) - \ell_t^*(\mathbf{x})}{\ell_t^*(\mathbf{x})} \tag{73}$$

$$\log\left(\frac{\mathbb{E}_{p(\mathbf{x})}[\ell_t^*(\mathbf{x})]}{\mathbb{E}_{p(\mathbf{x})}[\ell_t(\mathbf{x})]}\right) = \log\left(1 + \frac{\mathbb{E}_{p(\mathbf{x})}[\ell_t^*(\mathbf{x}) - \ell_t(\mathbf{x})]}{\mathbb{E}_{p(\mathbf{x})}[\ell_t(\mathbf{x})]}\right) \leq \frac{\mathbb{E}_{p(\mathbf{x})}[\ell_t^*(\mathbf{x}) - \ell_t(\mathbf{x})]}{\mathbb{E}_{p(\mathbf{x})}[\ell_t(\mathbf{x})]}. \tag{74}$$

Combining the above into Equation 72 yields:

$$\mathbb{D}_{\mathrm{KL}}[p(\mathbf{x}|y > \tau_t, \mathcal{D}_t) \| p(\mathbf{x}|y > \tau_t, f_\bullet)] \leq \mathbb{E}_{p(\mathbf{x}|y > \tau_t, \mathcal{D}_t)}\left[\frac{\ell_t(\mathbf{x}) - \ell_t^*(\mathbf{x})}{\ell_t^*(\mathbf{x})}\right] + \frac{\mathbb{E}_{p(\mathbf{x})}[\ell_t^*(\mathbf{x}) - \ell_t(\mathbf{x})]}{\mathbb{E}_{p(\mathbf{x})}[\ell_t(\mathbf{x})]}. \tag{75}$$

The denominator in the expression above is such that:

$$\forall t \geq 0, \quad \ell_t^*(\mathbf{x}) = p(y > \tau_t | \mathbf{x}, f_\bullet) = \Psi\left(\frac{f_\bullet(\mathbf{x}) - \tau_t}{\sigma_\epsilon}\right) \geq \Psi\left(-\frac{\|f_\bullet\|_\infty + \tau_*}{\sigma_\epsilon}\right), \quad \forall \mathbf{x} \in \mathcal{X}. \tag{76}$$

By Lemma E.4, we know that $\mathbb{E}[\|f_\bullet\|_\infty] < \infty$, which implies that $\mathbb{P}[\|f_\bullet\|_\infty < \infty] = 1$ by Markov's inequality. Next, we derive a bound for the approximation error term.

*Error bound.* We now derive an upper bound for the difference $\Delta\ell_t(\mathbf{x}) := \ell_t(\mathbf{x}) - \ell_t^*(\mathbf{x})$ and then show that it asymptotically vanishes. Applying Taylor's theorem to $\Psi$, we can bound $\Delta\ell_t$ as a function of the approximation error between the mean $\mu_t$ and the true function $f_\bullet$ as:

$$\begin{aligned}
\forall t \geq 0, \quad |\Delta\ell_t(\mathbf{x})| &= \left|\Psi\left(\frac{\mu_t(\mathbf{x}) - \tau_t}{\sqrt{\sigma_t^2(\mathbf{x}) + \sigma_\epsilon^2}}\right) - \Psi\left(\frac{f_\bullet(\mathbf{x}) - \tau_t}{\sigma_\epsilon}\right)\right| \\
&\leq \frac{1}{\sqrt{2\pi}}\left|\frac{\mu_t(\mathbf{x}) - \tau_t}{\sqrt{\sigma_t^2(\mathbf{x}) + \sigma_\epsilon^2}} - \frac{f_\bullet(\mathbf{x}) - \tau_t}{\sigma_\epsilon}\right| \\
&= \frac{1}{\sqrt{2\pi}}\left|\frac{\sigma_\epsilon \mu_t(\mathbf{x}) - f_\bullet(\mathbf{x})\sqrt{\sigma_t^2(\mathbf{x}) + \sigma_\epsilon^2} + \tau_t(\sqrt{\sigma_t^2(\mathbf{x}) + \sigma_\epsilon^2} - \sigma_\epsilon)}{\sigma_\epsilon\sqrt{\sigma_t^2(\mathbf{x}) + \sigma_\epsilon^2}}\right| \\
&\leq \frac{|\sigma_\epsilon \mu_t(\mathbf{x}) - f_\bullet(\mathbf{x})\sqrt{\sigma_t^2(\mathbf{x}) + \sigma_\epsilon^2}| + |\tau_t|\sigma_t(\mathbf{x})}{\sigma_\epsilon^2\sqrt{2\pi}} \\
&\leq \frac{\sigma_\epsilon|\mu_t(\mathbf{x}) - f_\bullet(\mathbf{x})| + \sigma_t(\mathbf{x})(|f_\bullet(\mathbf{x})| + |\tau_t|)}{\sigma_\epsilon^2\sqrt{2\pi}}, \quad \forall \mathbf{x} \in \mathcal{X},
\end{aligned} \tag{77}$$

since $\sup_{\epsilon \in \mathbb{R}}\left|\frac{\mathrm{d}\Psi(\epsilon)}{\mathrm{d}\epsilon}\right| = \frac{1}{\sqrt{2\pi}} < 1$, and we used the fact that $\sigma_\epsilon \leq \sqrt{\sigma_t^2(\mathbf{x}) + \sigma_\epsilon^2} \leq \sigma_t(\mathbf{x}) + \sigma_\epsilon$ to obtain the last two inequalities.

*Convergence rate.* To derive a convergence rate, given any $\mathbf{x} \in \mathcal{X}$ and $t \geq 0$, we have that:

$$\mathbb{E}[|\Delta\ell_t(\mathbf{x})| \mid \mathfrak{F}_t] \leq \frac{\sigma_\epsilon \mathbb{E}[|\mu_t(\mathbf{x}) - f_\bullet(\mathbf{x})| \mid \mathfrak{F}_t] + \sigma_t(\mathbf{x})(\mathbb{E}[|f_\bullet(\mathbf{x})| \mid \mathfrak{F}_t] + |\tau_t|)}{\sigma_\epsilon^2\sqrt{2\pi}}. \tag{78}$$

We know that $\mathbb{E}[|f_\bullet(\mathbf{x})| \mid \mathfrak{F}_t]$ is almost surely bounded, and by Jensen's inequality, it also holds that:

$$\mathbb{E}[|\mu_t(\mathbf{x}) - f_\bullet(\mathbf{x})| \mid \mathfrak{F}_t] \leq \sigma_t(\mathbf{x}). \tag{79}$$

Applying Theorem E.1, we then have that:

$$|\Delta\ell_t(\mathbf{x})| \in \mathcal{O}_\mathbb{P}(t^{-1/2}). \tag{80}$$

Since $\|\mu_t\|_\infty \leq \mathbb{E}[\|f_\bullet\|_\infty | \mathfrak{F}_t] \in \mathcal{O}_\mathbb{P}(1)$, we also have that:

$$\frac{1}{\mathbb{E}_{p(\mathbf{x})}[\ell_t(\mathbf{x})]} \in \mathcal{O}_\mathbb{P}(1). \tag{81}$$

Lastly, we know that $\frac{1}{\ell_t^*(\mathbf{x})} \in \mathcal{O}_\mathbb{P}(1)$ by Equation 76 and the observation that $\|f_\bullet\|_\infty \in \mathcal{O}_\mathbb{P}(1)$. The main result then follows by combining the rates above into Equation 75. $\qquad\square$

### E.5 PERFORMANCE ANALYSIS

At every iteration $t \geq 1$, VSD samples $\mathbf{x}_t$ from (an approximation to) the target $p(\mathbf{x}|y > \tau_{t-1}, \mathcal{D}_{t-1})$ and obtains an observation $y_t \sim p(y|\mathbf{x}_t)$. A positive hit consists of an event $y_t > \tau_{t-1}$, where $\tau_{t-1}$ is computed based on the data available in $\mathcal{D}_{t-1}$ or a constant. Therefore, we can compute the probability of a positive hit for a given realization of $f_\bullet$ as:

$$\mathbb{P}[y_t > \tau_{t-1} \mid \mathcal{D}_{t-1}, f_\bullet] = \mathbb{E}_{p(\mathbf{x}|y>\tau_{t-1}, \mathcal{D}_{t-1})}[p(y > \tau_{t-1}|\mathbf{x}, f_\bullet)]. \tag{82}$$

Then the expected number of hits $H_T$ after $T \geq 1$ iterations is given by:

$$\mathbb{E}[H_T \mid f_\bullet] = \sum_{t=1}^{T} \mathbb{E}_{p(\mathbf{x}|y>\tau_{t-1}, \mathcal{D}_{t-1})}[p(y > \tau_{t-1}|\mathbf{x}, f_\bullet)]. \tag{83}$$

We will compare this quantity with the expected number of hits $H_T^*$ obtained by a sampling distribution with full knowledge of the objective function $f_\bullet$:

$$\mathbb{E}[H_T^* \mid f_\bullet] = \sum_{t=1}^{T} \mathbb{E}_{p(\mathbf{x}|y>\tau_{t-1}, f_\bullet)}[p(y_t > \tau_{t-1}|\mathbf{x}, f_\bullet)]. \tag{84}$$

The next result allows us to bound the difference between these two quantities.

**Corollary 2.1.** *Under the settings in Theorem 2.1, we also have that:*

$$\mathbb{E}[|H_T - H_T^*|] \in \mathcal{O}(\sqrt{T}). \tag{13}$$

*Proof.* For all $T \geq 1$, we have that:

$$
\begin{aligned}
\mathbb{E}[H_T - H_T^*] &= \mathbb{E}\left[\sum_{t=1}^{T} \mathbb{E}_{p(\mathbf{x}|y>\tau_{t-1}, \mathcal{D}_{t-1})}[p(y > \tau_{t-1}|\mathbf{x}, f_\bullet)] - \mathbb{E}_{p(\mathbf{x}|y>\tau_{t-1}, f_\bullet)}[p(y_t > \tau_{t-1}|\mathbf{x}, f_\bullet)]\right] \\
&= \mathbb{E}\left[\sum_{t=1}^{T} \sum_{\mathbf{x}\in\mathcal{X}} p(y > \tau_{t-1}|\mathbf{x}, f_\bullet)\left(p(\mathbf{x}|y > \tau_{t-1}, \mathcal{D}_{t-1}) - p(\mathbf{x}|y > \tau_{t-1}, f_\bullet)\right)\right] \\
&= \mathbb{E}\left[\sum_{t=0}^{T-1} \sum_{\mathbf{x}\in\mathcal{X}} p(y > \tau_t|\mathbf{x}, f_\bullet)p(\mathbf{x})\left(\frac{\ell_t(\mathbf{x})}{\mathbb{E}_{p(\mathbf{x}')}[\ell_t(\mathbf{x}')]} - \frac{\ell_t^*(\mathbf{x})}{\mathbb{E}_{p(\mathbf{x}')}[\ell_t^*(\mathbf{x}')]}\right)\right] \\
&\leq \mathbb{E}\left[\sum_{t=0}^{T-1} \sum_{\mathbf{x}\in\mathcal{X}} p(\mathbf{x})\left(\frac{|\Delta\ell_t(\mathbf{x})|}{\min\{\mathbb{E}_{p(\mathbf{x}')}[\ell_t(\mathbf{x}')], \mathbb{E}_{p(\mathbf{x}')}[\ell_t^*(\mathbf{x}')]\}}\right)\right],
\end{aligned} \tag{85}
$$

since $p(y > \tau_{t-1}|\mathbf{x}, f_\bullet) \leq 1$, for all $t \geq 1$. As both $\|\mu_t\|_\infty$ and $\|f_\bullet\|_\infty$ are in $\mathcal{O}_\mathbb{P}(1)$, $\min\{\mathbb{E}_{p(\mathbf{x}')}[\ell_t(\mathbf{x}')], \mathbb{E}_{p(\mathbf{x}')}[\ell_t^*(\mathbf{x}')]\}$ is lower bounded by some constant. As $\Delta\ell_t(\mathbf{x}) \in \mathcal{O}_\mathbb{P}(t^{-1/2})$, for $T$ large enough and some $C > 0$, we then have that:

$$\mathbb{E}[|H_T - H_T^*|] \leq C \sum_{t=1}^{T} \frac{1}{\sqrt{t}} \leq 2C\sqrt{T} \in \mathcal{O}(\sqrt{T}), \tag{86}$$

which follows by an application of the Euler-Maclaurin formula, since $\int_1^T \frac{1}{\sqrt{t}}\, \mathrm{d}t = 2\sqrt{T} - 2$ and the remainder term asymptotically vanishes. $\qquad\square$

**Remark E.2.** *If the oracle achieves $\mathbb{E}[H_T^*] = T$, the error bound in Corollary 2.1 suggests an increasing rate of positive hits by VSD as $\frac{1}{T}\mathbb{E}[H_T] \geq 1 - CT^{-1/2}$, for some constant $C > 0$ and large enough $T$. Therefore, VSD should asymptotically achieve a full rate of 1 positive hit per iteration in the single-point batch setting we consider. Note, however, that the results above do not discount for repeated samples, though should still indicate that VSD achieves a high discovery rate over the course of its execution.*

# F VSD WITH NEURAL NETWORK CPES

In this section, we consider VSD with class probability estimators that are not based on GP regression, which was the case for the previous section, while specifically focusing on neural network models. We will, however, show that with a kernel-based formulation we are able to capture the classification models based on neural networks which we use. This is possible by analyzing the behavior of infinite-width neural networks (Jacot et al., 2018; Lee et al., 2019), whose approximation error with respect to the finite-width model can be bounded (Liu et al., 2020; Eldan et al., 2021).

Although our classifiers are learned by minimizing the cross-entropy (CE) loss, we can connect their approximations with theoretical results from the infinite-width neural network (NN) literature, which are mostly based on the mean squared error (MSE) loss. Recall that, given a dataset $\mathcal{D}_N^z :=$ $\{(\mathbf{x}_n, z_n)\}_{n=1}^N$ with binary labels $z_n \in \{0, 1\}$, the cross-entropy loss for a probabilistic classifier $\pi_\theta : \mathcal{X} \to [0, 1]$ parameterized by $\theta$ is given by[3]:

$$\mathcal{L}_{\mathrm{CPE}}(\theta, \mathcal{D}_N^z) := -\frac{1}{N} \sum_{n=1}^N z_n \log \pi_\theta(\mathbf{x}_n) + (1 - z_n) \log(1 - \pi_\theta(\mathbf{x}_n)). \tag{87}$$

The MSE loss for the same model corresponds to:

$$\mathcal{L}_{\mathrm{MSE}}(\theta, \mathcal{D}_N^z) := \frac{1}{N} \sum_{n=1}^N (z_n - \pi_\theta(\mathbf{x}_n))^2. \tag{88}$$

The following result establishes a connection between the two loss functions.

**Proposition F.1.** *Given a binary classification dataset $\mathcal{D}_N^z$ of size $N \geq 1$, the following holds for the cross-entropy and the mean-square error losses:*

$$\mathcal{L}_{CPE}(\theta, \mathcal{D}_N^z) \geq \mathcal{L}_{\mathrm{MSE}}(\theta, \mathcal{D}_N^z), \quad \forall N \in \mathbb{N}. \tag{89}$$

*Proof.* Applying the basic logarithmic inequality $\log(1 + a) \leq a$, for all $a > -1$, to the cross-entropy loss definition yields:

$$
\begin{aligned}
\mathcal{L}_{\mathrm{CPE}}(\theta, \mathcal{D}_N^z) &:= -\frac{1}{N} \sum_{n=1}^N z_n \log \pi_\theta(\mathbf{x}_n) + (1 - z_n) \log(1 - \pi_\theta(\mathbf{x}_n)) \\
&\geq -\frac{1}{N} \sum_{n=1}^N z_n (\pi_\theta(\mathbf{x}_n) - 1) - (1 - z_n)\pi_\theta(\mathbf{x}_n) \\
&= -\frac{1}{N} \sum_{n=1}^N 2z_n \pi_\theta(\mathbf{x}_n) - z_n - \pi_\theta(\mathbf{x}_n) \\
&= \frac{1}{N} \sum_{n=1}^N z_n - 2z_n \pi_\theta(\mathbf{x}_n) + \pi_\theta(\mathbf{x}_n).
\end{aligned}
\tag{90}
$$

Now note that $z_n = z_n^2$, for $z_n \in \{0, 1\}$, and $\pi_\theta(\mathbf{x}_n) \geq \pi_\theta(\mathbf{x}_n)^2$, as $\pi_\theta(\mathbf{x}_n) \in [0, 1]$, for all $n \in \{1, \dots, N\}$. Making these substitutions in Equation 90, we obtain:

$$\mathcal{L}_{\mathrm{CPE}}(\theta, \mathcal{D}_N^z) \geq \frac{1}{N} \sum_{n=1}^N z_n^2 - 2z_n \pi_\theta(\mathbf{x}_n) + \pi_\theta(\mathbf{x}_n)^2 = \mathcal{L}_{\mathrm{MSE}}(\theta, \mathcal{D}_N^z), \tag{91}$$

which concludes the proof. □

The result in Proposition F.1 suggests that minimizing the cross-entropy loss will lead us to minimize the MSE loss as well, since the latter is upper bounded by the former. This result provides us with theoretical justification to derive convergence results based on the MSE loss, which has been better analyzed in the NN literature (Jacot et al., 2018; Lee et al., 2019), as a proxy to establish convergence guarantees for the CE-based VSD setting.

---

[3]We implicitly assume that $0 < \pi_\theta(\mathbf{x}_n) < 1$, for $n \in \{1, \dots, N\}$, so that the CE loss is well defined. This assumption can, however, be relaxed when dealing with the MSE loss, which remains well defined otherwise.

## F.1 LINEAR APPROXIMATIONS VIA THE NEURAL TANGENT KERNEL

For this analysis, we will follow a frequentist setting. Namely, let $\pi^*$ denote the unknown true classifier, i.e., $\pi(\mathbf{x}) := p(y > \tau | \mathbf{x}, f_*)$, for $\mathbf{x} \in \mathcal{X}$. We assume that $\pi^*$ is an unknown, fixed element of a reproducing kernel Hilbert space (RKHS) associated with a given kernel (Schölkopf & Smola, 2001). In the case of infinite-width neural networks, we know that under certain assumptions the NN trained via gradient descent under the MSE loss will asymptotically converge to a kernel ridge regression solution whose kernel is given by the neural tangent kernel (NTK, Jacot et al., 2018). This asymptotic solution is equivalent to the posterior mean of a Gaussian process that assumes no observation noise. However, for a finite number of training steps $s < \infty$, the literature has shown that gradient-based training provides a form of implicit regularization, which we use to ensure robustness to label noise. Moreover, although our analysis will be based on the NTK, the approximation error between the infinite-width and the finite-width NN vanishes with the square root of the network width for most popular NN architectures (Liu et al., 2020). Therefore, we can assume that these approximation guarantees will remain useful for wide-enough, finite-width NN models.

**Reproducing kernel Hilbert spaces.** The RKHS $\mathcal{F}_k$ associated with a positive-semidefinite kernel $k : \mathcal{X} \times \mathcal{X} \to \mathbb{R}$ is a Hilbert space of functions over $\mathcal{X}$ with an inner product $\langle \cdot, \cdot \rangle_k$ and corresponding norm $\|\cdot\|_k := \sqrt{\langle \cdot, \cdot \rangle_k}$ such that, for every $\pi \in \mathcal{F}_k$, the reproducing property $\pi(\mathbf{x}) = \langle \pi, k(\cdot, \mathbf{x}) \rangle_k$ holds for all $\mathbf{x} \in \mathcal{X}$ (Schölkopf & Smola, 2001).

**Implicit regularization.** Several results in the literature have shown that training overparameterized neural networks via gradient descent provides a form of implicit regularization on the learned model (Fleming, 1990; Yao et al., 2007; Soudry et al., 2018; Barrett & Dherin, 2021), with some of the same behavior extending to the stochastic gradient setting (Smith et al., 2021). In earlier works, Fleming (1990) showed a direct equivalence between an early stopped gradient-descent linear model and the solution of a regularized least-squares problem with a penalty on the parameters vector Euclidean norm. In the NTK regime, the network output predictions at iteration $s \in \mathbb{N}$ of gradient descent are given by (Lee et al., 2019):

$$\hat{\pi}_N(\mathbf{x}) = \pi_0(\mathbf{x}) + \mathbf{k}_N(\mathbf{x})^\top \mathbf{K}_N^{-1}(\mathbf{I} - e^{-\nu s \mathbf{K}_N})(\mathbf{z}_N - \pi_0(\mathcal{X}_N)), \quad \mathbf{x} \in \mathcal{X}, \tag{92}$$

where $\pi_0$ represents the network's initialization, $\nu > 0$ denotes the learning rate, the kernel $k$ corresponds to the NTK associated with the given architecture, and the data is represented by $\mathcal{X}_N := \{\mathbf{x}_i\}_{i=1}^N \subset \mathcal{X}$ and $\mathbf{z}_N := [z_i]_{i=1}^N \in \{0, 1\}^N$. Rearranging terms and performing basic algebraic manipulations, the equation above can be shown to be equivalent to:

$$\hat{\pi}_N(\mathbf{x}) = \pi_0(\mathbf{x}) + \mathbf{k}_N(\mathbf{x})^\top (\mathbf{K}_N + \mathbf{\Sigma}_N)^{-1}(\mathbf{z}_N - \pi_0(\mathcal{X}_N)), \quad \mathbf{x} \in \mathcal{X}, \tag{93}$$

where $\mathbf{\Sigma}_N := \mathbf{K}_N(e^{\nu s \mathbf{K}_N} - \mathbf{I})^{-1}$ corresponds to a data-dependent regularization matrix. The above is equivalent to the solution of the a regularized least-squares problem, as we show below.

**Lemma F.1.** *Assume $k : \mathcal{X} \times \mathcal{X} \to \mathbb{R}$ is positive definite and $\pi_0 = 0$. Then Equation 93 solves the following regularized least-squares problem:*

$$\hat{\pi}_N \in \underset{\pi \in \mathcal{F}_k}{\operatorname{argmin}} \sum_{i=1}^N (\pi(\mathbf{x}_i) - z_i)^2 + \|\mathbf{R}_N^{1/2} \pi\|_k^2, \tag{94}$$

*where $\mathbf{R}_N := \mathbf{\Phi}_N(e^{-\nu s \mathbf{K}_N} - \mathbf{I})^{-1}\mathbf{\Phi}_N^\top$, $\mathbf{\Phi}_N := [\varphi(\mathbf{x}_1), \ldots, \varphi(\mathbf{x}_N)]$, and $\varphi(\mathbf{x}) := k(\cdot, \mathbf{x}) \in \mathcal{F}_k$ represents the kernel's canonical feature map, for $\mathbf{x} \in \mathcal{X}$.*

*Proof.* The least-squares loss can be rewritten as:

$$\ell_N(\pi) := \sum_{i=1}^N (\pi(\mathbf{x}_i) - z_i)^2 + \|\pi\|_{\mathbf{R}_N}^2 = \|\mathbf{\Phi}_N^\top \pi - \mathbf{z}_N\|_2^2 + \|\mathbf{R}_N^{1/2} \pi\|_k^2, \tag{95}$$

where $\|\cdot\|_2$ denotes the Euclidean norm of a vector. Taking the functional gradient with respect to $\pi \in \mathcal{F}_k$ and equating it to zero, we have that an optimal solution $\hat{\pi}$ satisfies:

$$\nabla \ell_N(\hat{\pi}) = 2\mathbf{\Phi}_N(\mathbf{\Phi}_N^\top \hat{\pi} - \mathbf{z}_N) + 2\mathbf{R}_N \hat{\pi} = 0. \tag{96}$$

The minimum-norm solution is then given by:

$$\hat{\pi} = (\mathbf{\Phi}_N \mathbf{\Phi}_N^\top + \mathbf{R}_N)^+ \mathbf{\Phi}_N \mathbf{z}_N \,, \tag{97}$$

where $\mathbf{A}^+$ denotes the Moore-Penrose pseudo-inverse of an operator $\mathbf{A} : \mathcal{F}_k \to \mathcal{F}_k$.

Let $\mathbf{\Phi}_N = \mathbf{U}_N \mathbf{S}_N \mathbf{V}_N^\top$ represent the singular value decomposition of $\mathbf{\Phi}_N$ in the RKHS (Mollenhauer et al., 2020), where $\mathbf{U}_N := [u_1, \ldots, u_N]$, $\mathbf{V}_N := [\mathbf{v}_1, \ldots, \mathbf{v}_N]$, with $\{u_i\}_{i=1}^N \subset \mathcal{F}_k$ and $\{\mathbf{v}_i\}_{i=1}^N \subset \mathbb{R}^N$ denoting the left and right singular vectors, respectively, and $\mathbf{S}_N \in \mathbb{R}^{N \times N}$ corresponds to the diagonal matrix of singular values of $\mathbf{\Phi}_N$. There are $N$ non-zero singular values, since $k$ is assumed to be positive definite, and we have the correspondence $\mathbf{K}_N = \mathbf{\Phi}_N^\top \mathbf{\Phi}_N = \mathbf{V}_N \mathbf{\Lambda}_N \mathbf{V}_N^\top$ with $\mathbf{\Lambda}_N = \mathbf{S}_N^2$ representing the diagonal matrix of eigenvalues of $\mathbf{K}_N$, which is full-rank for a positive-definite kernel with distinct entries $\mathcal{X}_N := \{\mathbf{x}_i\}_{i=1}^N \subset \mathcal{X}$. Applying the SVD to derive the pseudo-inverse in Equation 97 then yields:

$$
\begin{aligned}
\hat{\pi} &= (\mathbf{\Phi}_N \mathbf{\Phi}_N^\top + \mathbf{\Phi}_N (e^{-\nu s \mathbf{K}_N} - \mathbf{I})^{-1} \mathbf{\Phi}_N^\top)^+ \mathbf{\Phi}_N \mathbf{z}_N \\
&= (\mathbf{U}_N \mathbf{\Lambda}_N \mathbf{U}_N^\top + \mathbf{U}_N \mathbf{S}_N \mathbf{V}_N^\top (e^{-\nu s \mathbf{V}_N \mathbf{\Lambda}_N \mathbf{V}_N^\top} - \mathbf{I})^{-1} \mathbf{V}_N \mathbf{S}_N \mathbf{U}_N^\top)^+ \mathbf{U}_N \mathbf{S}_N \mathbf{V}_N^\top \mathbf{z}_N \\
&= \mathbf{U}_N (\mathbf{\Lambda}_N + \mathbf{\Lambda}_N (e^{-\nu s \mathbf{\Lambda}_N} - \mathbf{I})^{-1})^{-1} \mathbf{S}_N \mathbf{V}_N^\top \mathbf{z}_N \\
&= \mathbf{U}_N \mathbf{S}_N \mathbf{V}_N^\top \mathbf{V}_N (\mathbf{\Lambda}_N + \mathbf{\Lambda}_N (e^{-\nu s \mathbf{\Lambda}_N} - \mathbf{I})^{-1})^{-1} \mathbf{V}_N^\top \mathbf{z}_N \\
&= \mathbf{\Phi}_N (\mathbf{K}_N + \mathbf{K}_N (e^{-\nu s \mathbf{K}_N} - \mathbf{I})^{-1})^{-1} \mathbf{z}_N \\
&= \mathbf{\Phi}_N (\mathbf{K}_N + \mathbf{\Sigma}_N)^{-1} \mathbf{z}_N \,,
\end{aligned}
\tag{98}
$$

which concludes the proof. $\qquad\square$

For our analysis, we will assume that the classifier network is zero initialized with $\pi_0 = 0$, noting that the least-squares problem can always be solved for the residuals $z - \pi_0(\mathbf{x})$ and then have $\pi_0$ added back to the solution. We refer the reader to Lee et al. (2019) for further discussion on the effect of the network initialization.

**Approximation for finite-width networks.** For fully connected, convolutional or residual networks equipped with smooth activation functions (e.g., sigmoid or $\tanh$), Liu et al. (2020) showed that the approximation error between the linear model and the finite-width NN is $\widetilde{\mathcal{O}}(m^{-1/2})$, where $m$ denotes the minimum layer width, and the $\widetilde{\mathcal{O}}$ notation corresponds to the $\mathcal{O}$-notation with logarithmic factors suppressed. NTK results for other activation functions and different neural network architectures, such as multi-head attention (Hron et al., 2020), are also available in the literature.

### F.2 ASSUMPTIONS

In the following, we present a series of mild technical assumptions needed for our theoretical analysis of NN-based CPEs. For this analysis, we mainly assume that the true classifier $\pi^*(\mathbf{x}) = p(y > \tau | \mathbf{x}, f_\bullet)$ is a fixed, though unknown, element of the RKHS $\mathcal{F}_k$ given by a bounded NTK $k$, which is formalized by the following two assumptions. As in the GP case, we assume batches of size $B = 1$ to simplify the analysis.

**Assumption F.1.** *There is $\pi^* \in \mathcal{F}_k$ such that:*

$$\pi^*(\mathbf{x}) = p(y > \tau | \mathbf{x}, f_\bullet), \quad \forall \mathbf{x} \in \mathcal{X}. \tag{99}$$

For a rich enough RKHS, such assumption is mild, especially given that most popular NN architectures offer universal approximation guarantees (Hornik et al., 1989).

**Assumption F.2.** *The NTK $k$ corresponding to the network architecture in $\pi_\theta$ is positive definite and bounded in $\mathcal{X}$.*

We will also assume that the threshold is fixed to simplify the analysis. However, our results should asymptotically hold for time-varying thresholds as long as the limit $\lim_{t \to \infty} \tau_t = \tau$ exists.

**Assumption F.3.** *The threshold is fixed, i.e., $\tau_t = \tau \in \mathbb{R}$, for all $t \geq 1$.*

The following assumption on label noise should always hold for Bernoulli random variables (Boucheron et al., 2013). Any upper bound on the sub-Gaussian parameter should suffice for the analysis (e.g., $\sigma_\zeta \leq 1$ for Bernoulli variables).

**Assumption F.4.** *For all $t \in \mathbb{N}$ and all $\mathbf{x} \in \mathcal{X}$, label noise $\zeta = \mathbb{1}[y > \tau] - \pi^*(\mathbf{x})$, with $y \sim p(y|\mathbf{x}, f_\bullet)$, is $\sigma_\zeta$-sub-Gaussian:*

$$\forall a \in \mathbb{R}, \quad \mathbb{E}\left[\exp\left(a\zeta\right)\right] \leq \exp\left(\frac{a^2 \sigma_\zeta^2}{2}\right), \tag{100}$$

*for some $\sigma_\zeta \geq 0$.*

The next assumption ensures a sufficient amount of sampling is asymptotically achieved over the domain $\mathcal{X}$, which we still assume is finite.

**Assumption F.5.** *For any $t \geq 1$, the variational family is such that sampling probabilities are bounded away from 0, i.e.:*

$$\exists b > 0 : \quad \forall t \in \mathbb{N}, \quad q(\mathbf{x}|\phi_t) \geq b, \quad \forall \mathbf{x} \in \mathcal{X}. \tag{101}$$

The assumption above only imposes mild constraints on the generative models $q(\mathbf{x}|\phi)$, so that probabilities for all candidates $\mathbf{x} \in \mathcal{X}$ are never exactly 0, though still allowed to be arbitrarily small.

**Assumption F.6.** *The learning rate $\nu_t$ at each round $t$ is such that:*

$$0 < \nu_* \leq \nu_t \leq \frac{1}{\lambda_{\max}(\mathbf{K}_t)}, \quad \forall t \in \mathbb{N}, \tag{102}$$

*for some $\nu_* > 0$, where $\lambda_{\max}(\cdot)$ denotes the maximum eigenvalue of a matrix, and $\mathbf{K}_t$ denotes the NTK matrix evaluated at the training points available at iteration $t \geq 1$.*

This last assumption ensures that a gradient descent algorithm is convergent (Fleming, 1990), though we use it to bound the spectrum of the implicit regularization matrix $\boldsymbol{\Sigma}_t$ after a finite number of training steps $s < \infty$ (a.k.a. early stopping), which is needed for our results. We highlight that, under mild assumptions on the data distribution, $\lambda_{\max}(\mathbf{K}_t) \in \mathcal{O}(1)$ w.r.t. the number of data points (Murray et al., 2023), so that the bound in Assumption F.6 will not vanish. Therefore, such assumption is easily satisfied by maintaining a sufficiently small learning rate.

### F.3 Approximation error for NN-based CPEs

Similar to the GP-PI setting, we will assume a batch size of 1, so that we can simply use the iteration index $t \geq 0$ for our estimators. We recall that convergence rates for the batch setting should only be affected by a batch-size-dependent multiplicative factor, preserving big-$\mathcal{O}$ convergence rates. We start by defining the following *proxy* variance:

$$t \geq 1, \quad \hat{\sigma}_t^2(\mathbf{x}) = k(\mathbf{x}, \mathbf{x}) - \mathbf{k}_t(\mathbf{x})^\top (\mathbf{K}_t + \boldsymbol{\Sigma}_t)^{-1} \mathbf{k}_t(\mathbf{x}), \quad \mathbf{x} \in \mathcal{X}, \tag{103}$$

where $\boldsymbol{\Sigma}_t$ is the implicit regularization matrix in Equation 93 due to early stopping. The proxy variance is then equivalent to a GP posterior variance under the assumption of heteroscedastic (i.e., input dependent) Gaussian noise with covariance matrix given by $\boldsymbol{\Sigma}_t$. Given its similarities, we have that if enough sampling is asymptotically guaranteed, we can apply the same convergence results available for the GP-PI-based CPE, i.e., $\hat{\sigma}_t^2 \in \mathcal{O}(t^{-1})$ almost surely.

**Lemma F.2.** *Let assumptions F.2, F.5 and F.6 hold. Then the following almost surely holds for the proxy variance:*

$$\hat{\sigma}_t^2 \in \mathcal{O}(t^{-1}). \tag{104}$$

*Proof.* We first observe that $\hat{\sigma}_t^2$ (103) is upper bounded by the posterior predictive variance of a GP model assuming i.i.d. Gaussian noise (cf. Sec. E.1) with variance $\rho^*$ satisfying:

$$\rho^* \geq \lambda_{\max}(\boldsymbol{\Sigma}_t), \quad \forall t \in \mathbb{N}, \tag{105}$$

which is such that:

$$\begin{aligned}
\mathbf{\Sigma}_t &= \mathbf{K}_t(e^{\nu_t s \mathbf{K}_t} - \mathbf{I})^{-1} \\
&\preceq \mathbf{K}_t(\nu_t s \mathbf{K}_t)^{-1} \\
&\preceq \frac{1}{s\nu_t}\mathbf{I}
\end{aligned} \tag{106}$$

since $e^{\mathbf{A}} \succeq \mathbf{I} + \mathbf{A}$, for any Hermitian matrix $\mathbf{A}$, where $\succeq$ denotes the Loewner partial ordering in the space of positive-semidefinite matrices, i.e., $\mathbf{A} \succeq \mathbf{B}$ if and only if $\mathbf{A} - \mathbf{B}$ is positive semidefinite. Noting that the sum of sampling probabilities at any point $\mathbf{x} \in \mathcal{X}$ diverges as $t \to \infty$ by Assumption F.5, the result then follows by applying Lemma E.5 to the GP predictive variance upper bound with noise variance set to $\rho^* := (s\nu_*)^{-1}$ (Assumption F.6). □

**Lemma F.3.** *Let assumptions F.1 to F.6 hold. Then, given any $\delta \in (0, 1]$, the following holds with probability at least $1 - \delta$ for the approximation error between $\hat{\pi}_t$ and $\pi^*$:*

$$\forall t \geq 1, \quad |\hat{\pi}_t(\mathbf{x}) - \pi^*(\mathbf{x})| \leq \beta_t(\delta)\hat{\sigma}_t(\mathbf{x}), \quad \mathbf{x} \in \mathcal{X}, \tag{107}$$

*where $\beta_t(\delta) := \|\pi^*\|_k + \sigma_\zeta \sqrt{2\rho^{-1}\log(\det(\mathbf{I} + \rho^{-1}\mathbf{K}_t)^{1/2}/\delta)}$, and $\rho := \frac{e^{-s}}{s\nu_*}$.*

*Proof.* The result above is a direct application of Theorem 3.5 in Maillard (2016) which provides an upper confidence bound on the kernelized least-squares regressor approximation error (another version of the same result is also available in Durand et al. (2018, Thr. 1)).

Let $\zeta_i := z_i - \pi^*(\mathbf{x}_i)$ denote the label noise in observation $i$, for $i \in \{1, \ldots, t\}$. Expanding the definition of $\hat{\pi}_t$ (93) with $\pi_0 = 0$, given any $t \in \mathbb{N}$ and $\mathbf{x} \in \mathcal{X}$, we can decompose the approximation error as:

$$\begin{aligned}
|\pi^*(\mathbf{x}) - \hat{\pi}_t(\mathbf{x})| &= |\pi^*(\mathbf{x}) - \mathbf{k}_t(\mathbf{x})^\top(\mathbf{K}_t + \mathbf{\Sigma}_t)^{-1}\mathbf{z}_t| \\
&= |\pi^*(\mathbf{x}) - \mathbf{k}_t(\mathbf{x})^\top(\mathbf{K}_t + \mathbf{\Sigma}_t)^{-1}(\boldsymbol{\pi}_t^* + \boldsymbol{\zeta}_t)| \\
&\leq |\pi^*(\mathbf{x}) - \mathbf{k}_t(\mathbf{x})^\top(\mathbf{K}_t + \mathbf{\Sigma}_t)^{-1}\boldsymbol{\pi}_t^*| + |\mathbf{k}_t(\mathbf{x})^\top(\mathbf{K}_t + \mathbf{\Sigma}_t)^{-1}\boldsymbol{\zeta}_t|
\end{aligned} \tag{108}$$

where we applied the triangle inequality to obtain the last line. Analyzing the two terms on the right-hand side, by the reproducing property, we now have for the first term:

$$\begin{aligned}
|\pi^*(\mathbf{x}) - \mathbf{k}_t(\mathbf{x})^\top(\mathbf{K}_t + \mathbf{\Sigma}_t)^{-1}\boldsymbol{\pi}_t^*| &= |\langle \pi^*, (\mathbf{I} - \mathbf{\Phi}_t(\mathbf{K}_t + \mathbf{\Sigma}_t)^{-1}\mathbf{\Phi}_t^\top)\varphi(\mathbf{x})\rangle_k| \\
&= |\langle \pi^*, (\mathbf{I} + \mathbf{\Phi}_t\mathbf{\Sigma}_t^{-1}\mathbf{\Phi}_t^\top)^{-1}\varphi(\mathbf{x})\rangle_k| \\
&\leq \|\pi^*\|_k\|(\mathbf{I} + \mathbf{\Phi}_t\mathbf{\Sigma}_t^{-1}\mathbf{\Phi}_t^\top)^{-1}\varphi(\mathbf{x})\|_k \\
&= \|\pi^*\|_k\sqrt{\varphi(\mathbf{x})^\top(\mathbf{I} + \mathbf{\Phi}_t\mathbf{\Sigma}_t^{-1}\mathbf{\Phi}_t^\top)^{-2}\varphi(\mathbf{x})} \\
&\leq \|\pi^*\|_k\sqrt{\varphi(\mathbf{x})^\top(\mathbf{I} + \mathbf{\Phi}_t\mathbf{\Sigma}_t^{-1}\mathbf{\Phi}_t^\top)^{-1}\varphi(\mathbf{x})} \\
&= \|\pi^*\|_k\hat{\sigma}_t(\mathbf{x}),
\end{aligned} \tag{109}$$

where the second equality follows by an application of Woodbury's identity, the first inequality is due to Cauchy-Schwarz, the second inequality is due to the fact that $\mathbf{A}^{-2} \preceq \mathbf{A}^{-1}$ whenever $\mathbf{A} \succeq \mathbf{I}$, and the last line follows from the definition of $\hat{\sigma}_t^2$. For the remaining in term (108), we have that:

$$\begin{aligned}
|\mathbf{k}_t(\mathbf{x})^\top(\mathbf{K}_t + \mathbf{\Sigma}_t)^{-1}\boldsymbol{\zeta}_t| &= |\langle \varphi(\mathbf{x}), \mathbf{\Phi}_t(\mathbf{K}_t + \mathbf{\Sigma}_t)^{-1}\boldsymbol{\zeta}_t\rangle_k| \\
&= |\langle \varphi(\mathbf{x}), (\mathbf{I} + \mathbf{\Phi}_t\mathbf{\Sigma}_t^{-1}\mathbf{\Phi}_t^\top)^{-1}\mathbf{\Phi}_t\mathbf{\Sigma}_t^{-1}\boldsymbol{\zeta}_t\rangle_k| \\
&= |\langle (\mathbf{I} + \mathbf{\Phi}_t\mathbf{\Sigma}_t^{-1}\mathbf{\Phi}_t^\top)^{-1/2}\varphi(\mathbf{x}), (\mathbf{I} + \mathbf{\Phi}_t\mathbf{\Sigma}_t^{-1}\mathbf{\Phi}_t^\top)^{-1/2}\mathbf{\Phi}_t\mathbf{\Sigma}_t^{-1}\boldsymbol{\zeta}_t\rangle_k| \\
&\leq \sqrt{\varphi(\mathbf{x})^\top(\mathbf{I} + \mathbf{\Phi}_t\mathbf{\Sigma}_t^{-1}\mathbf{\Phi}_t^\top)^{-1}\varphi(\mathbf{x})}\|(\mathbf{I} + \mathbf{\Phi}_t\mathbf{\Sigma}_t^{-1}\mathbf{\Phi}_t^\top)^{-1/2}\mathbf{\Phi}_t\mathbf{\Sigma}_t^{-1}\boldsymbol{\zeta}_t\|_k \\
&= \hat{\sigma}_t(\mathbf{x})\sqrt{\boldsymbol{\zeta}_t^\top\mathbf{\Sigma}_t^{-1}\mathbf{\Phi}_t^\top(\mathbf{I} + \mathbf{\Phi}_t\mathbf{\Sigma}_t^{-1}\mathbf{\Phi}_t^\top)^{-1}\mathbf{\Phi}_t\mathbf{\Sigma}_t^{-1}\boldsymbol{\zeta}_t},
\end{aligned} \tag{110}$$

where we applied the identity $\mathbf{B}^\top(\mathbf{B}\mathbf{B}^\top + \mathbf{A})^{-1} = (\mathbf{I} + \mathbf{B}^\top\mathbf{A}^{-1}\mathbf{B})^{-1}\mathbf{B}^\top\mathbf{A}^{-1}$, which holds for an invertible matrix $\mathbf{A}$ (Searle, 1982), to obtain the second equality, and the upper bound follows by the Cauchy-Schwarz inequality. For the norm of the noise-dependent term, we will apply a

concentration bound by Abbasi-Yadkori (2012), which first requires a few transformations towards a non-time-varying regularization factor. Applying the SVD $\mathbf{\Phi}_t = \mathbf{U}_t \mathbf{S}_t \mathbf{V}_t^\top$ (Mollenhauer et al., 2020), as in the proof of Lemma F.1, we have that $\mathbf{\Sigma}_t = \mathbf{V}_t \mathbf{D}_t \mathbf{V}_t^\top$, where $\mathbf{D}_t := \mathbf{\Lambda}_t(e^{\nu_t s \mathbf{\Lambda}_t} - \mathbf{I})^{-1}$ and $\mathbf{\Lambda}_t = \mathbf{S}_t^2$, which leads us to:

$$
\begin{aligned}
\|(\mathbf{I} + \mathbf{\Phi}_t \mathbf{\Sigma}_t^{-1} \mathbf{\Phi}_t^\top)^{-1/2} \mathbf{\Phi}_t \mathbf{\Sigma}_t^{-1} \boldsymbol{\zeta}_t\|_k^2 &= \boldsymbol{\zeta}_t^\top \mathbf{\Sigma}_t^{-1} \mathbf{\Phi}_t^\top (\mathbf{I} + \mathbf{\Phi}_t \mathbf{\Sigma}_t^{-1} \mathbf{\Phi}_t^\top)^{-1} \mathbf{\Phi}_t \mathbf{\Sigma}_t^{-1} \boldsymbol{\zeta}_t \\
&= \boldsymbol{\zeta}_t^\top \mathbf{V}_t \mathbf{D}_t^{-1} \mathbf{S}_t \mathbf{U}_t^\top (\mathbf{I} + \mathbf{U}_t \mathbf{S}_t \mathbf{D}_t^{-1} \mathbf{S}_t \mathbf{U}_t^\top)^{-1} \mathbf{U}_t \mathbf{S}_t \mathbf{D}_t^{-1} \mathbf{V}_t^\top \boldsymbol{\zeta}_t \\
&= \boldsymbol{\zeta}_t^\top \mathbf{V}_t \mathbf{S}_t^2 \mathbf{D}_t^{-2} (\mathbf{I} + \mathbf{D}_t^{-1} \mathbf{S}_t^2)^{-1} \mathbf{V}_t^\top \boldsymbol{\zeta}_t \\
&= \boldsymbol{\zeta}_t^\top \mathbf{V}_t (\mathbf{S}_t^{-2} \mathbf{D}_t^2 + \mathbf{D}_t)^{-1} \mathbf{V}_t^\top \boldsymbol{\zeta}_t,
\end{aligned}
\tag{111}
$$

where we applied the identity $(\mathbf{I} + \mathbf{AB})^{-1}\mathbf{A} = \mathbf{A}(\mathbf{I} + \mathbf{BA})^{-1}$ (Searle, 1982). For the eigenvalues of $\mathbf{\Sigma}_t$, we have the following lower bound:

$$
\begin{aligned}
\mathbf{D}_t &= \mathbf{\Lambda}_t (e^{\nu_t s \mathbf{\Lambda}_t} - \mathbf{I})^{-1} \\
&= \mathbf{\Lambda}_t e^{-\nu_t s \mathbf{\Lambda}_t} (\mathbf{I} - e^{-\nu_t s \mathbf{\Lambda}_t})^{-1} \\
&\succeq \mathbf{\Lambda}_t e^{-\nu_t s \mathbf{\Lambda}_t} (\nu_t s \mathbf{\Lambda}_t)^{-1} \\
&= \frac{1}{\nu_t s} e^{-\nu_t s \mathbf{\Lambda}_t} \\
&\succeq \frac{1}{\nu_* s} e^{-s} \mathbf{I},
\end{aligned}
\tag{112}
$$

where the first inequality is due to $e^{-\mathbf{A}} \succeq \mathbf{I} - \mathbf{A}$, and the last inequality holds by Assumption F.6. Hence, setting $\rho := \frac{e^{-s}}{s\nu_*}$, we have that:

$$
\begin{aligned}
\|(\mathbf{I} + \mathbf{\Phi}_t \mathbf{\Sigma}_t^{-1} \mathbf{\Phi}_t^\top)^{-1/2} \mathbf{\Phi}_t \mathbf{\Sigma}_t^{-1} \boldsymbol{\zeta}_t\|_k^2 &\leq \boldsymbol{\zeta}_t^\top \mathbf{V}_t (\rho^2 \mathbf{S}_t^{-2} + \rho \mathbf{I})^{-1} \mathbf{V}_t^\top \boldsymbol{\zeta}_t \\
&= \rho \boldsymbol{\zeta}_t^\top \mathbf{V}_t \mathbf{S}_t (\rho \mathbf{I} + \mathbf{S}_t^2)^{-1} \mathbf{S}_t \mathbf{V}_t^\top \boldsymbol{\zeta}_t \\
&= \rho^{-1} \boldsymbol{\zeta}_t^\top \mathbf{V}_t \mathbf{S}_t (\rho \mathbf{I} + \mathbf{S}_t^2)^{-1} \mathbf{S}_t \mathbf{V}_t^\top \boldsymbol{\zeta}_t \\
&= \rho^{-1} \boldsymbol{\zeta}_t^\top \mathbf{\Phi}_t^\top (\rho \mathbf{I} + \mathbf{\Phi}_t \mathbf{\Phi}_t^\top)^{-1} \mathbf{\Phi}_t \boldsymbol{\zeta}_t \\
&= \|(\rho \mathbf{I} + \mathbf{\Phi}_t \mathbf{\Phi}_t^\top)^{-1/2} \mathbf{\Phi}_t \boldsymbol{\zeta}_t\|_k^2
\end{aligned}
\tag{113}
$$

By Abbasi-Yadkori (2012, Cor. 3.6), given any $\delta \in (0, 1]$, we then have that the following holds with probability at least $1 - \delta$:

$$
\forall t \geq 1, \quad \|(\mathbf{I} + \mathbf{\Phi}_t \mathbf{\Sigma}_t^{-1} \mathbf{\Phi}_t^\top)^{-1/2} \mathbf{\Phi}_t \mathbf{\Sigma}_t^{-1} \boldsymbol{\zeta}_t\|_k^2 \leq 2\sigma_\zeta^2 \log \left( \frac{\det(\mathbf{I} + \rho^{-1} \mathbf{K}_t)^{1/2}}{\delta} \right).
\tag{114}
$$

Finally, combining the bounds above into Equation 108 leads to the result in Lemma F.3. $\square$

For the next result, we need to define the following quantity:

$$
\xi_T := \max_{\mathcal{X}_T \subset \mathcal{X} : |\mathcal{X}_T| \leq T} \frac{1}{2} \log \det(\mathbf{I} + \rho^{-1} \mathbf{K}(\mathcal{X}_T)),
\tag{115}
$$

where $\mathbf{K}(\mathcal{X}_T) := [k(\mathbf{x}, \mathbf{x}')]_{\mathbf{x}, \mathbf{x}' \in \mathcal{X}_T} \in \mathbb{R}^{|\mathcal{X}_T| \times |\mathcal{X}_T|}$. Note that $\xi_T$ corresponds to the maximum information gain of a GP model (Srinivas et al., 2010) with covariance function given by the NTK, assuming Gaussian observation noise with variance given by $\rho$. Then $\xi_T$ is mainly dependent on the eigenvalue decay of the kernel under its spectral decomposition (Vakili et al., 2021). For the spectrum of the NTK, a few results are available in the literature (Murray et al., 2023).

**Proposition F.2.** *Let assumptions F.1 to F.6 hold. Then, given $\delta \in (0, 1]$, the following holds with probability at least $1 - \delta$ for VSD equipped with a wide enough NN-based CPE model $\hat{\pi}_t$:*

$$
\mathbb{D}_{\mathrm{KL}}[p(\mathbf{x}|y > \tau_t, \mathcal{D}_t) \| p(\mathbf{x}|y > \tau_t, f_\bullet)] \in \mathcal{O}_{\mathbb{P}} \left( \sqrt{\frac{\xi_t}{t}} \right).
\tag{116}
$$

*Proof.* The result follows by applying the same steps as in the proof of Theorem 2.1. We note that $\ell_t^*(\mathbf{x}) = \pi^*(\mathbf{x}) > 0$, due to observation noise, so that $\ell_t^*(\mathbf{x})^{-1} \in \mathcal{O}_\mathbb{P}(1)$. Similarly, Lemma F.3 implies that $|\hat{\pi}_t(\mathbf{x}) - \pi^*(\mathbf{x})| \le \beta_t(\delta)\sigma_t(\mathbf{x})$ with probability at least $1 - \delta$ simultaneously over all $\mathbf{x} \in \mathcal{X}$, so that ratio-dependent terms in Theorem 2.1 should remain bounded in probability. The upper bound in the result then follows by noticing that in our case $|\Delta\ell_t(\mathbf{x})| \le \beta_t(\delta)\hat{\sigma}_t(\mathbf{x})$ with high probability, where $\hat{\sigma}_t \in \mathcal{O}(t^{-1/2})$ by Lemma F.2, and $\beta_t(\delta) \in \mathcal{O}(\sqrt{\xi_t})$ by Lemma F.3 and the definition of $\xi_t$ in Equation 115. $\qquad\square$

The result above tells us that VSD equipped with an NN-based CPE can recover a similar asymptotic convergence guarantee to the one we derived for the GP-PI case, depending on the choice of NN architecture and more specifically on the spectrum of its associated NTK. In the case of a fully connected multi-layer ReLU network, for example, Chen & Xu (2021) showed an equivalence between the RKHS of the ReLU NTK and that of the Laplace kernel $k(\mathbf{x}, \mathbf{x}') = \exp(-C\|\mathbf{x} - \mathbf{x}'\|)$. As the latter is equivalent to a Matérn kernel with smoothness parameter set to $0.5$ (Rasmussen & Williams, 2006), the corresponding information gain bound is $\xi_t \in \widetilde{\mathcal{O}}(t^{\frac{d}{1+d}})$, where $d$ here denotes the dimensionality of the domain $\mathcal{X}$ (Vakili & Olkhovskaya, 2023). In the case of discrete sequences of length $M$, the dimensionality of $\mathcal{X}$ is determined by $M$. Hence, we have proven Corollary 2.2.[4]

**Corollary 2.2.** *Let $\pi_\theta$ be modeled via a fully connected ReLU network. Then, under assumptions on identifiability and sampling (F.1 to F.6), in the infinite-width limit, VSD with CPE-PI achieves:*

$$\mathbb{D}_{\mathrm{KL}}[p(\mathbf{x}|y > \tau_t, \mathcal{D}_t)\|p(\mathbf{x}|y > \tau_t, f_\bullet)] \in \widetilde{\mathcal{O}}_\mathbb{P}\left(t^{-\frac{1}{2(M+1)}}\right). \tag{14}$$

Similar steps can be applied to derive convergence guarantees for VSD with other neural network architectures based on the eigenspectrum of their NTK (Murray et al., 2023) and following the recipe in, e.g., Vakili et al. (2021) or Srinivas et al. (2010).

## G   VSD AS A BLACK-BOX OPTIMIZATION LOWER BOUND

A natural question to ask is how VSD relates to the BO objective for probability of improvement (Garnett, 2023, Ch.7),

$$\mathbf{x}_t^* = \underset{\mathbf{x}}{\operatorname{argmax}} \log \alpha_{PI}(\mathbf{x}, \mathcal{D}_N, \tau). \tag{117}$$

Firstly, we can see that the expected log-likelihood of term of Equation 6 lower-bounds this quantity.

**Proposition G.1.** *For a parametric model, $q(\mathbf{x}|\phi)$, given $\phi \in \Phi \subseteq \mathbb{R}^m$ and $q \in \mathcal{P} : \mathcal{X} \times \Phi \to [0, 1]$,*

$$\max_{\mathbf{x}} \log \alpha_{PI}(\mathbf{x}, \mathcal{D}_N, \tau) \ge \max_\phi \mathbb{E}_{q(\mathbf{x}|\phi)}[\log \alpha_{PI}(\mathbf{x}, \mathcal{D}_N, \tau)], \tag{118}$$

*and the bound becomes tight as $q(\mathbf{x}|\phi_t^*) \to \delta(\mathbf{x}_t^*)$, a Dirac delta function at the maximizer $\mathbf{x}_t^*$.*

Taking the argmax of the RHS will result in the variational distribution collapsing to a delta distribution at $\mathbf{x}_t^*$ for an appropriate choice of $q(\mathbf{x}|\phi)$. The intuition for Equation 118 is that the expected value of a random variable is always less than or equal to its maximum. The proof of this is in Daulton et al. (2022); Staines & Barber (2013). Extending this lower bound, we can show the following.

**Proposition G.2.** *For a divergence $\mathbb{D} : \mathcal{P}(\mathcal{X}) \times \mathcal{P}(\mathcal{X}) \to [0, \infty)$, and a prior $p_0 \in \mathcal{P}(\mathcal{X})$,*

$$\max_{\mathbf{x}} \log \alpha_{PI}(\mathbf{x}, \mathcal{D}_N, \tau) \ge \max_\phi \mathbb{E}_{q(\mathbf{x}|\phi)}[\log \alpha_{PI}(\mathbf{x}, \mathcal{D}_N, \tau)] - \mathbb{D}[q(\mathbf{x}|\phi)\|p_0(\mathbf{x})]. \tag{119}$$

We can see that this bound is trivially true given the range of divergences, and this covers VSD as a special case. However, this bound is tight if and only if $p_0$ concentrates as a Dirac delta at $\mathbf{x}_t^*$ with an appropriate choice of $q(\mathbf{x}|\phi)$. In any case, the lower bound remains valid for any choice of informative prior $p_0$ or even a uninformed prior, which allows us to maintain the framework flexible to incorporate existing prior information whenever that is available.

---

[4]Here $\widetilde{\mathcal{O}}_\mathbb{P}$ suppresses logarithmic factors, as in $\widetilde{\mathcal{O}}$, and holds in probability.

