# OpenReview forum: "Variational Search Distributions"
_ICLR.cc/2025/Conference — ICLR 2025 Poster_

### Official Review · Reviewer_9cf6 · 2024-10-30

**Soundness:** 4
**Presentation:** 3
**Contribution:** 3
**Rating:** 6
**Confidence:** 3

**Summary:**

The authors present a novel variational method for learning to sample from rarely observed events, aiming to minimize a distance between the distribution of interest, namely $p(x∣y>t)$, and its parametric variational counterpart $q(x|\phi)$. The problem is reformulated to leverage the “whole dataset,” not just rarely observed events, and is expressed as Equation (5), which comprises two terms: $log p (y>t∣x)$ and the negative KL divergence between$q(x|\phi)$ and $p(x)$. The authors' final proposal is to estimate $p(y>t∣x)$ using a parametric function instead of a simple PI estimate. The variational distribution is optimized by a REINFORCE gradient estimator.

**Strengths:**

The paper is clear, well-written, and aligns with well-established benchmarks in the field, such as CBAS (Brookes et al.).
The model is supported by convergence analysis and an extensive set of well-handled experiments.

**Weaknesses:**

While the model description is clear, the model comprise a parametric distribution $p(x|D_0)$ which might be the biggest model shortcoming originating from the model own formulation.

Its major impact is that it reweights the gradient estimates of $q(x|\phi)$. Intuitively, how would that compare simply to the iterative strategy of Cbas ?

**Questions:**

1.Since your algorithm heavily relies on another model ($p(x | D)$), I would be highly interested in better understanding the influence of a good prior on your variational distribution.
2. Regarding the GFP experiments, do you sample already existing sequences ? What is the influence of the relative poor performance of the oracle on ood data on the interpretation of the results  ?
3. How can you explain that only a very simple prior such as a mean field performs on average better ?  It seems quite logical for GFP for instance where a wild type exists, however it is less intuitive for datasets without wild type.

Typo: the recall and precision have the same expression.

---

> ### Author Response · Authors · 2024-11-22
>
> Thank you to reviewer 9cf6 for your questions concerning the impact of the choice of prior and the GFP dataset.
>
> __Weaknesses__:
>
> *While the model description is clear, the model comprise a parametric distribution $p(x|D_0)$ which might be the biggest model shortcoming originating from the model own formulation. Its major impact is that it reweights the gradient estimates of $q(x|\phi)$. Intuitively, how would that compare simply to the iterative strategy of Cbas ?*
>
> Thanks for the insightful question. We do not understand why using a parametric prior $p(x|D_0)$ in this context is a shortcoming? Perhaps you could expand on what you mean? Thank you.
>
> In terms of comparison to CbAS, we present a comparison of re-weighted gradient estimators in Eqn. 14 and Table 1. We note that CbAS also requires a parametric prior distribution $p(x|D_0)$. VSD and CbAS have quite similar gradient re-weighting strategies, the largest differences being that VSD using log-probabilities for its reweighing scheme (implying better numerical behavior), and most importantly, its samples, $x^{(s)}$, for estimating the expectation in Eqn. 14 are drawn from the most recent estimate of $q(x|\phi)$ within each iteration of maximizing the ELBO -- i.e. the gradient weights and samples are adapted. This is unlike CbAS in which the weights are a function of $q(x|\phi _{t-1})$ and we only sample $q(x|\phi _{t-1})$ once at the beginning of the $t$th round, and keep these samples fixed while maximizing the weighted log-likelihood of q -- i.e. the gradient weights and samples are fixed.
>
> __Questions__:
>
> 1. *Since your algorithm heavily relies on another model ($p(x|D_0)$), I would be highly interested in better understanding the influence of a good prior on your variational distribution.*
>
> This is an important question -- and we provide some empirical exploration in the ablation studies in  Appendix C.3 for the high dimensional BBO experiments. In particular if we consider the mean field prior and variational distributions in Figure 9 -- we can see an independent and uniform prior over tokens (VSD-IU) leads to almost no progress. Whereas simply fitting a mean field prior to the sequences in the initial training data (VSD-I) leads to a massive performance gain. More expressive priors, e.g. LSTM's and transformers fit to the initial training data using cross entropy loss/maximum likelihood lead to even better initial performance.
>
> This shows that it is of vital importance to have some way of initially constraining/guiding the search for feasible sequences in these high-dimensional and combinatorial settings.
>
> 2. *Regarding the GFP experiments, do you sample already existing sequences ? What is the influence of the relative poor performance of the oracle on ood data on the interpretation of the results ?*
>
> We use experimentally validated sequences for the initial training data (CPE and prior), but then we use the oracle predictions for the black box evaluations. Unfortunately, use of this oracle does mean these results may be less applicable to real-world sequential optimization tasks, and this is a known limitation in this body of literature generally. However, we have incorporating more experiments based on reviewer xamp's feedback though the use of the newly released Ehrlich function in the `poli` suite of benchmarks, see Appendix C.4 for early results.
>
> 3. *How can you explain that only a very simple prior such as a mean field performs on average better ? It seems quite logical for GFP for instance where a wild type exists, however it is less intuitive for datasets without wild type.*
>
> The mean field prior only performs as well as the more complex priors on the lower-dimensional fitness landscape tasks (DHFR, TrpB, TFBIND8). On the higher dimensional BBO tasks (GFP and AAV), it performs worse, see Figure 9 in Appendix C.3. We do not use wild-type information for the AAV and GFP experiments as this makes the tasks too easy, see \citet{kirjner2024improving} for a discussion.
>
> *Typo: the recall and precision have the same expression.*
>
> Precision (Eqn. 15) and Recall (Eqn. 16) differ in their normalization constants, with precision's being a function of round, t. At t=T these quantities will be the same.

---

> > ### Comment · Reviewer_9cf6 · 2024-11-25
> > **Response to the authors' rebuttal**
> >
> > First, thanks for answering my questions.
> >
> > >  We do not understand why using a parametric prior in this context is a shortcoming? Perhaps you could expand on what you mean? Thank you.
> >
> > In a limited and  high dimensional data regimen, the model $p(x|D)$ can be inaccurate or even difficult to fit.It is also dependent on the collection method, for instance in the GFP setting, mutants are observed based on random mutation conducted in wet-labs experiments making it more difficult to interpret.
> >
> > Given the overall rebuttal, I maintain my score.

---

> > > ### Author Response · Authors · 2024-11-26
> > >
> > > Thank you for the clarification -- this is an interesting observation, and in this context highlights an advantage of VSD over competing methods.
> > >
> > > VSD used $p(x|D)$ as a prior only, and only requires $\log p(x|D)$ scores as a means of regularising the generative posterior distribution $q(x|y > \tau)$ to areas of $\mathcal{X}$ we preference. While it may help if $p(x|D)$ is a powerful generative model, VSD does not require it to be so. However, we would also not want to over-fit $p(x|D)$, thereby placing low probability mass on areas we may expect there to be feasible designs. This is a common consideration in Bayesian modelling approaches. To this end, a user could consider using a pre-trained model (e.g. ESM, ProtGPT etc) as a prior in the VSD framework. Pragmatically, we find using a small held-out validation set, and then early stopping maximum likelihood fitting of $p(x|D)$ to be an effective and simple method.
> > >
> > > Contrast this to competing approaches like latent space optimisation (LSO) [tripp2020sample, gonzalez2024survey], where an encoder, $p(z|x, D)$, and a decoder $p(x|z, D)$ are both required for operating BO in the latent space, $z$. Constructing such a latent space in a high-dimensional low data regime is a real challenge, and can be highly detrimental to a method's performance if not done well. Much of the literature cited by reviewer xamp is concerned with rectifying this issue -- which VSD is able to side-step.
> > >
> > > Lastly, VSD, CbAS, DbAS and Random all make use of some form of $p(x|D)$ either for prior and or initialisation of the variational distribution. We can see from our BBO experiments that these methods outperform PEX, AdaLead and BORE on the higher-dimensional GFP and AAV datasets. The training data (used for both the CPE and prior/initial $q$) for these experiments is set to be low in fitness (much lower than the respective wild-types), and uses 2000 examples. It is worth noting that even random samples (Random method) from $p(x|D)$ from the LSTM and transformer models outperform PEX, AdaLead and BORE. We are incorporating a new high dimensional BBO experiment, based on Ehrlich functions, where we only use 128 training points -- and again we perceive a strong benefit to using the information in this dataset for p(x|D). See Appendix C.4 for an early result, we will very soon be expanding upon these results in the BBO experimental section of the main document.

---

### Official Review · Reviewer_z76B · 2024-10-31

**Soundness:** 3
**Presentation:** 2
**Contribution:** 3
**Rating:** 6
**Confidence:** 2

**Summary:**

The paper develops the variational search distribution method to solve the active search problem in biological design. VSD estimates the super level-set distribution in a sequential manner by generating batches of data points for sequential experiments. Empirical results on optimizing protein fitness in several datasets showcase the effectiveness of VSD.

**Strengths:**

- The paper formulates the batch active search problem in the variational inference framework and provides theoretical guarantees to the learned distribution based on the sequentially attained data.
- Experimental results on real-world biological datasets demonstrate the practical use of the algorithm and its effectiveness to solve the problem.

**Weaknesses:**

- The precision of VSD and most other methods is decreasing with more rounds in TrpB and TFBIND8 datasets while the recall values are in general low. However, an ideal method should achieve a better estimation of the ground truth super level-set distribution as more samples are collected. This may be due to the initial training set size being too large or the fitness landscape being easy to model. How do the models perform with a smaller initial training set size?
- How is VSD compared with the simple and commonly used directed evolution method?

**Questions:**

- How robust are the results to the selection of the threshold $\tau$ and the batch size $B$?
- While the reviewer is not familiar with the field, could the authors give some intuitions about the difference between VSD and active learning approaches like Bayesian optimization, and why VSD is better?

---

> ### Author Response · Authors · 2024-11-22
>
> Thank you to reviewer z76B for the insightful questions. We will attempt to address them all now.
>
> __Weaknesses__:
>
> *The precision of VSD and most other methods is decreasing with more rounds in TrpB and TFBIND8 datasets while the recall values are in general low. However, an ideal method should achieve a better estimation of the ground truth super level-set distribution as more samples are collected. This may be due to the initial training set size being too large or the fitness landscape being easy to model. How do the models perform with a smaller initial training set size?*
>
> For a difficult problem or a hard start, we may reasonably expect recall to be low. There could be multiple reasons for precision to be decreasing though. Firstly, we should clarify that we are not counting repeated sequences recommendations in our definition of precision, which will contribute to this effect -- and as $T \to \infty$ precision must go to 0 (if we did not truncate the normaliser at |S|), I.e. when the fitness landscape is being exhausted of novel feasible designs. In this case, reducing the size of the training data may allow for more feasible designs to be discovered, particularly in the case of the TFBIND8 dataset. However, making this change could also have the effect of worsening the performance (in precision and recall) of the algorithms initially, potentially confounding the issue. Another likely reason for the issue is that the methods are simply no longer exploring the fitness landscape as the experiment progresses, and so a finding fewer and fewer novel sequences -- which is what we suspect is happening for TrpB. If we can manage in time, we will attempt to construct an ablation study that teases apart these potential causes.
>
> *How is VSD compared with the simple and commonly used directed evolution method?*
>
> We expect VSD to dramatically outperform directed evolution (DE). This is because we already compare to an improved version of DE -- AdaLead, which uses a fitness prediction model to inform the sequences selected for experimentation.
>
> __Questions__:
>
> *How robust are the results to the selection of the threshold and the batch size ?*
>
> For the fitness landscapes experiments and setting, the threshold is a fixed quantity given by nature or experimental constraints. For the BBO setting, adaptive setting of the threshold is important, as it controls the exploration/exploitation trade-off (lower settings allow for exploration, whereas higher settings encourage exploitation of known good candidates). We have yet to formulate an optimal schedule for this parameter as has been done for $\\beta_t$ in UCB [srinivas2010gaussian]. We will attempt to run an ablation study on this for the BBO experiments. We note however that VSD, CbAS, DbAS and BORE all use the same threshold function. A heuristic method we find works well for BBO is to use Eqn. 18 with $p _0 = 0.7$ to $0.8$, and then to choose an $\eta$ such that by round $T$, $p _T = 0.99$.
>
> For a given experimental budget, a batch size of 1 would be optimal. However, this is not a quantity that is typically wholly within our control, or we must trade-off other costs (e.g. experimental setup costs) with batch size. There is a well understood trade-off between batch size and performance for this class of algorithm. For a good theoretical discussion, we recommend the discussion about the _adaptivity gap_ in Ch.~11.3 [garnett2023bayesian].
>
> *While the reviewer is not familiar with the field, could the authors give some intuitions about the difference between VSD and active learning approaches like Bayesian optimization, and why VSD is better?*
>
> Definitely, for a concrete comparison, we would refer the reviewer to Appendix F where we show that VSD can be viewed as lower bound on Bayesian optimization with the probability of improvement acquisition function.
>
> Intuitively, (traditional) Bayesian optimization is formulated to optimize over candidates, $\\mathbf{x}$, directly, to solve $\\mathrm{argmax}_\\mathbf{x} f(\\mathbf{x})$, but where $f$ is a black box. Typically we estimate $f$ with a Gaussian process surrogate, and optimize using this surrogate (in combination with an acquisition function) in place of $f$. However, in this work we consider $\\mathbf{x}$ as a high dimensional discrete object, and so gradient based optimization of the surrogate or enumeration of all candidates is not possible. VSD allows us to use gradient based techniques to directly optimize a _generative_ model over $\\mathbf{x}$ instead, which we can still use to find the optimal $\\mathbf{x}^*$ -- or a distribution over feasible $\\mathbf{x}$ if we wish. Ultimately, traditional Bayesian optimization approaches are not applicable in the situations for which we use VSD.

---

### Official Review · Reviewer_1vHR · 2024-11-06

**Soundness:** 3
**Presentation:** 2
**Contribution:** 3
**Rating:** 6
**Confidence:** 3

**Summary:**

The authors propose a black box varoiational inference approachfor discrete designs generation. The authors derive asymptotic convergence rates for learning the true conditional generative distribution of designs. Compelling results on high dimensional sequence-design problems are demonstrated.

**Strengths:**

* The problem is important as it has applications in pharmaceutical drugs/enzyme design.
* The paper paper is well written and the method is sound
* Experimental results on high dimensional datasets demonstrate superiority of the approach

**Weaknesses:**

* The method lacks novelty, it's based on putting together blocks that have already been proposed in the litterature
* The paper clarity can be improved with an overview plot of the method

**Questions:**

* What's 'x' in the title of Figure 1?
* What are the limitations of this approach?
* How is diversity within a batch enforced?
* The reverse KLD is known to result in mode collapse. Why wasn't this an issue?
* Which variation reduction method did you use for the gradient estimator?

---

> ### Author Response · Authors · 2024-11-22
>
> Thank you to reviewer 1vHR for the constructive criticism and recommendations. We will attempt to address all of your concerns and questions.
>
> __Weaknesses__:
>
> *The method lacks novelty, it's based on putting together blocks that have already been proposed in the literature.*
>
> We respectfully disagree that this is a weakness. That is, we feel this is a valuable ``novel combination of well-known techniques'', which counts as original as per the [NeurIPS 2024 reviewer guidlines](https://neurips.cc/Conferences/2024/ReviewerGuidelines). Furthermore, the analysis is entirely novel for this type of method, and required extending existing results, e.g. Theorem D.1 which is a first for GP-PI.
>
> *The paper clarity can be improved with an overview plot of the method*
>
> This is a good recommendation. We will include this in the next version of the paper.
>
> __Questions__:
>
> *What's `x` in the title of Figure 1?*
>
> `x` in figure 1 refers to the white x-mark in figure 1a -- the maximum of the fitness landscape. We will clarify this in the figure description (white `x`-mark).
>
> *What are the limitations of this approach?*
>
> One limitation is that using an autoregressive model for the variational distribution, like an LSTM or transformer, can be computationally demanding in a method like VSD compared to say, CbAS/DbAS. This is because the score function gradient estimator requires samples from the variational distribution for gradient estimates (Eqn. 9) each optimizer iteration, which are relatively expensive. Whereas CbAS/DbAS used fixed samples, and then just maximize the weighted log-likelihood of the variational distribution under these samples. However, VSD is able to adapt its samples along with the variational distribution during optimization unlike CbAS/DbAS. See Eqn. 14 and Table 1 for a comparison of these gradient estimators. Other limitations include theoretical assumptions, with guarantees that are only valid for finite discrete domains for now and can be extended in future work. We will expand our discussion on limitations in the revision.
>
>
> *How is diversity within a batch enforced?*
>
> We do not explicitly enforce batch diversity in VSD, or in any of the other methods in the paper, i.e. any method can re-recommend the same sequence. Rather batch diversity is rewarded though the KL divergence term in Eqn. 7, assuming a diverse prior over sequences. Without this term, the variational distribution tends to collapse to a delta distribution, as we see with the BORE method. See Appendix C.2 for more results using a diversity measure.
>
> *The reverse KLD is known to result in mode collapse. Why wasn't this an issue?*
>
> We assume the reviewer is referring to mode collapse in a variational inference context? I.e. the reverse KLD can encourage a compact variational distribution compared to other divergences, such as forward KL and expectation propagation.
>
> We suspect the basic reason we do not see mode collapse when using the mean field variational distributions is that categorical distributions, which we use to model tokens (amino/nucleic acids), are inherently multi-modal, and so even the most basic variational distributions can exhibit per-token multi-modality.
>
> Furthermore, for the higher dimensional problems, we suspect the LSTM and transformer variational distributions are flexible enough that they can model the true posterior distribution over these short sequences (compared to natural language tasks). It is also shown in [knoblauch2019generalized] that the reverse KL divergence/ELBO is always preferable to the alternatives when trying to optimize the true Bayesian posterior.
>
> *Which variation reduction method did you use for the gradient estimator?*
>
> We just used the same simple baseline method from [daulton2022bayesian] -- where we subtract an exponentially smoothed average of the previous ELBO values. This is mentioned at the end of section 2.2.

---

### Official Review · Reviewer_xamp · 2024-11-07

**Soundness:** 4
**Presentation:** 3
**Contribution:** 3
**Rating:** 8
**Confidence:** 5

**Summary:**

This paper casts sequential black-box optimization as a variational inference (i.e. amortized optimization) problem, and uses this perspective to unify a collection of different black-box optimization algorithms under a common theoretical framework and presents some proof of concept results on easy sequence optimization tasks.

#### UPDATE - 12/02/2024 ####
After extended discussion and an exceptionally thorough response from the authors, my concerns have been addressed and I am recommending acceptance. I believe this paper presents a very nice conceptual view of the topic of active generation and the empirical results raise interesting questions. I encourage my fellow reviewers to review the updated manuscript and re-evaluate their scores. I have left my original review unaltered for any interested readers.

**Strengths:**

This paper demonstrates a clarity of thought and composition that is commendable, I particularly enjoyed the related work section.

Likewise I do not have any major concerns regarding the technical soundness of the results presented.

As a good conceptual introduction to the topic, I think this draft could be useful to researchers new to the topic with some revisions.

**Weaknesses:**

I have two general impressions of this paper.

First, it seems like the authors have not really chosen a direction for the paper. There are at least three different directions here, A) a unifying view of sequential black box optimization algorithms, B) a practical algorithm for sequential BBO, and C) theoretical analysis of convergence rates of a particular sequential BBO algorithm under strong assumptions. I would suggest you pick no more than two directions, preferably one. I actually think this particular subfield could really benefit from a more holistic perspective of the work that has been done, as I constantly see minor variations of these algorithms in my social media feed and review stack with no apparent awareness of the relationships between them. From what I can tell from this draft, it seems that A and C likely play more to your strengths.


Second, the authors seem blissfully unaware of a substantial body of work on this topic. To be quite candid, the paper reads like it was written circa September 2021. This is not mere rhetoric. The most recent baseline the authors consider was published at ICML 2021. It is also odd that two of the baselines you did include, DbAS and CbAS, are not even designed for the sequential setting. As a very active researcher in this exact area, I struggle to understand who this paper is for and how the authors pictured their place in the broader dialogue on this topic. I am sure you worked very hard on this paper and I commend your effort, but I honestly believe the best advice I can give you is to talk to more people working on this topic, preferably from outside your immediate academic circle. While it is difficult to hear this feedback, one of the functions of peer review is to reveal "unknown unknowns". I want to be sure this review is constructive, so I will provide some key references if you are serious about diving into this topic. You should also consider making use of tools like [Connected Papers](https://www.connectedpapers.com/) to improve your literature review process and avoid this situation in the future.

You can start with [A survey and benchmark of high-dimensional Bayesian optimization of discrete sequences](https://arxiv.org/abs/2406.04739). This work is the most up-to-date complete survey on the topic I have seen, and the benchmarking rigor is notably good. This paper is associated with two repositories, [poli](https://github.com/MachineLearningLifeScience/poli) and [poli-baselines](https://github.com/MachineLearningLifeScience/poli-baselines). The former contains a suite of test functions that are much more up to date than the combinatorially complete landscapes considered in this paper, and the latter contains a suite of baseline solvers. You may even want to consider contributing your method as a solver to poli-baselines at some point.

Some key axes of variation to consider:

How is the optimization problem solved? Most fall into one of three categories, directed evolution (which you seem to be familiar with based on your inclusion of AdaLead and PEX), generative search with explicit guidance, e.g. [2, 3, 4, 5, 6], and generative search with implicit guidance [7, 8], which can also be seen as a kind of amortized search. I could cite more papers but I believe I have made my point. Algorithms also differ in their handling of constraints, and their approach to managing the feedback covariate shift induced by online active data collection by an agent.

In particular I will draw your attention to [a tutorial for LaMBO-2](https://github.com/prescient-design/cortex/blob/main/tutorials/4_guided_diffusion.ipynb) if you want to start considering more up to date baselines, however I would recommend using the solver interface provided in poli-baselines for actual experiments. You may also be interested in Ehrlich functions if you would like a convenient test function that is much more difficult to solve than small combinatorially complete landscapes but still easy to work with [9]. Ehrlich functions are available in [a small standalone package](https://github.com/prescient-design/holo-bench) or [as part of the poli package](https://machinelearninglifescience.github.io/poli-docs/using_poli/objective_repository/ehrlich_functions.html).

While I'm sure this is not the outcome you hoped for, science is a dialogue, and good science requires awareness of what is happening outside your academic niche. Hopefully my feedback is clear and actionable enough to benefit this work and your progression as a scientist.

References

- [1] González-Duque, M., Michael, R., Bartels, S., Zainchkovskyy, Y., Hauberg, S., & Boomsma, W. (2024). A survey and benchmark of high-dimensional Bayesian optimization of discrete sequences. arXiv preprint arXiv:2406.04739.
- [2] Tripp, A., Daxberger, E., & Hernández-Lobato, J. M. (2020). Sample-efficient optimization in the latent space of deep generative models via weighted retraining. Advances in Neural Information Processing Systems, 33, 11259-11272.
- [3] Stanton, S., Maddox, W., Gruver, N., Maffettone, P., Delaney, E., Greenside, P., & Wilson, A. G. (2022, June). Accelerating bayesian optimization for biological sequence design with denoising autoencoders. In International Conference on Machine Learning (pp. 20459-20478). PMLR.
- [4] Gruver, N., Stanton, S., Frey, N., Rudner, T. G., Hotzel, I., Lafrance-Vanasse, J., ... & Wilson, A. G. (2023). Protein design with guided discrete diffusion. Advances in neural information processing systems, 36.
- [5] Maus, N., Jones, H., Moore, J., Kusner, M. J., Bradshaw, J., & Gardner, J. (2022). Local latent space bayesian optimization over structured inputs. Advances in neural information processing systems, 35, 34505-34518.
- [6] Maus, N., Wu, K., Eriksson, D., & Gardner, J. (2022). Discovering many diverse solutions with bayesian optimization. arXiv preprint arXiv:2210.10953.
- [7] Tagasovska, N., Gligorijević, V., Cho, K., & Loukas, A. (2024). Implicitly Guided Design with PropEn: Match your Data to Follow the Gradient. arXiv preprint arXiv:2405.18075.
- [8] Chen, A., Stanton, S. D., Alberstein, R. G., Watkins, A. M., Bonneau, R., Gligorijevi, V., ... & Frey, N. C. (2024). LLMs are Highly-Constrained Biophysical Sequence Optimizers. arXiv preprint arXiv:2410.22296.
- [9] Stanton, S., Alberstein, R., Frey, N., Watkins, A., & Cho, K. (2024). Closed-Form Test Functions for Biophysical Sequence Optimization Algorithms. arXiv preprint arXiv:2407.00236.

**Questions:**

The following questions are sincere:

- Who is the audience for this paper?

- What questions is this paper answering?

- What does the variational inference framing get us in the end? Access to a set of tools for theoretical analysis?

---

> ### Author Response · Authors · 2024-11-22
> **Weaknesses (1)**
>
> We thank the reviewer for the constructive and actionable feedback. We believe their input will result in a much stronger experimental evaluation of VSD. This is something we have already been able to make progress on -- see the most recent draft of the paper Appendix C.4, which may augment or replace the existing BBO experiments. We intend to finish the experimental results by the conclusion of the discussion period.
>
> We would like to note that a lot of the related work pointed out by the reviewer, or benchmarks/baselines to compare against are considered concurrent/contemporaneous works under the [ICLR 2025 Guidelines](https://iclr.cc/Conferences/2025/FAQ), and so should not be used as a basis for decision making. That said, we realize a lot of these recommendations were for our benefit to make the work stronger and, therefore, we are very grateful to the reviewer for this.
>
> __Weaknesses__:
>
> *First, it seems like the authors have not really chosen a direction for the paper. ...*
>
> We respectfully disagree with this point. We believe having all three aspects, A (unifying view), B (practical algorithm) and C (theoretical analysis) strengthens our paper's contribution. In fact, these three elements align gracefully  with the guidelines of top ML conferences such as [NeurIPS](https://neurips.cc/Conferences/2015/PaperInformation/EvaluationCriteria) on writing a good machine learning paper. We would also like to note that our major motivating factor is to formulate a practical algorithm (B). As such, VSD is a framework that allows a practitioner to readily adapt the underlying components to the task at hand:
>
> - Simple and scalable off-the-shelf class probability estimators can be used -- we do not even require model ensembles or predictive uncertainties, dramatically simplifying implementation.
> - The prior and variational distributions are easily adaptable to the problem at hand. Very simple distributions can be used, up to complex models like pre-trained decoder only (GPT-like) architectures. Various design constraints can also be encoded in these generative models, e.g. we can use various masking strategies and context with transformers to only sample from certain sites in a sequence.
> - Fewer specialized components than many competing methods. E.g. We do not require (sometimes specialized) encoders like latent space optimization (LSO) methods.
>
> *Second, the authors seem blissfully unaware of a substantial body of work on this topic ...*
>
> Though VSD's primary motivation is not black-box optimization (BBO) -- rather it is a variant of active search -- we agree with the reviewer's feedback in that it was an oversight on our part to not include LSO and related works (LaMBO) in our related work section, as we do compare on BBO tasks. We are aware of this body of research, and in fact a major motivating reason for VSD was to circumvent the need for construction (and adaptation) of a latent space entirely. We have incorporated this literature into section 3, and included key methods in Table 2.
>
> *DbAS and CbAS, are not even designed for the sequential setting ...*
>
> While it is true that the original authors have designed these for offline black box optimization tasks, there is precedent in the literature for using these methods as baselines for sequential optimization tasks, e.g., AdaLead, PEX, LSO, [sinai2020adalead, ren2022proximal, tripp2020sample]. The original authors also mention they can be used (with a $\\tau=\\max \\{ y : y \\in \\mathcal{D}_N \\} $) for exploitation-focused sequential optimization.
>
> *The former contains a suite of test functions that are much more up to date than the combinatorially complete landscapes considered in this paper,*
>
> Two of these combinatorially complete landscape tasks (DHFR, TrpB) have not been used in the machine learning literature yet to our knowledge, only being released in 2023-2024 [papkou2023rugged, johnston2024combinatorially]. Also, we do not use these tasks for BBO -- but rather to test the ability of our, and other, methods for super-level set distribution estimation. This is a very challenging task, and we believe these datasets still provide challenging benchmarks, even if they are not challenging for BBO.

---

> ### Author Response · Authors · 2024-11-22
> **Weaknesses (2)**
>
> *I will draw your attention to a tutorial for LaMBO-2 if you want to start considering more up to date baselines, however I would recommend using the solver interface provided in poli-baselines for actual experiments.*
>
> Thanks for this recommendation. Even though this is concurrent work, we have been able to compare VSD and LaMBO-2 on the Ehrlich (M=32) example [here](https://github.com/MachineLearningLifeScience/poli-baselines/tree/main/examples/07_running_lambo2_on_ehrlich). Please see the updated version of our paper, Appendix C.4 for an early result -- in which we show VSD outperforming LaMBO-2.
>
> We are currently expanding upon these results, and we intend to compare to more of the baselines in our paper (CbAS, DbAS, BORE), and also for some configurations of the Ehrlich function that are listed on the [HDBO Benchmarks](https://machinelearninglifescience.github.io/hdbo_benchmark/benchmarks/). We intend to put these experiments in the main paper -- potentially in place of the existing BBO tasks.
>
> *How is the optimization problem solved? Most fall into one of three categories ...*
>
> Our approach is fundamentally different from LSO and more closely related to probabilistic reparameterisation [daulton2022bayesian] or continuous relaxations for discrete BO [michael2024continuous], which transform the optimization over discrete inputs into an optimization over the expectation under a discrete distribution. These approaches are not properly categorized under the survey paper [gonzalez2024survey], as only one of them appears under the structured inputs category, which also contains manifold BO methods that are essentially different. The main difference between our approach and the existing probabilistic reparameterisation methods for BO is that we include a KL penalty to the acquisition function, which leads to a very natural interpretation of our framework as approximate inference over level-set distributions. In addition, the posterior resulting from the optimization allows us to sample diverse solutions, has a Bayesian interpretation, etc.
>
> LSO methods also have to implement an encoding direction, since their objective function surrogate models are built on the latent space (cf. LaMBO-1/2). VSD does not need to encode anything. VSD's models operate on the sequences directly. Besides that, [michael2024continuous] presents arguments about the issue with learning conventional GPs over the latent space. One is that distances in the latent space are not necessarily preserved, and translation-invariant kernels rely on distances to infer correlations. Two sequences that end up close to each other may not actually be that close to each in the original space under some suitable metric, while sequences encoded far apart could in fact be neighbors. These pathological cases could confuse GPs and similar models and deteriorate LSO methods' performances due to misleading encoder models, issues which VSD can sidestep.

---

> > ### Author Response · Authors · 2024-11-22
> > **Questions, Changes**
> >
> > __Questions__:
> >
> > *Who is the audience for this paper? (I struggle to understand who this paper is for and how the authors pictured their place in the broader dialogue on this topic)*
> >
> > The intended audience of this paper is for machine learning practitioners and researchers who are concerned with understanding fitness landscapes as in, for example, the synthetic biology space (cf. recent works such as [papkou2023rugged, johnston2024combinatorially, kirjner2024improving, sandhu2024computationa]). Our take is to formulate this problem as a generalization of active search [garnett2012bayesian] for modeling the super level-set density of feasible candidates. It so happens that VSD is also a powerful method for BBO (and generalizes BORE [tiao2021bore]) when used with an adaptive threshold.
> >
> >
> > *What questions is this paper answering?*
> >
> > As pointed out in the introduction, this paper is concerned with sequentially learning generative models (e.g. decoder transformers) for feasible sequences, under the theme of understanding fitness landscapes. Pragmatically -- VSD could be viewed as solving a similar problem as active learning, but instead of attempting to efficiently learn the best predictor, $E[y|x]$ or $p(y|x)$, we are concerned with efficiently learning a generative model $p(x|y)$ that can be then used for further down-stream tasks. For example, we can use VSD to learn a generative model for feasible sequences from a relatively inexpensive high throughput screening assay, e.g., on enrichment factors from microfluidics \citep{thomas2024engineering}, which we can then use to inform (lower variance) experimental designs for specific tasks.
> >
> > To aid understanding and categorization we have modified the draft to label this problem as "active generation" (as distinct from, but related to, active search and active learning).
> >
> > *What does the variational inference framing get us in the end? Access to a set of tools for theoretical analysis?*
> >
> > Variational inference gives us a number of desirable features:
> >
> > - An intuitive loss function for the exact problem we wish to solve, see Eqn. 7, that trades off BBO (e.g. GP-PI) with trust-region like regularization;
> > - A Bayesian interpretation of the density, $p(x|y)$, cf. [knoblauch2019generalized];
> > - From above, as you say, a set of tools for theoretical guarantees, which translate to real performance;
> > - A modular framework in which we can use ``off-the-shelf'' components for the task at hand (e.g. CNN class probability estimators, our choice of prior and variational posterior models) that are supported by our guarantees.
> >
> > __Changes__:
> >
> > In summary, we will implement these changes:
> >
> > - Incoporate LSO and LaMBO-1/2 into our related work section, and Table 2 (done).
> > - Add BBO experiments based on `poli` benchmarks, in particular the Ehrlich functions (we will attempt sequence lengths 15, 32 and 64).
> > - Based on the outcomes of the above, possibly move (some of) our current BBO experiments to the appendix.

---

> > > ### Comment · Reviewer_xamp · 2024-11-22
> > > **response under consideration**
> > >
> > > thanks for your detailed response, I will read it carefully and consider. You can expect my response on Monday, Nov 25th since the discussion period ends on Nov 26th. Have a nice weekend!

---

> ### Comment · Reviewer_xamp · 2024-11-26
> **score improved**
>
> I've read your response and re-read sections of your paper. I think it's a substantial improvement and will increase my score to 5. The reason my score is not higher is because I believe the authors are still missing or downplaying important conceptual connections to prior work, particularly the field of guided/conditional generation and its relation to discrete BBO. I'm making this a sticking point because I believe you clearly have the ability and understanding to make these connections clear, and I believe you will have a much more impactful paper if you do.
>
> **Active generation vs. black-box optimization**
>
> It is true that finding a single solution $x^* \textrm{ s.t. } f(x^*) = \max_{x \in \mathcal{X}} f(x)$ is not the same as active generation. It is also true that many methods for discrete BBO (particularly LaMBO-2) *already cast the problem in terms of active generation*, namely actively learning to sample from $p(x | y)$, where $y$ is some event indicating the optimality of the outcome you wish to attain. Methods like LaMBO-2 can and do make use of a classifier head to guide sample output towards the satisfaction of objective thresholds, for example see [1]. This capability is particularly useful for the handling of constraints. I particularly wish to draw your attention to the similarity between Eq 7. in your paper and Eq. 4 in [2]. Hopefully it is clear to you that if you take the value function to be $\log \alpha_{PI}$ then the two methods are pursuing the same objective. You are amortizing the solution of that problem into the weights of a network during training, whereas LaMBO-2 reparameterizes the problem with latent variables and explicitly searches for solutions to the problem at test time. This is a real and important difference, and the one you *should* be discussing. There is very active discussion on the merits of amortized vs test-time search, and your contribution here could be quite valuable if the situation was presented clearly.
>
> To be clear, I see potential here, and your initial results on Ehrlich functions are promising! I would really encourage a "big-tent" perspective here. It's easy (especially during review) to mistake differences in focus and framing between papers as fundamental differences in approach when they are in fact solving essentially the same problem from a different point of view. Read prior work to the same depth with which you want your work to be read.
>
> **Question regarding Table 2**
>
> How do you determine whether a method does or does not seek to find rare events in the search space (R1)? Does the method have to explicitly state that in the introduction of the paper in which it first appeared? On what grounds do you believe LaMBO 1/2 (or some of the other baselines) do not seek this goal? I think most people who work on hard BBO problems take that condition as a given, otherwise the search problem would be trivial.
>
>
> **References**
>
> - [1] Park, J. W., Stanton, S., Saremi, S., Watkins, A., Dwyer, H., Gligorijevic, V., ... & Cho, K. (2022). Propertydag: Multi-objective bayesian optimization of partially ordered, mixed-variable properties for biological sequence design. arXiv preprint arXiv:2210.04096.
>
> - [2] Gruver, N., Stanton, S., Frey, N., Rudner, T. G., Hotzel, I., Lafrance-Vanasse, J., ... & Wilson, A. G. (2023). Protein design with guided discrete diffusion. Advances in neural information processing systems, 36.

---

> ### Comment · Reviewer_xamp · 2024-11-26
> **a note on concurrency**
>
> I included some concurrent work in the references I provided to the authors to bring them as much up to speed as possible given the time I had to write the review. I hope the authors will not use this good-faith gesture to engage in a fallacy of composition to argue my feedback should be disregarded for decision-making. Also please note the year in the LaMBO-2 citation was incorrect, it appeared in the proceedings of NeurIPS 2023 and has already been further developed by [1] (which appeared in the proceedings of ICML 2024). I can expand on this point further, but I would prefer to keep the focus of the discussion on more substantive aspects of this and prior work.
>
>
> References
> - [1] Klarner, L., Rudner, T. G., Morris, G. M., Deane, C. M., & Teh, Y. W. (2024). Context-guided diffusion for out-of-distribution molecular and protein design. arXiv preprint arXiv:2407.11942.

---

> ### Author Response · Authors · 2024-11-28
>
> We thank reviewer xamp for the score increase – and we truly appreciate the help and time dedicated in making sure this work is topical and impactful.
>
> **Updated related work discussion**
>
> To this end, we have included a discussion of the direct conditional generative approach of VSD, and the guided generation of LaMBO-2 in the related work section (last paragraph) for the purposes of active generation. We have also added a small note to our conclusion. We are in total agreement that LaMBO-2 (and 1) can also be considered to be solving this problem, especially when using something like a PI acquisition function. Generation of samples from a pareto set (as opposed to a super-level set) potentially also fits this framing nicely.
>
> _How do you determine whether a method does or does not seek to find rare events in the search space (R1)? … On what grounds do you believe LaMBO 1/2 (or some of the other baselines) do not seek this goal?_
>
> We consider the task of BBO (finding the maximiser) a less general task than (R1) – which is to find (and generate) a set, $\mathcal{S}$, of rare feasible solutions (which presumably includes the maximizer), ideally $|\mathcal{S}| > 1$. We do not mean to trivialise BBO, but merely whish to make a distinction. However, since LaMBO 1/2 are doing active generation, and can use PI, we have also included them as fulfilling R1 (we apologise, the last draft we could upload does not have LaMBO-1 ticked – but in our latest working draft it is). Furthermore, finding the pareto set in a MOO setting, for which LaMBO 1 and 2 have been designed, naturally also fits this definition, where we can define $\mathcal{S}$ as the set of non-pareto dominated solutions.
>
> __Updated experiments__
>
> We have also included a new BBO experiment in the main text (sec. 4.3) using the Ehrlich functions, and comparing to LaMBO-2, among other baselines. This is on the poli implementation of the Ehrlich functions, in which VSD performs favourably, and we include experiments on the holo implementation in the appendix (C.2 and C.3). VSD (and CbAS) performs significantly worse than LaMBO-2 on the holo-64 function –  We are currently investigating this and suspect this is from our strategy of training the prior (it could be overfitting as it is only trained on 128 samples), thereby unduly dominating the CPE in our ELBO objective.
>
> We are currently investigating more robust training schemes (early stopping/leave-out) for priors. We have also noted that LaMBO-2 weights the contribution of KL in its objective (Equation 4, Gruver et. al 2023), which is a similar strategy to beta-VAE and Power-VI for down-weighting the effect of a mis-specified prior. We have some preliminary results (not included) that show for this Ehrlich function, applying a weight of $\lambda = 0.25$ to the KL term in our ELBO (the same as LaMBO-2 in the configuration we are using) improves performance of VSD and CbAS significantly. As long as the prior still has support over all $\mathcal{X}$ this strategy should not overly affect our convergence results. We may include a section on this in the appendix of a future draft.
>
>
> __Concurrency__
>
> _I hope the authors will not use this good-faith gesture to engage in a fallacy of composition to argue my feedback should be disregarded for decision-making._
>
> We have no intention of doing so – and have already included many of these references in the manuscript. We thank reviewer xamp for sharing this literature.

---

> > ### Comment · Reviewer_xamp · 2024-12-02
> > **score improved to 8**
> >
> > thank you for a scholarly rebuttal. you have resolved my objections and we have reached agreement. I think you have a very nice paper on your hands and I will argue for its acceptance. As a minor note, because your paper has such ambitious scope and so many facets, another editing pass just focused on streamlining and focus on the main points of the story may make the paper more accessible to a broader audience. Based on your attention to detail during the rebuttal, I feel confident this is already important to you and feel comfortable leaving matters to your judgement. Until next time, cheers!

---

> ### Comment · Reviewer_xamp · 2024-12-02
> **one more paper...**
>
> your definition of R1 reminds me of a very under-appreciated paper, [Stopping Bayesian Optimization with Probabilistic Regret Bounds](https://arxiv.org/abs/2402.16811) [1]. You may find it interesting and worth mentioning somewhere for more theoretically inclined readers. I think I have also heard "Bayesian Satisficing" used in place of "Bayesian Optimization" to indicate a more general stopping criterion.
>
> References:
>
> - [1] Wilson, J. T. (2024). Stopping Bayesian Optimization with Probabilistic Regret Bounds. arXiv preprint arXiv:2402.16811.

---

> ### Author Response · Authors · 2024-12-03
> **Thank you & Experimental update**
>
> Thank you again to reviewer xamp -- their input has dramatically improved our submission and our understanding of where VSD sits in relation to the literature. Upon your advice, we will continue to sharpen the focus of the paper to make it more approachable for a broader audience.
>
> Furthermore, thank you for highlighting "Stopping Bayesian Optimization with Probabilistic Regret Bounds" -- there are some interesting connections here with our work.
>
> ---
>
> For your information, we have managed to track down the reason for VSD and CbAS performing significantly worse that LaMBO-2 on the holo-64 Ehrlich function. It is not to do with the prior distribution as we previously thought; rather it stems from the quantisation of the labels in the Ehrlich functions. There are two effects we have noticed, which interact:
>
> 1. Quantised targets and moving thresholds, $\tau_t$. We have noticed that quantisation in the targets can lead to situations in which there can be a severe attrition of the positive labels when the threshold increases (e.g from over 100 positive labels to one) when using a CPE.
> 2. Under-training the CPE. We also noticed that in latter rounds, we were under-training the CPEs as the data set increased, since we were using early stopping as a training strategy for the small initial dataset (128).
>
> The net result of effects (1) and (2) was that as the rounds progressed and the threshold increased, the class imbalance for the CPEs could get suddenly worse, and the training regime was not compensating for this label imbalance. This resulted in the reward signal from the CPE becoming drastically weaker than the KL penalty, resulting in CbAS and VSD "giving up".
>
> The fix for this is straightforward and generally applicable -- we only allow the threshold function to increase if there is a minimum number of labelled instances (e.g. 10). And we allow the CPE to train for many more iterations. With dropout (p=0.2), early round overfitting is not an issue.
>
> Early experimental results are very encouraging and place VSD on a more even footing with LaMBO-2 for this experiment.
>
> As an aside: the quantisation of $y$ means there is no way of constructing an RKHS (or Hilbert space) solely comprised of Ehrlich functions (or quantised functions in general) since, e.g., their addition and linear combinations would not lead to other Ehrlich functions. So this situation is no longer covered by our theoretical results, or potentially many convergence results in the BO literature that rely on the black-box function being representable by a RKHS. This is an interesting benchmark, as it tests these methods in a setting not necessarily covered by current theoretical results.

---

### Author Response · Authors · 2024-11-22

We thank the reviewers for their constructive comments and criticisms of our work, and for the feedback that has made this paper stronger.

The major concerns and criticisms of our work come from reviewer xamp:

1. We have neglected to cite major works in the related setting of latent space optimization (LSO), and where VSD sits in relation to this work.
2. We have neglected to compare to LSO on difficult BBO baselines.

We discuss these major criticisms below, but we have also updated our manuscript to (1) include LSO in our related work section and Table 2 and (2) we have benchmarked VSD against LaMBO-2 on the suggested Ehrlich function BBO task, in which VSD gets the best performance. The latter experiment is currently in appendix C.4, but we are working on improving and expanding it, and we aim to move it to the main paper by the end of this discussion period.

Update: also based on xamp's feedback -- we have slightly reframed the problem we are solving as "active generation" (as distinct from, but related to active learning, active search and BBO), and updated the introduction and problem formulations to clarify this.

---

> ### Author Response · Authors · 2024-11-22
> **References**
>
> We make use of the below references in our rebuttals:
>
> [daulton2022bayesian] Samuel Daulton, Xingchen Wan, David Eriksson, Maximilian Balandat, Michael A Osborne, and
> Eytan Bakshy. Bayesian optimization over discrete and mixed spaces via probabilistic reparame-
> terization. Advances in Neural Information Processing Systems, 35:12760–12774, 2022.
>
> [garnett2023bayesian] Roman Garnett. Bayesian optimization. Cambridge University Press, 2023.
>
> [garnett2012bayesian] Roman Garnett, Yamuna Krishnamurthy, Xuehan Xiong, Jeff G. Schneider, and Richard P. Mann.
> Bayesian optimal active search and surveying. In Proceedings of the 29th International Confer-
> ence on Machine Learning, ICML 2012, Edinburgh, Scotland, UK, June 26 - July 1, 2012. icml.cc
> / Omnipress, 2012.
>
> [gonzalez2024survey] Miguel Gonz´alez-Duque, Richard Michael, Simon Bartels, Yevgen Zainchkovskyy, Søren Hauberg,
> and Wouter Boomsma. A survey and benchmark of high-dimensional bayesian optimization of
> discrete sequences. arXiv preprint arXiv:2406.04739, 2024.
>
> [johnson2024combinatorially] Kadina E. Johnston, Patrick J. Almhjell, Ella J. Watkins-Dulaney, Grace Liu, Nicholas J. Porter,
> Jason Yang, and Frances H. Arnold. A combinatorially complete epistatic fitness landscape in
> an enzyme active site. Proceedings of the National Academy of Sciences, 121(32):e2400439121,
> 2024. doi: 10.1073/pnas.2400439121.
>
> [kirjner2024improving] Andrew Kirjner, Jason Yim, Raman Samusevich, Shahar Bracha, Tommi S Jaakkola, Regina Barzi-
> lay, and Ila R Fiete. Improving protein optimization with smoothed fitness landscapes. In The
> Twelfth International Conference on Learning Representations, 2024.
>
> [knoblauch2019generalized] Jeremias Knoblauch, Jack Jewson, and Theodoros Damoulas. Generalized variational inference:
> Three arguments for deriving new posteriors. arXiv preprint arXiv:1904.02063, 2019.
>
> [michael2024continuous] Richard Michael, Simon Bartels, Miguel Gonz´alez-Duque, Yevgen Zainchkovskyy, Jes Frellsen,
> Søren Hauberg, and Wouter Boomsma. A continuous relaxation for discrete bayesian optimiza-
> tion. arXiv preprint arXiv:2404.17452, 2024.
>
> [papkou2024rugged] Andrei Papkou, Lucia Garcia-Pastor, Jos´e Antonio Escudero, and Andreas Wagner. A rugged yet
> easily navigable fitness landscape. Science, 382(6673):eadh3860, 2023.
>
> [ren2022proximal] Zhizhou Ren, Jiahan Li, Fan Ding, Yuan Zhou, Jianzhu Ma, and Jian Peng. Proximal exploration
> for model-guided protein sequence design. In International Conference on Machine Learning,
> pp. 18520–18536. PMLR, 2022.
>
> [sandhu2024computational] Mahakaran Sandhu, John Chen, Dana Matthews, Matthew A Spence, Sacha B Pulsford, Barnabas
> Gall, James Nichols, Nobuhiko Tokuriki, and Colin J Jackson. Computational and experimental
> exploration of protein fitness landscapes: Navigating smooth and rugged terrains, 2024.
> Sam Sinai, Richard Wang, Alexander Whatley, Stewart Slocum, Elina Locane, and Eric D Kelsic.
> Adalead: A simple and robust adaptive greedy search algorithm for sequence design. arXiv
> preprint arXiv:2010.02141, 2020.
>
> [srinivas2010gaussian] Niranjan Srinivas, Andreas Krause, Sham Kakade, and Matthias Seeger. Gaussian process optimiza-
> tion in the bandit setting: no regret and experimental design. In Proceedings of the 27th Interna-
> tional Conference on International Conference on Machine Learning, ICML’10, pp. 1015–1022,
> Madison, WI, USA, 2010. Omnipress. ISBN 9781605589077.
>
> [thomas2024engineering] Neil Thomas, David Belanger, Chenling Xu, Hanson Lee, Kathleen Hirano, Kosuke Iwai, Vanja
> Polic, Kendra D Nyberg, Kevin Hoff, Lucas Frenz, et al. Engineering highly active and diverse
> nuclease enzymes by combining machine learning and ultra-high-throughput screening. bioRxiv,
> pp. 2024–03, 2024.
>
> [tiao2021bore] Louis C Tiao, Aaron Klein, Matthias W Seeger, Edwin V Bonilla, Cedric Archambeau, and Fabio
> Ramos. Bore: Bayesian optimization by density-ratio estimation. In International Conference on
> Machine Learning, pp. 10289–10300. PMLR, 2021.
>
> [tripp202sample] Austin Tripp, Erik Daxberger, and Jos´e Miguel Hern´andez-Lobato. Sample-efficient optimization
> in the latent space of deep generative models via weighted retraining. Advances in Neural Infor-
> mation Processing Systems, 33:11259–11272, 2020.

---

### Meta-Review · Area_Chair_GRmd · 2024-12-18

**Metareview:**

This paper considers an active search problem and presents a method to generate new designs of rare desired class under budget constraints. The target problem is of discrete and combinatorial nature. The proposed algorithm is based on variational inference. Theoretical analysis is conducted to understand its performance. All reviewers agree the paper contains interesting and potentially impactful results. One major concern is the lacking of comparison with strong baselines. One reviewer also points out the paper misses some important related work. Overall, this is a nice contribution to active search algorithms.

**Additional Comments On Reviewer Discussion:**

One major concern is the lacking of comparison with strong baselines. One reviewer also points out the paper misses some important related work. The authors have clarified these points and made modifications accordingly.

---

### Decision · Program_Chairs · 2025-01-22

Accept (Poster)